# Self-Supervised Disentanglement via Cluster-Dependent Rotational Equivariance

## Abstract

Conventional self-supervised learning methods extract robust features by enforcing invariance to data augmentations. While effective for obtaining clustered representations, this objective provides limited control over how data variations structure the feature space, hindering disentanglement. Recent methods improve feature space structure by imposing equivariant predictability on feature transformations induced by data augmentations. However, existing approaches suffer from two significant limitations: (i) the incorporation of invariance in their final objective interferes with the learning of neat equivariance; (ii) the imposition of uniform equivariance across all samples forces semantic clusters into a parallel arrangement, leading to reduced inter-cluster distances (for features on the hypersphere). To overcome these issues, we propose in this paper Cluster-Dependent Rotational Equivariance for Disentanglement (CD-RED), a framework that enables learning neat equivariance and uniformly distributed clusters, while further supporting perfect disentanglement. Notably, CD-RED explicitly encodes variations as rotations via a direct product of $SO(2)$ groups within orthogonal hyperspherical subspaces, providing a principled mechanism for precise equivariance. We theoretically and experimentally establish that CD-RED achieves perfectly disentangled representations, suggesting a promising new direction for self-supervised disentanglement.

## 1 Introduction

Learning disentangled representations, where independent factors of data variation are encoded into separate feature dimensions, is a fundamental goal in machine learning (Bengio et al., 2013; Hinton & Salakhutdinov, 2006; Higgins et al., 2018; Locatello et al., 2019; Weiler & Cesa, 2019). Such representations are critical for a wide range of purposes, including improving generalization, enhancing interpretability, data generation, and transfer learning (Locatello et al., 2019; Ren et al., 2021).

The pursuit of disentanglement has historically followed two main paradigms: supervised learning with labeled factors (Reed et al., 2014; Cheung et al., 2014), and unsupervised learning with tailored inductive biases designed to encourage statistical independence in the latent space (Higgins et al., 2017; Kim & Mnih, 2018; Chen et al., 2018). The former is limited by the cost of labels, while the latter often relies on strong assumptions that may not align with the real-world data (Locatello et al., 2019). This landscape has motivated a recent shift towards self-supervised learning, which seeks to discover factors directly from the structure of unlabeled data.

Conventional self-supervised learning methods (Tian et al., 2020; He et al., 2020; Wu et al., 2018; Chen et al., 2020a) extract robust features by enforcing invariance to data augmentations, typically by minimizing a contrastive loss such as InfoNCE (Oord et al., 2018). Despite their popularity and effectiveness for clustering, these methods often fail to achieve feature disentanglement in practice, since they lack precise control over how the learned representation structures features associated with data variations — merely requiring proximity. Although some analyses suggest InfoNCE can promote disentanglement under strong assumptions (Von Kügelgen et al., 2021; Zimmermann et al., 2021; Ngweta et al., 2023), the conditions rely on idealizations that are seldom met in practice.

To impose more structure, one line of work (Xiao et al., 2020; Wang et al., 2021; Eastwood et al., 2023) stacks a group of contrastive objectives, working on different partitions of the data or working with distinct sets of augmentations, to extract factors of data variation. While effective to some

degree, these methods exhibit high conceptual complexity and still typically rely on strong assumptions or independence regularizations to achieve disentanglement.

Another prominent line of work seeks to go beyond mere invariance by learning equivariant representations (Dangovski et al., 2021), where feature transformations predictably correspond to data transformations. While this provides a principled way to organize features towards disentanglement, existing equivariant methods face significant limitations. For instance, the approach by Shakerinava et al. (2022) is limited to single-cluster manifolds. Garrido et al. (2023); Marchetti et al. (2023) rely on modeling cluster and variation features in separate spaces.

Among these distinct approaches, our work follows Gupta et al. (2023): learning representations that capture discrete latent structures on a single hypersphere. Nevertheless, they require jointly optimizing the InfoNCE and equivariance losses. This coupling creates an inherent trade-off that prevents neat equivariance and, consequently, equivariance-based disentanglement. To address this issue, we propose a two-stage framework in this paper that confines the use of InfoNCE to the initialization stage, which allows the second stage to focus solely on equivariance learning. We thereby achieve neat equivariance, a fundamental prerequisite for equivariance-based disentanglement.

However, neat equivariance alone is insufficient for disentanglement, as the geometric structure of the equivariance is also critical Shakerinava et al. (2022). A common issue in existing methods is that they employ an overly arbitrary equivariance modeling, which often fails to impose a meaningful geometric transformation. For instance, Devillers & Lefort (2022) uses a learnable feed-forward network to predict feature changes, which cannot guarantee a consistent geometric structure. In contrast, Gupta et al. (2023) adopts a more structured approach by modeling the changes as an implicit rotation on the hypersphere. Although this approach has clear potential to structure the feature space, a significant limitation is its mandate of a global transformation that is identical for all inputs. This inherently forces the semantic clusters into a parallel arrangement, thereby reducing inter-cluster distances and, consequently, the representational efficiency on the hypersphere. To overcome this limitation, we propose in this paper Cluster-Dependent Rotational Equivariance, which resolves the issue by making the rotational transformation local to each cluster.

Another key challenge in learning neat equivariance lies in the imprecise nature of implicit modeling. To address this, we shift to an explicit model of rotational transformation, leveraging a composite rotation group formed by the direct product of $SO(2)$ subgroups (Quessard et al., 2020; Shakerinava et al., 2022) to cleanly encode data variations as rotations within a set of orthogonal 2D subspaces.

Integrating these insights, we arrive at Cluster-Dependent Rotational Equivariance for Disentanglement (CD-RED). Our theoretical and experimental analyses show that CD-RED achieves perfectly disentangled representations, constituting a significant advance in self-supervised learning.

Our primary contributions are summarized as follows:

- A novel two-stage learning framework that decouples invariance and equivariance learning, enabling learning neat equivariant representations on a single hypersphere.
- The introduction of Cluster-Dependent Rotational Equivariance that overcomes the inherent limitations of global rotational equivariance by making the transformation local to each cluster.
- CD-RED, a concrete method that provably achieves perfect disentangled representations.
- A theoretical extension of disentanglement theory, moving beyond the idealized assumptions of Higgins' formulation to handle more practical, complex augmentations.

## 2 PRELIMINARY AND RELATED WORK

We begin by formalizing the representation learning pipeline and clarifying how invariance, equivariance, and disentanglement arise within this context.

We adopt the standard generative formulation where data originates from a latent semantic space $\mathcal{Z} \in \mathbb{R}^m$ composed of independent semantic factors $(z_1, \ldots, z_m)$. Each $z \in \mathcal{Z}$ is rendered into the observation space $\mathcal{X}$ via a possibly unknown invertible and non-linear function $\varphi : \mathcal{Z} \mapsto \mathcal{X}$, and mapped into a feature space by a **feature extractor** $f : \mathcal{X} \mapsto \mathcal{Y}$, giving rise to the representation pipeline: $\Phi : \mathcal{Z} \xrightarrow{\varphi} \mathcal{X} \xrightarrow{f} \mathcal{Y}$. We assume $f(x)$ is $\ell_2$-normalized unless otherwise stated.

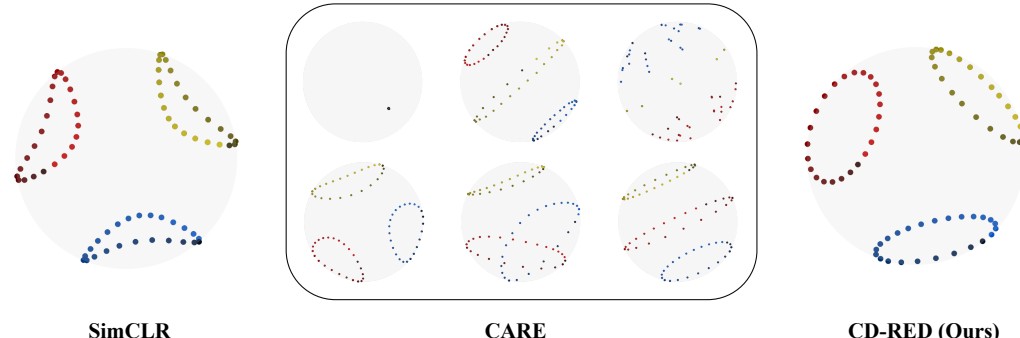

**SimCLR**                          **CARE**                          **CD-RED (Ours)**

Figure 1: An Illustration of the essential differences between closely related methods with results on a dataset of three rotating objects. **SimCLR** (Chen et al. (2020a), InfoNCE): without equivariance, the learned features are not well-structured, though forming uniformly distributed clusters. **CARE** (Gupta et al., 2023): jointly optimizing the InfoNCE and global rotational equivariance losses, the results are highly sensitive to the weighting of each component; under a good balance, the learned features form parallelly distributed clusters with varying radius. **CD-RED**: with cluster-dependent rotational equivariance, the learned features form uniformly distributed, well-structured clusters.

To enable training in the self-supervised setting, a set of $m$ data augmentations $\mathcal{A} = \{a_i\}_{i=1}^m$ is considered, where each $a_i$ denotes a transformation (e.g., rotation, cropping), possibly parameterized by a latent variable $t$ drawn from $T_{a_i}$. Each $a_i$ can be applied individually or in combination with others. For simplicity, we abuse $a \in \mathcal{A}$ as a single or composite augmentation, while retaining the ability to refer to internal parameters $t$ when needed.

Given an input $x = \varphi(z)$, we construct a positive sample $x^+ = a(x)$ by applying a sampled augmentation $a \in \mathcal{A}$, and aim to relate $f(x)$ and $f(x^+)$ through appropriate learning objectives.

A core setting is that each augmentation in $\mathcal{A}$ preserves some latent factors while perturbing others. We categorize the latent space $\mathcal{Z}$ into two subsets[1]:

- **Style Latents**: Latent factors that are altered by at least one augmentation in $\mathcal{A}$, which encodes changes such as views, colors, and positions.

- **Content Latents**: Latent factors that are preserved under all augmentations in $\mathcal{A}$. These define the semantic identity of the sample and are expected to remain unchanged under transformation.

This distinction underpins the goals of self-supervised learning: to capture content-related variations through invariance, while optionally modeling style-related variations through equivariance.

**Augmentations in Invariance** A foundational goal in self-supervised learning is to learn invariant representations, i.e., feature embeddings that remain stable under semantic-preserving augmentations. The objective can be characterized by $f(a(x)) \approx f(x), \forall a \in \mathcal{A}$, while an alternative formula can be $f(a_1((x)) \approx f(a_2(x)), \forall a_1, a_2 \in \mathcal{A}$, which is primarily the same. We would express it in terms of the former for simplicity. A widely-used objective for learning such invariant representations is the InfoNCE (Oord et al., 2018; Tian et al., 2020; He et al., 2020; Wu et al., 2018; Chen et al., 2020a), which is formulated as:

$$\mathcal{L}_{\text{Info}} = \frac{1}{2N} \sum_{i=1}^{2N} - \log \frac{\exp(\text{sim}(f(x_i), f(x_i^+))/\tau)}{\sum_{j=1, j\neq i}^{2N} \exp(\text{sim}(f(x_i), f(x_j))/\tau)}, \tag{1}$$

where the features $f(x)$ are confined to the hypersphere $\mathcal{Y} = \mathbb{S}^{d-1} \triangleq \{z \in \mathbb{R}^d : \|z\|_2 = 1\}$, $\text{sim}(a, b) = a^\top b$ denotes the cosine similarity between two unit vectors, and $\tau > 0$ is the temperature parameter. This loss encourages features of augmented views of the same semantic instance to cluster, while driving features of different instances to be uniformly distributed on the hypersphere (Parulekar et al., 2023; Wang et al., 2022; Wang & Isola, 2020).

---

[1]Note that the assignment of latents as content or style is contingent upon the chosen augmentations.

**Augmentations in Equivariance** In contrast to invariance, equivariance structures the feature space by requiring that transformations in the input space induce predictable changes in the feature space, which can be formulated as $f(a(x)) \approx M_a(f(x))$, where $M_a$ is a feature-space transformation aligned with $a$. The corresponding objective can be:

$$\mathcal{L}_{\text{equi}_0} = \mathbb{E}_{x,a} \left[ \|f(a(x)) - M_a(f(x))\|^2 \right]. \quad (2)$$

Optimizing this objective alone may lead to collapsed features, and this objective is therefore usually optimized jointly with a collapse-avoiding loss (e.g., InfoNCE). Existing methods differ in their modeling of $M_a$. One class of methods learns $M_a$ as a neural network conditioned on both $f(x)$ and $a$, i.e., $M_a(f(x)) = \text{MLP}(f(x), a)$ (Xiao et al., 2020; Dangovski et al., 2021; Devillers & Lefort, 2022). Predicting change with an arbitrarily expressive non-linear neural network can enable the model to learn richer and more informative features; however, it may contribute little to structuring the feature space. In contrast, more recent methods impose stronger geometric structures. For instance, Garrido et al. (2023) models the transformation more structurally as parametrized linear mappings, while (Shakerinava et al., 2022; Gupta et al., 2023) implicitly constrain $M_a$ to be, e.g., an orthogonal matrix, giving the loss as $\mathcal{L}_{\text{equi}_1} = \mathbb{E}_{a \in \mathcal{A}} \mathbb{E}_{x,x' \in \mathcal{X}} [ f(a(x'))^\top f(a(x)) - f(x)^\top f(x') ]^2$. In fact, for features constrained to a hypersphere, these two approaches are nearly equivalent, since any valid linear transformation operating on a hypersphere reduces to an orthogonal matrix. Notably, they enforce such orthogonal transformation globally across all samples on the hypersphere, which drives features of distinct semantic clusters to be parallelly distributed on different hyperplanes, conflicting with the commonly coupled InfoNCE objective that promotes uniformly distributed clusters across the hypersphere and, more critically, inherently leading to reducing inter-cluster distances.

**Augmentations in disentanglement** The objective of disentangled representation learning is to induce a feature mapping $f$ such that independent latent factors of variation are encoded into distinct, interpretable components within the learned representation $f(x)$ (Bengio et al., 2013). A foundational framework was put forward by Higgins et al. (2018), which defines a representation as disentangled if variations in a single generative factor correspond to changes in a single latent unit, with all other units remaining unaffected. Building on this conceptual foundation, recent self-supervised methods have aimed to approximate disentanglement through group-theoretic formulations (Wang et al., 2021; Marchetti et al., 2023; Shakerinava et al., 2022). In these approaches, data augmentations are assumed to form a decomposable group $\mathcal{G} = \mathcal{G}_1 \times \cdots \times \mathcal{G}_n$, with each subgroup $\mathcal{G}_i$ acting on a single generative factor and corresponding to a specific subspace $\mathcal{Y}_i$ in the feature space. While elegant in theory, this framework hinges on strong assumptions about the augmentations that each affects a single generative factor. In practice, however, many augmentations simultaneously affect multiple latent factors. In contrast to prior work, we also model augmentations that simultaneously impact multiple latent factors, enabling disentanglement under such entangled augmentations.

## 3 CLUSTER-DEPENDENT ROTATIONAL EQUIVARIANCE

We introduce in this section the proposed framework, Cluster-Dependent Rotational Equivariant for self-supervised Disentangled representation learning (CD-RED).

Our first insight regarding cluster-dependent rotational equivariance is that while equivariance is essential, its joint optimization with a collapse-preventing objective like InfoNCE inherently interferes with learning neat equivariance. However, the core purpose of the collapse-preventing objective is solely to avoid feature collapse, which can be better achieved with a two-stage method to avoid interference: a primary stage using self-supervised clustering (e.g., InfoNCE) to establish non-collapsed semantic clustering, followed by a secondary stage that learns neat equivariance within each cluster, where ensuring non-trivial equivariance suffices to prevent intra-cluster collapse.

Our second insight regarding cluster-dependent rotational equivariance is that enforcing global rotational equivariance (e.g., that employed by CARE, Gupta et al. (2023)) for features on the hypersphere would drive distinct semantic clusters to be parallelly distributed on different hyperplanes. This inherently results in reduced inter-cluster distances, blurs the boundaries between clusters, and ultimately limits the framework to supporting the joint modeling of only a relatively small number of clusters. This limitation can be naturally addressed by shifting to a cluster-dependent rotational system, i.e., making the transformation local to each cluster.

We summarize our method in Algorithm 1, outlining the overall objective and the training steps.

## 3.1 STAGE 1: SELF-SUPERVISED INVARIANCE LEARNING FOR CLUSTERING

In the first stage, we perform conventional self-supervised invariance learning by the InfoNCE loss to obtain well-structured representations. Theoretically, at its optimum, InfoNCE learns semantically meaningful clusters that are uniformly distributed across the hypersphere (Wang & Isola, 2020).

This initial stage yields features that are amenable to clustering. We then apply spherical k-means (Hornik et al., 2012) or agglomerative clustering (Müllner, 2011) to these features to obtain the initial cluster centroids and assignments.

## 3.2 STAGE 2: SELF-SUPERVISED EQUIVARIANCE LEARNING FOR DISENTANGLEMENT

In the second stage, we learn cluster-dependent rotational equivariance within each cluster.

### 3.2.1 CLUSTER-DEPENDENT ROTATION SYSTEMS

Cluster-dependent rotational equivariance necessitates a local rotation system for each cluster. This requires two key components: a rotation center and a rotation group.

**Rotation Center**. For the rotation center, we initialize it as the cluster centroid from the first stage, which serves as the anchor point for all rotations within the cluster. Concretely, each cluster $i$ is assigned a rotation center $r_i$, defined as the normalized centroid of its features: $r_i = c_i$. Since InfoNCE is proven to induce uniformly distributed clusters, these initial centroids already provide stable rotation anchors and can thus be fixed during subsequent training. Nevertheless, we may optionally introduce the loss in Eq. (31 to further promote its uniform distribution.

**Rotation Group**. Existing approaches often model equivariant transformations as generalized linear transformations. When applied to features constrained to hypersphere surfaces, they reduce to orthogonal matrices (Gupta et al., 2023). This, however, introduces two main issues.

- *Reflection ambiguity:* Orthogonal matrices may include reflections ($\det = -1$), which can flip semantic orientations. While the model may not learn reflections if they do not fit the data, this remains a potential drawback.
- *Unnecessary complexity:* Full $d \times d$ rotation involves many feature dimensions, making it unnecessarily complex for modeling simple augmentations that typically affect low-dimensional subspaces. Such complexity not only increases the learning difficulty but also hampers subsequent disentanglement.

To address these issues, we restrict each transformation to a 2D subspace and model it directly via an element of the special orthogonal group $SO(2)$, defined as

$$SO(2) \triangleq \left\{ R \in \mathbb{R}^{2 \times 2} \,\middle|\, R^\top R = I, \, \det(R) = 1 \right\}. \tag{3}$$

Each element of $SO(2)$ can be parameterized by a single angle $\theta \in [-\pi, \pi]$ via the mapping

$$R(\theta) = \begin{bmatrix} \cos\theta & -\sin\theta \\ \sin\theta & \cos\theta \end{bmatrix}. \tag{4}$$

We assign each orthogonal augmentation[2] to a dedicated 2D hyperspherical subspace within the full feature space $\mathbb{S}^{d-1}$ and restrict the corresponding feature rotations to that subspace. More formally, for the $i$-th augmentation, we assign it to operate on the $(2i-1)$-th and $(2i)$-th dimensions without loss of generality: each orthogonal augmentation must occupy at least two dimensions, but it does not matter which ones. Instead of letting the model learn to select these dimensions, we specify them directly, simplifying the process and avoiding unnecessary complexity. Specifically, for $n$ orthogonal augmentations, we define the composite rotation as a block-diagonal matrix:

$$\mathcal{R}(\theta) = \mathrm{diag}(R(\theta^{(1)}), R(\theta^{(2)}), \ldots, R(\theta^{(n)}), I_{d-2n}) \in \mathbb{R}^{d \times d}, \tag{5}$$

which subsequently acts on the full feature vector.

Each $\theta_i$ is predicted by a small **neural network** $g_i$ dedicated to the $i$-th augmentation. The input to $g_i$ can include the augmentation parameter $t$ (e.g., rotation angle) and, optionally, input-dependent

---

[2]Augmentations that cause variations in a single style factor that are distinct from each other.

signals (for input-dependent augmentations). The output is constrained to $[-\pi, \pi]$ via a tanh activation to: (i) avoid degenerate $2\pi$-periodic equivariance, and (ii) enable inverse operations (e.g., left vs. right shift) by supporting both positive and negative values.

This construction not only avoids reflections and unnecessary complexity but also provides a clear, modular way to encode independent variation directions. It naturally generalizes to more complex cases: for augmentation that may affect multiple latent factors, we assign it to the multiple orthogonal subspaces and learn corresponding rotation parameters (see Section 4.3).

**Coordinate Alignment**. The block-diagonal rotation matrix $\mathcal{R}(\theta)$ introduced earlier operates in a fixed coordinate system. To accommodate cluster-dependent centers, we align the rotation center of each cluster to a canonical reference direction, which is chosen as the north pole $e_d = (0, 0, \ldots, 1) \in \mathbb{S}^{d-1}$, before applying the rotation. This ensures that all rotations act on aligned coordinates while preserving cluster-local dynamics.

We achieve this alignment using the *Householder transformation* (Householder, 1958), a reflection over a hyperplane orthogonal to a given vector $u$:

$$H = I - 2\frac{uu^\top}{u^\top u}, \tag{6}$$

where $u$ is the normal vector of the reflecting hyperplane. In particular, a vector $v \in \mathbb{S}^{d-1}$ can be reflected to another vector $w \in \mathbb{S}^{d-1}$, i.e., $Hv = w$, by setting let $u = v - w$. To reflect the rotation center $r_i$ of cluster $i$ to $e_d$, we set $u_i = r_i - e_d$.

### 3.2.2 ENFORCING EQUIVARIANCE WITH ALIGNED ROTATIONS

With the constructed rotation systems, our equivariance condition can be expressed as:

$$H_{c(x)}f(a(t, x)) = \mathcal{R}(\theta_{a(t)})H_{c(x)}f(x), \ \forall x \in \mathcal{X}, \ a(t) \in \mathcal{A}, \tag{7}$$

where $c(x) := \arg\max_i \ f(x)^\top r_i$ denotes the dynamically reassigned cluster of $x$ determined by its proximity to the rotation centers, $H_{c(x)}$ is the corresponding Householder matrix, and $\mathcal{R}(\theta_{a(t)})$ is the block-rotation matrix for augmentation $a(t)$, defined as

$$\mathcal{R}(\theta_{a(t)}) := \mathrm{diag}\big(R(\theta_{a(t)}^{(1)}), \ldots, R(\theta_{a(t)}^{(m)}), I_{d-2m}\big), \tag{8}$$

with $\theta_{a(t)}^{(i)}$ being the rotation angle predicted by the corresponding $g$ network for augmentation $a$ in the $i$-th subspace. For composite augmentation, $\theta_{a(t)}^{(i)}$ is the accumulated rotation angle over the constituent single augmentations. We decompose the coordinate-aligned feature $H_{c(x)}f(x)$ for subsequent use:

$$H_{c(x)}f(x) = \big([H_{c(x)}f(x)]_1, \ldots, [H_{c(x)}f(x)]_m, \ [H_{c(x)}f(x)]_{\mathrm{extra}}\big), \tag{9}$$

where each subspace block $[\cdot]_i \in \mathbb{R}^2$ and $[\cdot]_{\mathrm{extra}} \in \mathbb{R}^{d-2m}$.

Equivariance is prone to trivial solutions. For instance, $\mathcal{R}(\theta_{a(t)})$ may collapse to the identity, yielding $f(a(x)) = f(x)$ and thus reducing to invariance rather than true equivariance. Existing remedies, such as combining with an InfoNCE loss (or its uniformity component) (Gupta et al., 2023; Devillers & Lefort, 2022), applying variance regularization (Garrido et al., 2023), or explicitly pushing features apart (van der Pol et al., 2020; Kipf et al., 2020), help mitigate such trivialities, but can inadvertently impair learning of neat equivariance due to their inferior compatibility with equivariance.

Our approach tackles this by directly avoiding the trivial solutions of equivariance:

- **Non-Identity Rotation**. We prevent the trivial case where $\mathcal{R}(\theta_{a(t)})$ becomes the identity by enforcing that the rotation angle $\theta_{a(t)}$ is non-zero. The $2\pi$ cyclic case is excluded by design, since $\theta_a$ is constrained to $[-\pi, \pi]$. For parameterized augmentations, we introduce a user-defined proxy $\tilde{q}_{a(t)} \in \mathbb{R}^m$, serving as a lower bound on the style variation magnitude, and impose:

$$\mathcal{L}_{\mathrm{theta}} = \mathbb{E}_{a(t) \in \mathcal{A}} \ \max(0, \ \epsilon * \tilde{q}_{a(t)} - |\theta_{a(t)}|), \tag{10}$$

  where $\theta_{a(t)} \in \mathbb{R}^m$ is the per-plane induced angles by $a(t)$ and $\epsilon > 0$ is a small margin. Note that for subspaces that are unaffected by $a(t)$, $|\theta_{a(t)}|$ is constantly zero by definition.

Table 1: DCI metrics comparison using Random Forest regression on four distinct datasets, shown as triples D/C/I for Disentanglement/Completeness/Informativeness, respectively.

| Method/Dataset | MPI3D (D/C/I) | Shape3D (D/C/I) | 3DIdent (D/C/I) | 3DIEBench (D/C/I) |
|---|---|---|---|---|
| *(Self-Supervised Invariance)* | | | | |
| SimCLR (Chen et al., 2020a) | 0.085 / 0.086 / 0.876 | 0.074 / 0.071 / 0.885 | 0.129 / 0.096 / 0.926 | 0.102 / 0.095 / 0.904 |
| *(Self-Supervised Equivariance)* | | | | |
| EquiMOD (Devillers & Lefort, 2022) | 0.143 / 0.127 / 0.959 | 0.125 / 0.118 / 0.968 | 0.067 / 0.024 / 0.933 | 0.091 / 0.088 / 0.911 |
| CARE(Gupta et al., 2023) | 0.084 / 0.085 / 0.891 | 0.078 / 0.0770 / 0.925 | 0.147 / 0.119 / 0.928 | 0.116 / 0.106 / 0.942 |
| *(Self-Supervised Disentanglement)* | | | | |
| IP-IRM (Wang et al., 2021) | 0.220 / 0.199 / 0.955 | 0.135 / 0.110 / 0.952 | 0.147 / 0.094 / 0.933 | 0.109 / 0.107 / 0.929 |
| Eastwood et al. (2023) | 0.950 / 0.826 / 1.000 | 0.937 / 0.766 / 1.000 | 0.943 / 0.772 / 0.996 | 0.362 / 0.297 / 0.952 |
| CD-RED (before post-proc.) | 0.628 / 0.505 / 1.000 | 0.630 / 0.494 / 1.000 | **0.989** / 0.940 / 1.000 | 0.595 / 0.481 / 1.000 |
| CD-RED (after post-proc.) | **1.000 / 1.000 / 1.000** | **1.000 / 1.000 / 1.000** | 0.987 / **0.987** / 1.000 | **0.996 / 0.996 / 1.000** |

- **Non-Zero Subspace Norm**. Since rotating a zero vector yields zero, we encourage each 2D plane to maintain a non-trivial radius close to a target $\omega > 0$:

$$\mathcal{L}_{\text{radius}} \;=\; \mathbb{E}_{x \in \mathcal{X}} \left[ \sum_{i=1}^{m} \left( \left\| \frac{[H_{c(x)} f(x)]_i}{\omega} \right\| - 1 \right)^2 \right].$$

We require $m\,\omega^2 < 1$ (with smaller $\omega$ preferred) to leave sufficient norm for the non-rotational complement ($[H_{c(x)} f(x)]_{\text{extra}}$) to enhance cluster separation.

With the target radius established, we arrive at our equivariance objective. We divide the features by $\omega$ as for $\mathcal{L}_{\text{radius}}$ to eliminate sensitivity to the hyperparameter $\omega$:

$$\mathcal{L}_{\text{equi}} = \mathbb{E}_{x \in \mathcal{X}, a(t) \in \mathcal{A}} \left[ \left\| \frac{H_{c(x)} f(a(t,x))}{\omega} - \frac{\mathcal{R}(\theta_{a(t)}) H_{c(x)} f(x)}{\omega} \right\|^2 \right]. \tag{11}$$

Our final training objective for CD-RED is thus given by:

$$\mathcal{L}(f, \{g_i\}_i) = \mathcal{L}_{\text{equi}} \;+\; \lambda_{\text{radius}} \, \mathcal{L}_{\text{radius}} \;+\; \lambda_{\text{theta}} \, \mathcal{L}_{\text{theta}}. \tag{12}$$

## 4 THEORETICAL AND EMPIRICAL RESULTS

In this section, we present theoretical results and empirical validation of the proposed CD-RED from Section 3. First, we show how our framework achieves neat equivariance and subsequent disentanglement, following the definition of Higgins et al. (2018), by leveraging transformations that induce orthogonal variations. Second, we extend the analysis to more complex transformations that may influence multiple latent factors. Formal statements and proofs are provided in Appendix C.

### 4.1 NEAT EQUIVARIANCE

**Proposition 1** (Perfect Equivariance). *CD-RED is capable of achieving perfect equivariance, in the sense that all losses ($\mathcal{L}_{\text{equi}}, \mathcal{L}_{\text{theta}}, \mathcal{L}_{\text{radius}}$) can simultaneously reach their optimum, where each augmentation $a(t)$ that induces a non-trivial variation in the input will correspondingly produce a non-trivial and unique transformation of the feature.*

The idea behind it is simple: by design, all these losses operate on separate aspects of the features and do not contradict each other, unlike most existing solutions (Devillers & Lefort, 2022; Garrido et al., 2023; Gupta et al., 2023).

### 4.2 AUGMENTATIONS INDUCE ORTHOGONAL VARIATIONS

We now show that achieving perfect equivariance, as defined above, already implies a basic form of disentanglement when augmentations act on distinct subspaces.

**Proposition 2** (Weak Disentanglement). *If perfect equivariance is achieved and under mild conditions, for any rotation subspace $i$ assigned exclusively to an orthogonal augmentation $a_i(t)$, the learned angle is linear in the induced latent displacement $q_{a_i}(t)$:*

$$\theta_{a_i(t)} \;=\; g_i(t) \;=\; c_0^{(i)}\, q_{a_i}(t), \quad c_0^{(i)} \neq 0,$$

*and only the $i$-th plane changes under $a_i(t)$:*

$$\big[H_{c(x)} f(a(t,x))\big]_i \;=\; \mathcal{R}(\theta_{a_i(t)}^{(i)}) \big[H_{c(x)} f(x)\big]_i, \quad \big[H_{c(x)} f(a(t,x))\big]_j \;=\; \big[H_{c(x)} f(x)\big]_j \quad \forall j \neq i.$$

That is, when the augmentations are orthogonal, the rotation parameters in their exclusively assigned subspaces linearly correspond to the latent displacement induced by the augmentations.

A limitation of this weak form is that the feature angle in a subspace may not monotonically track the variation $q_{a(t)}$ due to $2\pi$ wrap-around (Figure 2). To prevent ambiguous crossings, we constrain $\theta_{a(t)}$ so that even the corresponding rotation angle for the largest rotations in the data remains within $[-\pi, \pi]$, avoiding wrap-around:

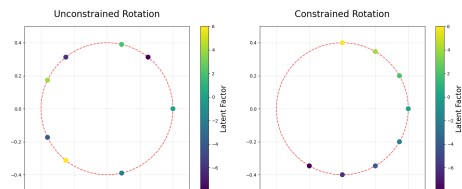

$$\theta_{a(t)} \;\in\; \eta \cdot \frac{\max_{t \in T_a} \tilde{q}_{a(t)}}{\max_{t \in T_{\text{data}}} \tilde{q}_a(t)} \cdot [-\pi, \pi], \quad (13)$$

where $\eta \in (0, 1]$ is a usage threshold, e.g., $\eta = 0.5$ for non-cyclic latents and $\eta = 1$ for cyclic. We use $\max_{t \in T_a} \tilde{q}_{a(t)}$ and $\max_{t \in T_{\text{data}}} \tilde{q}_{a(t)}$ to denote the approximate maximum variation strength present in the augmentation and data, respectively.

Figure 2: **Left:** Without constraining $\theta_a$, feature subspaces may overlap, causing crossed usage. **Right:** Constraining the range of $\theta_a$ prevents overlap, ensuring a monotonic relationship between the variation and the learned feature.

**Proposition 3** (Strong Disentanglement). *If $\theta_{a(t)}$ is constrained in range to avoid wrap-around as in Eq. (13), then the subspace features associated with $a(t)$ vary **monotonically and linearly** with the underlying style variation $q_{a(t)}$ in the data, achieving strong disentanglement.*

With $\theta_{a(t)}$ effectively constrained, we can further derive a shared, interpretable coordinate through post-processing. For any data $x = \varphi(z)$ and subspace $j$ write

$$[H_{c(x)} f(x)]_j := \big(u_j(x), v_j(x)\big), \quad \theta^{(j)}(x) := \text{atan2}\big(v_j(x), u_j(x)\big) \in (-\pi, \pi].$$

To obtain an interpretable readout, we align the angles by an offset $\alpha_{c(x),j}$ specific for cluster $c(x)$ and plane $j$ and compute the signed principal difference:

$$\widehat{f}^{(j)}(x) := \text{atan2}\big(\sin(\theta^{(j)}(x) - \alpha_{c(x),j}), \cos(\theta^{(j)}(x) - \alpha_{c(x),j})\big) \in (-\pi, \pi].$$

- **Noncyclic latents.** When $z^{(j)}$ within a cluster is noncyclic, choose $\alpha_{c(x),j}$ as the midpoint of the covered data arc, then Corollary 2 provides an exact affine readout of $\widehat{f}^{(j)}(x)$ relative to latent $z^{(j)}$, and $\widehat{f}^{(j)}$ is globally aligned across clusters whenever they share the same latent midpoint.
- **Cyclic latents.** For intrinsically cyclic factors, no lossless 1-D scalarization exists; we therefore retain the 2-D coordinates $[H_{c(x)} f(x)]_j$ on $\mathbb{S}^1$. By Corollary 3, each cluster $c_k$ differs only by a fixed in-plane rotation $A_{j,k} \in SO(2)$. Global comparison across clusters is achieved via a simple cluster-wise phase alignment, e.g., rotating each cluster by $A_{j,k}^{-1}$ toward a chosen reference.
- **Content latents.** Finally, appending a one-hot cluster indicator to the post-processed style coordinates produces a feature vector disentangled across *style latents* (per subspace as described above) and *content latents* (cluster identity).

**Empirical results.** We evaluate on **MPI3D** (Gondal et al., 2019) and **Shape3D** (Kim & Mnih, 2018) (primarily discrete latents), as well as on **3DIdent** (Zimmermann et al., 2021) and **3DIEBench** (Garrido et al., 2023) (continuous latents). Latent transformations are used as augmentations, with the transformation parameter $t = q_{a(t)}$. We adopt **DCI** (Eastwood & Williams, 2018) as our evaluation metric, which quantifies disentanglement, completeness, and informativeness via supervised regressors mapping features to ground-truth latents. Each component is in the range $[0, 1]$. Larger is

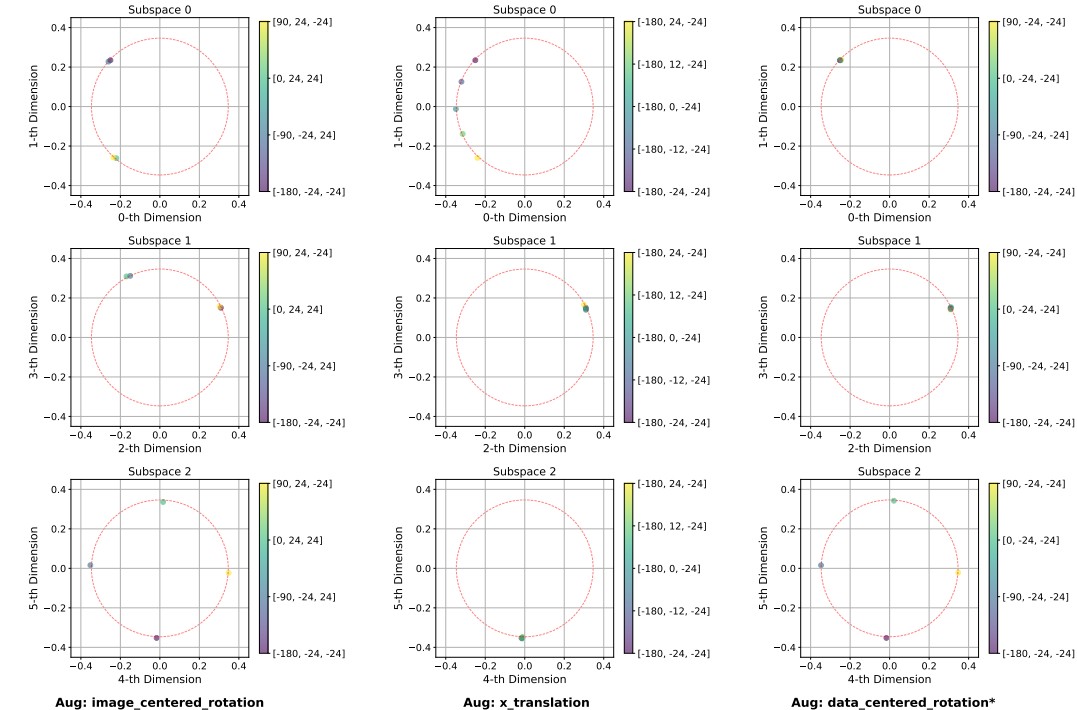

Figure 3: Experiment with x-translation, y-translation, and image-centered rotations. We visualize the learned features in the three subspaces. The image-centered rotations are perfectly disentangled into x-translation, y-translation, and data-centered rotation. The bar indicates the latents of samples.

better. Table 1 compares CD-RED with SimCLR (Chen et al., 2020a), recent self-supervised equivariant baselines (Devillers & Lefort, 2022; Gupta et al., 2023), and self-supervised disentanglement baselines (Wang et al., 2021; Eastwood et al., 2023). Across all four datasets, encompassing both discrete and continuous latent settings, CD-RED achieves near-perfect DCI scores and significantly outperforms the baselines. Details of datasets, metrics, and protocols are provided in Appendix D.

### 4.3 AUGMENTATIONS INDUCE COMPLEX VARIATIONS

Real-world transformations introduce complex variations that involve multiple underlying factors (e.g., rotating a non-centric object around the origin also induces coupled translations). We now formalize how our framework disentangles such complex augmentations.

**Proposition 4** (Disentanglement of Simple Composite Augmentations). *Given $m$ orthogonal augmentations, each assigned exclusively to a unique subspace, if perfect equivariance is achieved, then for any composite augmentation $a_{comp}$ that induces only $n\ (\leq m)$ of these style variations, the framework will learn to disentangle them into the corresponding $n$ orthogonal subspaces, with each variation captured by its associated rotation angle $\theta$.*

This is because the $m$ orthogonal augmentations define how each input variation corresponds to the rotation angles $\theta$ in their respective subspaces. Any additional augmentation that overlaps with these subspaces must follow these predefined mappings.

**Proposition 5** (Disentanglement of Complex Augmentation with Anchor). *Given $m-1$ orthogonal augmentations, each assigned to a unique rotation plane, let $a_{\mathrm{cmp}}$ be an additional augmentation such that there exists an anchor where $a_{\mathrm{cmp}}$ acts purely on the $m$-th style latent, and away from the anchor, $a_{\mathrm{cmp}}$ decomposes into a composite over $n\ (\leq m)$ already defined subspaces. Then, assigning a new dedicated plane $m$ to this anchor yields a linear readout $\theta^{(m)}_{a_{\mathrm{cmp}}(t)} = c^{(m)}_0 q_{\mathrm{cmp}}(t)$, and subsequently the angle readout in all affected $n\ (\leq m)$ subspaces is additive across planes, as described in Proposition 4.*

This is because the existence of an anchor defines the $m$-th subspace, and the remaining then directly follows from Proposition 4.

**Empirical results.** We further provide empirical validation for Proposition 5 in Figure 3. A similar validation of Proposition 4 is provided in Figure 13 in the Appendix F.2.

## 5 ADDITIONAL RELATED WORK

**Self-supervised Clustering.** Self-supervised clustering methods are commonly built on either contrastive or non-contrastive objectives. Most contrastive methods (Yeh et al., 2022; He et al., 2020; Chen et al., 2020b;a) use an InfoNCE-type loss (Oord et al., 2018). HaoChen et al. (2021) studies a spectral contrastive loss, and Caron et al. (2018; 2020); Li et al. (2020) treat cluster centroids (prototypes) as contrastive views instead of individual samples. Non-contrastive methods remove the need for explicit negative pairs, either by architectural design (Grill et al., 2020; Chen & He, 2021) or by regularizing the covariance of the embeddings (Ermolov et al., 2021; Zbontar et al., 2021; Bardes et al., 2021; 2022). Although there are works showing that contrastive and non-contrastive methods share theoretical connections (Garrido et al., 2022), Pokle et al. (2022) argue that non-contrastive methods may fail to learn cluster-aligned representations. Our first stage of training builds upon the SimCLR framework (Chen et al., 2020a), for which recent analyses provide theoretical guarantees that the learned features form well-separated clusters in the hypersphere (Wang & Isola, 2020; Wang et al., 2022; Parulekar et al., 2023).

**Group-equivariance Representation Learning.** Group-equivariance representation learning aims to learn feature maps whose response to structured transformations of the input follows a group representation, i.e., $\varphi(T_g(x)) = \rho(g)\varphi(x)$ for a group $g \in G$. Classical group-equivariance CNNs instantiate this idea by fixing both the group $G$ and its action on the input domain, then designing convolutional layers so that feature channels transform according to a prescribed representation $\rho$ (Cohen & Welling, 2016a;b; Cohen et al., 2018; Weiler et al., 2018; Weiler & Cesa, 2019; Finzi et al., 2020; Lengyel et al., 2023; Yang et al., 2024). More recently, the architecture extends to message-passing GNN (Anderson et al., 2019; Satorras et al., 2021) and Transformer (Tai et al., 2019; Fuchs et al., 2020). However, in these approaches, the group and its feature-space representation are fixed a priori and hard-coded into the architecture. In contrast, our method is more flexible by keeping the backbone architecture generic and instead learning equivariance in feature space.

**Disentanglement in Generative Models.** Separating latent factors of variation is one of the central objectives in generative modeling, and has been explored in VAE-based approaches (Higgins et al., 2017; Kim & Mnih, 2018; Chen et al., 2018; Li et al., 2019; Mathieu et al., 2019; Esmaeili et al., 2019), GAN-based methods (Chen et al., 2016; Jeon et al., 2021; Ojha et al., 2020), and more recent denoising-based generative models (Yang et al., 2023; Song et al., 2023; Wu & Zheng, 2024; Hu et al., 2024). These methods aim to map data to structured latent spaces, but in practice often struggle to balance sample quality, degree of disentanglement, and robustness to complex data distributions. There also exists a line of work that promotes disentanglement through weak supervision in the form of pairwise or grouped observations, which can be viewed as imposing conditional invariance constraints on subsets of the latent variables (Bouchacourt et al., 2018; Hosoya, 2018; Shu et al., 2019; Chen & Batmanghelich, 2020; Locatello et al., 2020). In contrast, our method is purely self-supervised while achieving cleaner and stronger disentanglement.

## 6 CONCLUSION

In this work, we have proposed CD-RED, a two-stage self-supervised disentanglement framework that learns initial semantic clustering via InfoNCE in the first stage and then learns cluster-dependent rotational equivariance in the second stage. CD-RED aligns rotation coordinates of different clusters via Householder transformations and models equivariance as $SO(2)$ groups within orthogonal hyperspherical subspaces. We have theoretically established that, under mild assumptions, the resulting representations achieve neat equivariance and perfect disentanglement of cluster and style latents. Our results go beyond standard group-disentanglement settings, providing disentanglement under more practical, non-orthogonal augmentations. Extensive empirical results have demonstrated that CD-RED can robustly achieve near-perfect disentanglement across various settings.

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

## A USE OF LARGE LANGUAGE MODELS

We used a large language model (LLM) only for language polishing (grammar, wording, and clarity) of drafts written by the authors. The model did not generate research ideas, methods, analyses, results, or figures, and it did not write any sections from scratch.

## B EXTENSIONS AND SETTINGS OF FIGURE 1

**Settings.** We construct a toy dataset of three uppercase letters ("R", "E", "D") rendered as $96 \times 96$ grayscale images. Each instance is generated by rotating the glyph over angles $\{-165°, -150°, \ldots, 180°\}$ (step $15°$), forming a full cyclic orbit. During training, we use only a $15°$ rotation augmentation (bilinear interpolation). All models use temperature $\tau = 0.5$.

We follow Wang & Isola (2020) to decompose the contrastive objective InfoNCE into *invariance* and *uniformity* terms. For CARE (Gupta et al., 2023), we sweep the weights on these three components (invariance/uniformity/equivariance) to produce the results in Figure 1.

Empirically, the three terms *compete*: stronger equivariance encourages features to align along rotation orbits, whereas strong invariance/uniformity tends to collapse or spread them irrespective of orbit structure. The best equivariance is achieved when cluster layouts are aligned with the rotational orbits (i.e., parallel).

**Extension: non-cyclic setting.** We repeat the visualization on a non-cyclic range by restricting angles to $[-120°, 120°]$ (same dataset, step $15°$).

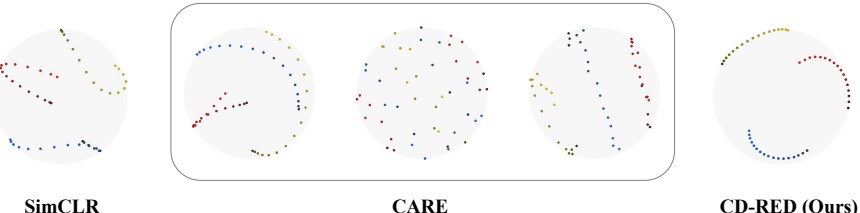

**SimCLR**  **CARE**  **CD-RED (Ours)**

Figure 4: Comparisons for non-cyclic setting.

## C PROOFS AND REMARKS

### C.1 PRELIMINARY

**Definition 1** (Augmentation Connectivity). *Let $\mathcal{A}$ be a set of augmentations acting on the observation space $\mathcal{X}$. Define a relation $\sim_{\mathcal{A}}$ on $\mathcal{X}$ by*

$$x \sim_{\mathcal{A}} y \iff \exists a, b \in \mathcal{A} : \varphi^{-1}(a(x)) = \varphi^{-1}(b(y)).$$

*The relation $\sim_{\mathcal{A}}$ is an* equivalence relation*:*

1. ***Reflexive.*** $x \sim_{\mathcal{A}} x$;

2. ***Symmetric.*** *If $x \sim_{\mathcal{A}} y$, then $y \sim_{\mathcal{A}} x$;*

3. ***Transitive.*** *If $x \sim_{\mathcal{A}} y$, $y \sim_{\mathcal{A}} z$, then $x \sim_{\mathcal{A}} z$.*

*A subset $D \subseteq \mathcal{X}$ is said to be $\mathcal{A}$-connected if $x \sim_{\mathcal{A}} y$ for every pair $x, y \in D$.*

**Assumption 1** (Intra-cluster Connectivity). *Fix a collection of augmentations $\mathcal{A}$ acting on the observation space $\mathcal{X}$ and let $\sim_{\mathcal{A}}$ denote the augmentation-connectivity relation defined in Definition 1. Let $\mathcal{X}_{\text{train}} \subset \mathcal{X}$ be the training set, and write*

$$\mathcal{Z}_{\text{train}} := \varphi^{-1}(\mathcal{X}_{\text{train}}) \subseteq U_{\mathcal{A}} \times S_{\mathcal{A}},$$

*where $U_{\mathcal{A}}$ is the cluster subspace and $S_{\mathcal{A}}$ is the style subspace.*

1. **Augmentation-wise factorization.** *Each $a \in \mathcal{A}$ lifts to a map*

$$\tilde{a} : \mathcal{S}_{\mathcal{A}} \times \mathcal{U}_{\mathcal{A}} \longrightarrow \mathcal{S}_{\mathcal{A}} \times \mathcal{U}_{\mathcal{A}}, \qquad \tilde{a}(s, u) = (\tilde{s}, u),$$

*where $\tilde{s} \neq s$ for non-identity transformations. Thus, every augmentation acts only on the style coordinates $s$ and leaves the cluster coordinates $u$ unchanged.*

2. **Intra-cluster $\mathcal{A}$-connectivity.** *Define the cluster partitions*

$$\mathcal{C}_{\mathcal{A}} := \big\{ c_u := \big\{ x \in \mathcal{X}_{\text{train}} \ : \ \varphi^{-1}(x) = (s, u), \ s \in \mathcal{S}_{\mathcal{A}} \big\}, \forall u \in \mathcal{U}_{\mathcal{A}} \big\}.$$

*We assume that each $c_u$ is $\mathcal{A}$-connected: for all $x, y \in c_u$ one has $x \sim_{\mathcal{A}} y$.*

*Because augmentations never modify the $u$-coordinate, no sequence of augmentations can connect points with different cluster coordinates, i.e., if $x \in c_u$ and $y \in c_{u'}$ with $u \neq u'$ then $x \not\sim_{\mathcal{A}} y$.*

**Definition 2** (Cluster-wise axis-aligned representation). *Let $f : \mathcal{X} \to \mathbb{S}^{d-1}$ be a unit-norm representation. Fix an augmentation set $\mathcal{A}$ that alters exactly $d_{\text{sty}}$ style factors and induces cluster partitions $\mathcal{C}_{\mathcal{A}}$ of $\mathcal{X}_{\text{train}}$. We say $f$ is **cluster-wise axis-aligned** if there exist orthogonal matrices $\{H_c \in O(n)\}_{c \in \mathcal{C}_{\mathcal{A}}}$ (one per cluster) such that, for the aligned representation*

$$\tilde{f}(x) := H_{c(x)} f(x) \ = \ \big( \tilde{f}_{\text{sty}}^{(1)}(x), \ldots, \tilde{f}_{\text{sty}}^{(d_{\text{sty}})}(x), \tilde{f}_{d - 2 d_{\text{sty}} - 1}(x) \big) \in \big( \mathbb{S}^1 \big)^{d_{\text{sty}}} \times \mathbb{S}^{d - 2 d_{\text{sty}} - 1},$$

*The following holds:*

1. **Center synchronization.** *For each cluster $c$, $H_c$ maps its rotation center $r_c$ to the north pole $e_d = (0, \ldots, 0, 1)^T$ (i.e., $H_c r_c = e_d$).*

2. **Style-plane separation.** *For each $1 \leq i \leq d_{\text{sty}}$, the component $\tilde{f}_{\text{sty}}^{(i)}(x) \in \mathbb{S}^1$ lies in the $i$-th canonical rotation plane $\pi_i := \text{span}\{e_{2i-1}, e_{2i}\}$. Equivalently, for any angle $\theta \in \mathbb{R}$, define the block-diagonal rotation*

$$\mathcal{R}_i(\theta) := \text{diag}\big( I_2, \ldots, \underbrace{\begin{bmatrix} \cos\theta & -\sin\theta \\ \sin\theta & \cos\theta \end{bmatrix}}_{i\text{-th block}}, \ldots, I_2, \ I_{n - 2 d_{\text{sty}}} \big).$$

*Then $\mathcal{R}_i(\theta)$ rotates only $\tilde{f}_{\text{sty}}^{(i)}$ and leaves $\tilde{f}_{\text{sty}}^{(j)}$ ($j \neq i$) and $\tilde{f}_{\text{res}}$ unchanged.*

Consequently, we focus solely on disentangling the augmentation-induced style latents in the aligned representation.

**Definition 3** (Augmentation-Induced Style Latent Variation). *Let $a(t) \in \mathcal{A}$ be an augmentation parameterized by $t \in \mathcal{T}_a$, its induced style displacement at latent point $z \in \mathcal{Z}_{\text{train}}$ is defined as:*

$$\Delta_a(t, z) := \varphi^{-1}(a(t, \varphi(z))) - z \in \mathbb{R}^m,$$

*which is continuous and bijective in $t$ for fixed $z$.*

*In general, the variation $\Delta_a(t, z)$ may depend on both the augmentation parameter $t$ and the latent state $z$. For example, a centric rotation may induce different latent changes at different positions of the object.*

**Definition 4** ($\mathcal{A}$-realized cyclic vs. non-cyclic on plane $j$). *Let $z^{(j)}$ be the $j$-th style latent and let $a_j$ denote the augmentation varying only this coordinate.*

1. **Intrinsic cyclicity.** *We say $z^{(j)}$ is* intrinsically cyclic *if it has a latent period $p > 0$ (i.e. $z^{(j)} \equiv z^{(j)} + p$, so values live on a circle). Otherwise, it is* intrinsically non-cyclic *(values live on an interval).*

2. *$\mathcal{A}$-**realized cyclicity.** On the training data support, we say the $j$-factor is $\mathcal{A}$-**realized cyclic** if:*

   (a) *$z^{(j)}$ is intrinsically cyclic; and*
   (b) *there exists an ordering of points $(x_1, x_2, \ldots, x_K)$ varying only in $z^{(j)}$ such that*

   $$a_i(t_k, x_k) = a_i(t_{k+1}, x_{k+1}) \quad (k = 1, \ldots, K-1), \quad a_i(t_\ell, x_k) = a_i(t_m, x_1),$$

   *Intuitively, the augmentation steps can be chained to* form a circle *through each sample.*

3. *$\mathcal{A}$-**non-cyclic.** If either the latent is intrinsically non-cyclic, or no such nontrivial closing chain exists, then the $j$-factor is $\mathcal{A}$-non-cyclic.*

## C.2 PROOF OF PROPOSITION 1

**Setup.** Stage 1 pretraining yields unit features $f(x)$ with $k$ spherical $k$-means centers $\{r_\ell\}_{\ell=1}^k \subset \mathbb{S}^{d-1}$. In Stage 2, we freeze the cluster assignment rule $c(x) = \arg\min_\ell \|u(x) - r_\ell\|_2$, build per–cluster Householder maps $H_\ell$ that send $r_\ell$ to the north pole, and optimize on losses $\mathcal{L}_{\text{equiv}}, \mathcal{L}_{\text{radius}}, \mathcal{L}_{\text{theta}}$.

**Lemma 1.** *Under the Assumption 1, if we have a relatively small radius $\omega$ such that $d_{\min} := \min_{\ell \neq r} \|c_\ell - c_r\|_2 \leq 2\omega\sqrt{m}$ and $\mathcal{L}_{\text{radius}} \to 0$, then for any $x$ in cluster $\ell$ and any augmentation $a(t)$, both $x$ and $a(t, x)$ remain assigned to $\ell$.*

*Proof.* Since all intra-cluster data are $\mathcal{A}$-connected, in main content, we know that all the clusters can be separated uniformly in different region after pretraining with Infonce loss, so as to the cluster centers, so there is possible that we found enough small radius $\omega$ satisfying the requirements.

When $\mathcal{L}_{\text{radius}} = 0$, both $H_\ell f(x)$ and $H_\ell f(a(t,x))$ lie on the product torus $S^1(\omega)^{\times m}$ in the aligned chart, so their *pairwise* chord distance is at most $2\omega\sqrt{m}$ (sum of $m$ plane-wise chord bounds). Hence

$$\|f(a(t,x)) - r_\ell\| \leq \|f(a(t,x)) - f(x)\| + \|f(x) - r_\ell\| \leq 2\omega\sqrt{m} + \|f(x) - r_\ell\|.$$

If some other center $r_o$ were closer to $f(a(t,x))$, then by the triangle inequality

$$\|r_\ell - r_o\| \leq \|r_\ell - f(a(t,x))\| + \|f(a(t,x)) - r_o\| < 2\omega\sqrt{m},$$

contradicting $d_{\min} > 2\omega\sqrt{m}$. Thus $c(a(t,x)) = \ell$. $\square$

**Proposition 6** (Convergence to perfect equivariance with three losses). *Under Lemma. 1, consider Stage 2 with losses $\mathcal{L}_{\text{equiv}}, \mathcal{L}_{\text{radius}}, \mathcal{L}_{\text{theta}}$. There exists a solution at which all three reach their optima simultaneously, and all data $x$ and augmentation $a$,*

$$H_{c(x)} f(a(x,t)) = \mathcal{R}(\theta_{a(t)}) H_{c(x)} f(x) \quad \text{for all } x \sim \mathcal{X}, \ a(t) \in \mathcal{A},$$

*Moreover, any nontrivial $a$ induces a non-identity $\mathcal{R}(\theta_{a(t)})$.*

*Proof. (i) Fixed cluster charts.* By Lemma 1, assignments are stable, so a single Householder chart $H_\ell$ consistently aligns both $x$ and $a(t, x)$ to the same "north–pole" frame.

*(ii) Radial normalization.* $\mathcal{L}_{\text{radius}} = 0$ puts features on the product of circles with radius $\omega$ in each style plane. Thus aligned features are points on $S^1(\omega)^{\times m}$, and any valid transformation within the representation must be a block–diagonal rotation $\in SO(2)^m$.

*(iii) Non–degenerate angles.* By $\mathcal{L}_{\text{theta}}$, the angle predictor $G$ is constrained so that $\theta_{a(t)}$ varies with $t$ and is bounded to avoid wrap–through ambiguities; in particular, nontrivial $t$ cannot map to $\theta_{a(t)} = 0$.

*(iv) Equivariance identification.* Minimizing $\mathcal{L}_{\text{equiv}}$ forces $\mathcal{R}(\theta_{a(t)}) H_{c(x)} f(x) = H_{c(x)} f(a(t,x))$ point-wise. On $S^1(\omega)^{\times m}$ this equality pins down $\mathcal{R}(\theta_{a(t)})$ uniquely per $t$ (up to $2\pi$ per axis, ruled out by the range constraint in $\mathcal{L}_{\text{theta}}$). Since nontrivial $t$ produce $\mathcal{R}(\theta_{a(t)}) H_{c(x)} f(x) = H_{c(x)} f(a(t,x))$, the corresponding $R(\theta_{a(t)})$ is non-identity. $\square$

**Remark 1** (Robustness to imperfect pretraining clusters). *Even if Stage 1 yields some misassigned samples, Stage 2 can recover from it. Under Assumption 1 (intra–cluster $\mathcal{A}$–connectivity) and a small radius $\omega$ with margin $d_{\min} > 2\omega\sqrt{m}$, the combined losses $\mathcal{L}_{\text{equi}} + \lambda_{\text{radius}}\mathcal{L}_{\text{radius}} + \lambda_{\text{theta}}\mathcal{L}_{\text{theta}}$ induce the following behavior:*

- **Attraction to the correct chart.** *For a misassigned $x$, the equivariance residual is strictly smaller in its true cluster chart than in its current one (the wrong Householder frame breaks torus consistency). Gradients of $\mathcal{L}_{\text{equi}}$ therefore move $u(x)$ toward the Voronoi cell of the correct center.*

- **Bounded motion & stability once corrected.** *$\mathcal{L}_{\text{radius}}$ confines in–plane motion to the torus $S^1(\omega)^{\times m}$, so the maximal within–cluster displacement during augmentation is $\leq 2\omega\sqrt{m}$. Once $f(x)$ crosses into the correct Voronoi cell, Lemma 1 applies and the assignment remains stable thereafter.*

Now, we consider the case that $\Delta_a(t, z)$ is independent of $z$, i.e., the induced variation depends only on the augmentation parameter $t$, never on the current latent state.

## C.3   PROOF OF PROPOSITION 2

**Lemma 2.** *Let $\mathcal{A} = \{a_i(t_i)\}_{i=1}^{d_{\mathrm{sty}}}$ be $d_{\mathrm{sty}}$ parametric augmentations, each acting solely on, without loss of generality, the $i$-th style coordinate and assigning to $i$ th canonical rotation plane. For every latent $z \in \mathcal{Z}_{\mathrm{train}}$ and parameter $t_i \in \mathcal{T}_{a_i}$ define*

$$\Delta_{a_i}(t_i, z) = q_{a_i}(t_i)\, e_i, \text{ where } |q_{a_i}(t_i)| \leq M_i q_0^{(i)},$$

*where $e_i$ is the $i$-th standard basis vector of $\mathbb{R}^m$, $q_0^{(i)} > 0$ the canonical step and $M_i \in \mathbb{N}_+$ a coordinate bound. For $k \in \mathbb{N}^{d_{\mathrm{sty}}}$ abbreviate $k \odot q_0 := \sum_j k^{(j)} q_0^{(j)} e_j$.*

**A. Construction of $\mathcal{D}$.**

> **R1  Seeds.** $\mathcal{D}^0 := \{\, 0, e_1, \ldots, e_{d_{\mathrm{sty}}}\,\}$.

> **R2  Two-of-three closure.** *Let $k, k_1 \in \mathbb{Z}^{d_{\mathrm{sty}}}$ satisfy $|k^{(i)}|, |(k+k_1)^{(i)}| \leq M_i$ for all $i$. Suppose there exist latent points*
>
> $$z_A, \ z_B := z_A + k \odot q_0, \ z_C := z_A + (k + k_1) \odot q_0,$$
>
> *in $\mathcal{Z}$ such that at least two of $\{z_A, z_B, z_C\}$ belong to the training set $\mathcal{Z}_{\mathrm{train}}$, then, whenever two vectors among $\{k, k_1, k+k_1\}$ already lie in the current set $\mathcal{D}^n$, insert the third to $\mathcal{D}^{n+1}$*

*Iterating **R2** yields an ascending chain $\mathcal{D}^0 \subset \mathcal{D}^1 \subset \cdots$; abuse $\mathcal{D} := \bigcup_{n \geq 0} \mathcal{D}^n$.*

**B. Training-time equivariance**   *For each augmentation $a_i(t_i)$, there is a transformation network $g_i : \mathcal{T}_{a_i} \to [-\pi, \pi]$. Assume the training satisfies:*

> **E1  Cluster-wise perfect equivariance.** *All the data are assigned to the corresponding cluster determined by $\mathcal{A}$. For any index set $S \subseteq \{1, \ldots, d_{\mathrm{sty}}\}$, parameters $(t_i)_{i \in S}$, and all $z \in \mathcal{Z}_{train}$, calling only $\{a_i(t_i)\}_{i \in S}$ yields*
>
> $$\tilde{f}\big(a_S(t, \varphi(z))\big) = \mathcal{R}(\sum_{i \in S} g_i(t_i)\, e_i)\tilde{f}(\varphi(z)),$$

> **E2  Non-zero canonical angle.** *$\exists t_+^{(i)} \in \mathcal{T}_{a_i}$ such that $q_{a_i}(t_+^{(i)}) = q_0^{(i)}$ and $g_i(t_+^{(i)}) = \theta_0^{(i)} \neq 0$.*

> **E3  Slice non-degeneracy.** *$\|\tilde{f}_{\mathrm{sty}}^{(i)}(\varphi(z))\|_2 = r > 0$ for all $z \in \mathcal{Z}_{\mathrm{train}}$ and all $i = 1, \cdots, d_{\mathrm{sty}}$.*

**C. Conclusion.**   (i) *$\mathcal{D}$ is well-defined and is the unique minimal subset satisfying **Seeds + Two-of-three closure***. (ii) *For any $k \in \mathcal{D}$ one may form a parameter tuple $t(k) = (t_1, \ldots, t_{d_{\mathrm{sty}}})$ satisfying*

$$q_{a_i}(t_i) = \begin{cases} k^{(i)}\, q_0^{(i)}, & k^{(i)} \neq 0, \\ 0 \quad (\text{either by omitting } a_i \text{ or } t_i \text{ with } q_{a_i}(t_i) = 0), & k^{(i)} = 0, \end{cases}$$

*and with this choice, write $g(t(k)) = (g_1(t1), \cdots, g_{\mathrm{sty}}(t_{\mathrm{sty}}))^T$*

$$g\big(t(k)\big) = k \odot \theta_0, \qquad \theta_0 := \big(\theta_0^{(1)}, \ldots, \theta_0^{(d_{\mathrm{sty}})}\big)^T.$$

Intuition: On the discrete set of style displacements actually "met" during training, the learned rotation angles are forced to be exactly linear in the semantic displacement. The linearity can be proved by propagating additivity from single-step moves to all reachable displacements.

*Proof. (i)* Rule **R2** always adds precisely the missing element of a triple $\{k, k_1, k+k_1\}$. The outcome after any finite sequence of rule applications therefore depends solely on the set of admissible triples, not on their order. Minimality follows by construction.

*(ii)* We prove the linearity by induction over the depth at which $k$ enters $\mathcal{D}$.

Base.

- *Zero vector.* Take the empty call of augmentations (no $a_i$ invoked). Perfect equivariance then supplies $\tilde{f}(\varphi(z)) = \mathcal{R}(g(t(0)))\tilde{f}(\varphi(z))$, since each $g_i(t_i)$ is in $[-\pi, \pi]$, we have $g(t(0)) = 0$.

- *Unit vector $e_i$* Activate *only* the $i$-th augmentation with its canonical parameter $t_+^{(i)}$ and leave every other augmentation uncalled: $a_{\{i\}}(t) = \{a_i(t_+^{(i)})\}$. Because $q_{a_i}(t_+^{(i)}) = q_0^{(i)}$ while all other displacements are zero, the latent displacement equals $e_i \odot q_0$. Perfect equivariance therefore yields

$$\tilde{f}\big(a_i(t_+^{(i)}), \varphi(z))\big) = \mathcal{R}_i\big(g_i(t_+^{(i)})\big)\,\tilde{f}(\varphi(z)),$$

and **E2** gives $g_i(t_+^{(i)}) = \theta_0^{(i)} \neq 0$. Consequently

$$g(t(e_i)) = (0, \ldots, \theta_0^{(i)}, \ldots, 0) = e_i \odot \theta_0.$$

Hence $g(t(k)) = k \odot \theta_0$ holds for every $k \in \mathcal{D}^0$.

Induction step. Assume $g(t(l)) = \ell \odot \theta_0$ for $\forall \ell \in \mathcal{D}^n$. Let **R2** add the previously missing vector of the triple $\{k, k_1, k + k_1\}$. Because the rule and the triple are permutation-symmetric, we may *without loss of generality* suppose $k$, $k_1 \in \mathcal{D}^n$, $k + k_1 \notin \mathcal{D}^n$. Likewise, the same symmetry lets us arrange the latent points so that $z_A, z_B \in \mathcal{Z}_{\text{train}}$ and $z_C \in \mathcal{Z}$. Augmentations are *only executed at training points* and each displacement of triplets is within the range of a single call of a (composite) augmentation, so all paths below are legal.

$$\begin{array}{lll} \textbf{Path I:} & z_A \xrightarrow{k \odot q_0} z_B & \tilde{f}(\varphi(z_B)) = \mathcal{R}(g(t(k)))\,\tilde{f}(\varphi(z_A)), \\[2mm] \textbf{Path II:} & z_B \xrightarrow{k_1 \odot q_0} z_C & \tilde{f}(\varphi(z_C)) = \mathcal{R}(g(t(k_1)))\,\tilde{f}(\varphi(z_B)), \\[2mm] \textbf{Path III:} & z_A \xrightarrow{(k+k_1) \odot q_0} z_C & \tilde{f}(\varphi(z_C)) = \mathcal{R}(g(t(k+k_1)))\,\tilde{f}(\varphi(z_A)), \end{array}$$

Composing Paths I and II and comparing with Path III, canceling the non-degenerated $\tilde{f}(\cdot)$, we should have the following matrix identity

$$\mathcal{R}(g(t(k+k_1))) = \mathcal{R}(g(t(k_1)))\mathcal{R}(g(t(k))).$$

Because $\theta \mapsto R_\theta$ is injective on $[-\pi, \pi]$ and the rotation blocks act on disjoint 2-planes (or add within the same plane), we deduce $g\big(t(k+k_1)\big) = g(t(k)) + g\big(t(k_1)\big)$. Applying the induction hypothesis yields $g(t(k+k_1)) = (k+k_1) \odot \theta_0$. Therefore $g(t(\ell)) = \ell \odot \theta_0$ holds for every $\ell \in \mathcal{D}^{(n+1)}$, closing the induction. $\qquad\square$

Using the above lemma, we are ready to prove that, on the discrete grid of latent displacements seen in training, any two features in the same cluster differ only by a block-diagonal rotations whose angle vector is linear in their latent difference, as formalized in the following proposition.

**Proposition 6a** (Weak cluster-wise disentanglement on the discrete grid)**.** *Assume*

    *A1 **Discrete linearity of** $g$ (Lemma 2): for all $k \in \mathcal{D}$ there exist parameters $t(k)$ with $q_{a_i}(t_i) = k^{(i)} q_0^{(i)}$ such that $g(t(k)) = k \odot \theta_0$.*

    *A2 **Chain connectivity** For any $z_A, z_B \in \mathcal{Z}_{\text{train}}$ with $z_A \sim_{\mathcal{A}} z_B$, there exists a finite chain*

$$z_A = z_0,\ z_1, \ldots, z_n = z_B \quad (\text{all in } \mathcal{Z}_{\text{train}})$$

*such that for each edge $(z_j, z_{j+1})$ there exist augmentations $u_j^+(t_j^+)$ and $u_j^-(t_j^-)$ with*

$$\varphi^{-1}\big(u_j^+(t_j^+, \varphi(z_j))\big) \;=\; \varphi^{-1}\big(u_j^-(t_j^-, \varphi(z_{j+1}))\big) \;=:\; z_j^\star \in \mathcal{Z},$$

*and associated style displacements $k_j^+ \odot q^0$ and $k_j^- \odot q^0$, respectively, with $k_j^+, k_j^- \in \mathcal{D}$.*

*Then every cluster admits **weak disentanglement** on the discrete grid: for any two training points $x_A, x_B$ in the same cluster,*

$$\tilde{f}\big(\varphi(z_B)\big) = \mathcal{R}(k \odot \theta_0)\tilde{f}\big(\varphi(z_A)\big), \quad k \odot q_0 = z_B - z_A$$

*Proof.* Fix an edge $(z_j, z_{j+1})$ of the chain in **A2**. By perfect equivariance and **A1** (which provides $g(t(k)) = k \odot \theta_0^{(i)}$ on $\mathcal{D}$), we obtain

$$\tilde{f}(\varphi(z_j^\star)) = \mathcal{R}(k_j^+ \odot \theta_0)\tilde{f}(\varphi(z_j)), \qquad \tilde{f}(\varphi(z_j^\star)) = \mathcal{R}(k_j^- \odot \theta_0)\tilde{f}(\varphi(z_{j+1})).$$

Equating and right-multiplying by the inverse rotation yields

$$\tilde{f}(\varphi(z_{j+1})) = \mathcal{R}((k_j^+ - k_j^-) \odot \theta_0)\tilde{f}(\varphi(z_j)).$$

Composing these relations along $j = 0, \ldots, n-1$, and using commutativity of block-diagonal rotations, gives

$$\tilde{f}(\varphi(z_B)) = \prod_{j=0}^{n-1} \mathcal{R}\Big((k_j^+ - k_j^-) \odot \theta_0\Big)\tilde{f}(\varphi(z_A)) = \mathcal{R}\Big(\sum_{j=0}^{n-1}(k_j^+ - k_j^-) \odot \theta_0\Big)\tilde{f}(\varphi(z_A)).$$

By the construction of edges, $z_{j+1} - z_j = (k_j^+ - k_j^-) \odot q_0$. Summing over $j$ gives

$$z_B - z_A = \sum_{j=0}^{n-1}(k_j^+ - k_j^-) \odot q_0 = k \odot q_0.$$

Hence,

$$\tilde{f}(\varphi(z_B)) = \mathcal{R}(k \odot \theta_0)\tilde{f}(\varphi(z_A)).$$

$\square$

Next, we will prove that the linearity succeed with continuous augmentation.

**Definition 5.** *A realized displacement $s \in [-M_i, M_i]$ means there are $z_0, z_1 \in \mathcal{Z}_{\text{train}}$ with $z_1 - z_0 = se_i$ and almost all intermediate states obtained by applying $a_i(t_i)$ to these two points remain inside $\mathcal{Z}$.*

**Lemma 3.** *Let $\mathcal{A} = \{a_i(t_i)\}_{i=1}^{d_{\text{sty}}}$ be $d_{\text{sty}}$ parametric augmentations, each acting solely on, without loss of generality, the $i$-th style coordinate and assigning to the $i$-th canonical rotation plane. For every latent $z \in \mathcal{Z}_{\text{train}}$ and parameter $t_i \in \mathcal{T}_{a_i}$ define*

$$\Delta_{a_i}(t_i, z) := q_{a_i}(t_i)\, e_i, \text{ where } q_{a_i}(t_i) \in [-M_i, M_i]$$

*where $e_i$ is the $i$-th standard basis vector of $\mathbb{R}^m$ and $[-M_i, M_i] \subset \mathbb{R}$ is coordinate bound.*

*Assume the training-time equivariance conditions **B** (items **E1** and **E3**) in Lemma 2, and each $g_i : \mathcal{T}_{a_i} \to [-\pi, \pi]$ and $q_{a_i}$ are continuous. Write $\widehat{g}_i(d) := g_i\big(q_{a_i}^{-1}(d)\big), d \in [-M_i, M_i]$, and assume $\widehat{g}_i(d) \neq 0$ for any non-zero displacement. We have*

1. *$\widehat{g}_i(0) = 0$;*

2. *For every realized displacement $s$, the one-step identity holds on admissible pairs:*

$$\widehat{g}_i(d) - \widehat{g}_i(d - s) = \theta_s \quad \text{whenever } d, d - s \in [-M_i, M_i], \qquad \theta_s := \widehat{g}_i(s) \neq 0. \quad (14)$$

*Proof.* For any $z \in \mathcal{Z}_{\text{train}}$, Condition **E1** gives $\tilde{f}(\varphi(z)) = \mathcal{R}_i(\widehat{g}_i(0))\tilde{f}(\varphi(z))$, since each $g_i(t_i)$ is in $[-\pi, \pi]$, we have $\widehat{g}_i(0) = 0$. Thus prove (1).

For any realized displacement $s$, fix an arbitrary $d \in [-M_i, M_i]$ with $d - s \in [-M_i, M_i]$, we have:

$$\textbf{Path I:} \qquad z_0 \xrightarrow{se_i} z_1 \qquad \tilde{f}(\varphi(z_1)) = \mathcal{R}_i(\widehat{g}_i(r))\,\tilde{f}(\varphi(z_0)),$$

$$\textbf{Path II:} \qquad z_1 \xrightarrow{(d-s)e_i} z_2 \qquad \tilde{f}(\varphi(z_2)) = \mathcal{R}_i\big(\widehat{g}_i(d - s)\big)\,\tilde{f}(\varphi(z_1)),$$

$$\textbf{Path III:} \qquad z_0 \xrightarrow{de_i} z_2 \qquad \tilde{f}(\varphi(z_2)) = \mathcal{R}_i\big(\widehat{g}_i(d)\big)\,\tilde{f}(\varphi(z_0)).$$

Composing Paths I and II and comparing with Path III gives

$$\mathcal{R}_i\big(\widehat{g}_i(d)\big)\tilde{f}(\varphi(z_0)) = \mathcal{R}_i\big(\widehat{g}_i(d - s)\big)\mathcal{R}_i(\widehat{g}_i(r))\tilde{f}(\varphi(z_0))$$

By the condition **E3**, $\tilde{f}(\varphi(z_0))$ is non-degenerated, we have

$$\mathcal{R}_i\big(\widehat{g}_i(d)\big) = \mathcal{R}_i\big(\widehat{g}_i(d-s)\big)\mathcal{R}_i(\widehat{g}_i(r)) = \mathcal{R}_i\big(\widehat{g}_i(d-s) + \widehat{g}_i(r)\big).$$

By injectivity of the rotation map, this yields $\widehat{g}_i(d) - \widehat{g}_i(d-s) = \theta_s$ with $\theta_s := \widehat{g}_i(s) \neq 0$. $\qquad\square$

**Lemma 4.** *Under the assumption of Lemma 3. Let $S \subset [-M_i, M_i]$ be a nonempty family of realized steps that is closed under admissible common multiples: for any $r, s \in S$, there exist $m, n \in \mathbb{Z}$ such that $L := mr = ns \in [-M_i, M_i]$. Then the quantity*

$$c_S := \frac{\theta_s}{s} \neq 0 \qquad (s \in S)$$

*is well-defined (independent of $s \in S$), and*

$$\widehat{g}_i(d) = c_S\,d + H_S(d), \qquad H_S(d-s) = H_S(d) \quad (\forall s \in S),$$

*on every subinterval where the pairs in Equation equation 14 are admissible. In particular, $H_S$ is $s$-periodic for each $s \in S$.*

*Proof.* Apply the three paths in Lemma 3 at the sequence $d_j := jr$ $(j = 1, \ldots, m)$, all of which lie in $[-M_i, M_i]$:

$$\widehat{g}_i(d_j) - \widehat{g}_i(d_{j-1}) = \theta_r \quad \Rightarrow \quad \widehat{g}_i(L) - \widehat{g}_i(0) = \sum_{j=1}^{m} \theta_r = m\,\theta_r.$$

Similarly, using step $s$ at $e_\ell := \ell s$ $(\ell = 1, \ldots, n)$,

$$\widehat{g}_i(L) - \widehat{g}_i(0) = n\,\theta_s.$$

Therefore

$$\frac{\theta_r}{r} = \frac{\theta_s}{s} =: c_S.$$

So the slope is independent of the realized step. Define the common remainder $H_S(d) := \widehat{g}_i(d) - c_S d$.

$$H_S(d) - H_S(d-s) = \big(\widehat{g}_i(d) - \widehat{g}_i(d-s)\big) - \frac{\widehat{g}_i(s)}{s}\big(d - (d-s)\big) = \widehat{g}_i(s) - \widehat{g}_i(s) = 0.$$

Therefore

$$\widehat{g}_i(d) = c_S d + H_S(d), \qquad H_S(d-s) = H_S(d) \quad (\forall\, s \in S). \tag{15}$$

$\square$

**Lemma 5.** *Under the conditions of Lemma 3 and Lemma 4. Let $S \subset [-M_i, M_i]$ be a nonempty common multiplier-closed family of realized displacements, so that $c_S = \theta_s/s$ is well defined for all $s \in S$ and $H_S(d) := \widehat{g}_i(d) - c_S d$ is $s$-periodic on admissible pairs for every $s \in S$. Suppose moreover that $\widehat{g}_i : [-M_i, M_i] \to \mathbb{R}$ be $L$-Lipschitz:*

$$|\widehat{g}_i(x) - \widehat{g}(y)| \leq L\,|x - y| \qquad \forall\, x, y \in [-M_i, M_i].$$

    (a) ***Uniform small remainder from the smallest step in a family.*** *Assume $S$ contains the smallest magnitude step*

$$s_* := \arg\min_{s \in S} |s|.$$

    *Then, using only the periodicity provided by Lemma 4 for the family $S$,*

$$|H_S(d)| \leq 2L\,|s_*|.$$

(b) **Near-loop slope matching across two families.** *Let $S_1, S_2 \subset [-M_i, M_i]$ be two common multiplier-closed families with slopes $c_{S_1}, c_{S_2}$ (from Lemma 4). Suppose there exist $r \in S_1$, $s \in S_2$ and integers $m, n \neq 0$ such that, for some basepoint $d \in [-M_i, M_i]$, the chains*

$$d \to d + mr, \qquad d \to d + ns$$

*are admissible and the endpoints are $\delta$-close: $|mr - ns| \leq \delta$. Then*

$$|c_{S_1} - c_{S_2}| \leq \frac{2L\,\delta}{\max\{\,m|r|,\, n|s|\,\}}$$

*and, for any $x \in [-M_i, M_i]$,*

$$\left| H_{S_1}(x) - H_{S_2}(x) \right| \leq M_i\,|c_{S_1} - c_{S_2}| \leq \frac{2L\,M_i\,\delta}{\max\{\,m|r|,\, n|s|\,\}}.$$

*Moreover, each family remainder is $2L\delta$–almost periodic across the other family's step:*

$$|H_{S_1}(d + ns) - H_{S_1}(d)| \leq 2L\delta, \qquad |H_{S_2}(d + mr) - H_{S_2}(d)| \leq 2L\delta,$$

*Proof.* By Lemma 4 the quotient $c_S = \theta_s/s$ is fixed for any $s \in S$, and $H_S(d) = \widehat{g}(d) - c_S d$ is $s$-periodic for each $s \in S$ on admissible pairs. Moreover, since $\widehat{g}$ is $L$-Lipschitz and $\theta_s = \widehat{g}_i(s) - 0 = \widehat{g}(s) - \widehat{g}(0)$,

$$|c_S| = \left| \frac{\theta_s}{s} \right| = \frac{|\widehat{g}(s) - \widehat{g}(0)|}{|s|} \leq L \qquad \text{for any } s \in S. \tag{$*$}$$

*(a) Uniform bound from the smallest step.* Fix $d \in [-M_i, M_i]$. By $s_*$-periodicity (iterated along the segment joining $d$ to $d - ks_* \in [-M_i, M_i]$ ), choose $k \in \mathcal{Z}$ such that

$$d^\circ := d - ks_* \in [-|s_*|, |s_*|] \cap [-M_i, M_i] \qquad \text{and} \qquad H_S(d) = H_S(d^\circ).$$

Then, using $H_S(0) = \widehat{g}(0)$ and equation $*$,

$$|H_S(d)| = |H_S(d^\circ)| = \left| \widehat{g}(d^\circ) - c_S d^\circ \right| \leq |\widehat{g}(d^\circ) - \widehat{g}(0)| + |c_S|\,|d^\circ|$$
$$\leq L\,|d^\circ| + L\,|d^\circ| \leq 2L\,|s_*|.$$

This proves the claimed uniform bound.

*(b) Near-loop slope matching and cross-almost-periodicity.* By admissibility and additivity along the two chains,

$$\widehat{g}(d + mr) = \widehat{g}(d) + m\,\theta_r = \widehat{g}(d) + m\,c_{S_1}\,r, \qquad \widehat{g}(d + ns) = \widehat{g}(d) + n\,\theta_s = \widehat{g}(d) + n\,c_{S_2}\,s.$$

Subtract and use the $L$-Lipschitz property:

$$\left| m\,c_{S_1}\,r - n\,c_{S_2}\,s \right| = \left| \widehat{g}(d + mr) - \widehat{g}(d + ns) \right| \leq L\,|mr - ns| \leq L\,\delta.$$

By the triangle inequality,

$$\left| m\,c_{S_1}\,r - n\,c_{S_2}\,s \right| = \left| mr\,(c_{S_1} - c_{S_2}) + c_{S_2}\,(mr - ns) \right|$$
$$\geq |mr||c_{S_1} - c_{S_2}| - |c_{S_2}||(mr - ns)|$$

Then,

$$|c_{S_1} - c_{S_2}| \leq \frac{\left| m\,c_{S_1}\,r - n\,c_{S_2}\,s \right| + |c_{S_2}|\,|mr - ns|}{m|r|}$$
$$\leq \frac{L\,\delta + |c_{S_2}|\,|mr - ns|}{m|r|}$$
$$\leq \frac{L\,\delta + L\,\delta}{m|r|} \quad \text{Using } (*)$$
$$= \frac{2L\,\delta}{m|r|}$$

Interchanging the roles of $(r, m)$ and $(s, n)$ gives

$$|c_{S_1} - c_{S_2}| \le \frac{2L\,\delta}{n|s|}.$$

Combining the two bounds yields the displayed estimate for $|c_{S_1} - c_{S_2}|$,

$$|c_{S_1} - c_{S_2}| \;\le\; \frac{2L\,\delta}{\max\{\,m|r|,\,n|s|\,\}}$$

For the remainders,

$$|H_{S_1}(x) - H_{S_2}(x)| = |(c_{S_2} - c_{S_1})\,x| \le |x|\,|c_{S_1} - c_{S_2}| \le M_i\,|c_{S_1} - c_{S_2}|,$$

which gives the second display.

Finally, for the cross-almost-periodicity, note that

$$|H_{S_1}(d + ns) - H_{S_1}(d)| = \left|\widehat{g}(d + ns) - \widehat{g}(d) - c_{S_1}\,ns\right|$$
$$= \left|n\,c_{S_2}\,s - c_{S_1}\,ns\right| = n|s|\,|c_{S_2} - c_{S_1}|.$$

Since $H_{S_1}$ is $r$-periodic and $H_{S_1}(d + mr) = H_{S_1}(d)$

$$|H_{S_1}(d + ns) - H_{S_1}(d)| = \left|H_{S_1}(d + ns) - H_{S_1}(d + mr)\right|.$$

Using the previously derived bound with the triangle trick, we have

$$\left|H_{S_1}(d + ns) - H_{S_1}(d)\right| = \left|\widehat{g}(d + ns) - \widehat{g}(d + mr) - c_{S_1}(ns - mr)\right|$$
$$\le |\widehat{g}(d + ns) - \widehat{g}(d + mr)| + |c_{S_1}|\,|ns - mr|$$
$$\le L\,\delta + L\,\delta = 2L\,\delta.$$

The estimate for $H_{S_2}$ is identical by symmetry. This completes the proof. $\qquad\square$

Lemma 3 simply shows that along a fixed style coordinate, each realized step $s$ must always induce the same angular increment $\theta_s$, independent of where you start. Lemma 4 then says that, within any family $S$ of steps that share common multiples, these increments are all compatible: they define a single coomom slope $c_S$, and $\widehat{g}_i$ decomposes into a linear part $c_S d$ plus an $S$–periodic remainder $H_S$. Finally, Lemma 5 uses Lipschitz regularity to show that this periodic remainder cannot oscillate much: small steps force $H_S$ to be uniformly small, and "almost closed loops" built from two step families force their slopes and remainders to almost coincide. In combination, these lemmas say that on a dense network of realized displacements, the transformation network $g_i$ is essentially linear in latent $z^{(i)}$ with only a very small oscillatory part.

**Corollary 1** (Linearity from either an irrational pair or sub-step refinement). *Fix a coordinate $i$. Assume Lemma 3 (one-step identity/additivity on admissible chains) and Lemma 4 (common slope on a common multiplier-closed family of realized displacements), so that for any realized step $r > 0$ we have*

$$c_i \;=\; \frac{\theta_r}{r} \ne 0 \quad \text{(independent of $r$)}, \qquad H_i(d) \;:=\; \widehat{g}_i(d) - c_i\,d$$

*and $H_i$ is $r$-periodic on admissible pairs. If, in addition, either*

    *(i) there exist realized displacements $r_i \in (0, M_i]$ and $\beta_i r_i \in (0, M_i]$ with $\beta_i \in (0,1) \setminus \mathbb{Q}$, or*

    *(ii) there exists a sequence of realized displacements $r_n = \frac{r_0}{n}$ with $r_n \to 0$,*

*then $H_i \equiv 0$ on $[-M_i, M_i]$ and hence*

$$\widehat{g}_i(d) \;=\; c_i\,d \quad \text{for all } d \in [-M_i, M_i], \qquad g_i(t_i) \;=\; c_i\,q_{a_i}(t_i) \quad \text{for all } t_i \in \mathcal{T}_{a_i}.$$

*Proof.* We give separate arguments.

*Case (i): incommensurate pair $r_i$ and $\beta_i r_i$.* Fix $r_i \in (0, M_i]$ realized and $\beta_i \in (0,1) \setminus \mathbb{Q}$ such that $\beta_i r_i$ is realized. From Equation 15 with $s = r_i$,

$$H_i(d) = H_i(d - r_i) \qquad \text{whenever } d, d - r_i \in [-M_i, M_i].$$

Next, derive the precise $\beta_i r_i$-increment identity for $H_i$. For any $l$ with $l, l - \beta_i r_i \in [-M_i, M_i]$,

$$
\begin{aligned}
H_i(l - \beta_i r_i) &= \widehat{g}_i(l - \beta_i r_i) - \frac{\theta_0^{(i)}}{r_i}(l - \beta_i r_i) \\
&= \left(\widehat{g}_i(l) - \widehat{g}_i(\beta_i r_i)\right) - \frac{\theta_0^{(i)}}{r_i}l + \theta_0^{(i)}\beta_i \\
&= \left(\widehat{g}_i(l) - \frac{\theta_0^{(i)}}{r_i}l\right) - \left(\widehat{g}_i(\beta_i r_i) - \theta_0^{(i)}\beta_i\right) \\
&= H_i(l) - C, \tag{16}
\end{aligned}
$$

where we set

$$C := \widehat{g}_i(\beta_i r_i) - \theta_0^{(i)}\beta_i \quad \text{(a constant independent of } l\text{)}.$$

Since $r_i \leq M_i$, it suffices to analyze $H_i$ on $I := (0, r_i] \subset [-M_i, M_i]$ and extend by $r_i$-periodicity. Define the sequence $\{d_m\}_{m \geq 1} \subset (0, r_i]$ by

$$d_1 := r_i - \beta_i r_i, \qquad d_{m+1} := \begin{cases} d_m - \beta_i r_i, & d_m \geq \beta_i r_i, \\ d_m - \beta_i r_i + r_i, & d_m < \beta_i r_i. \end{cases}$$

Claim 1. For all $m \geq 1$,
$$H_i(d_m) = -mC, \qquad d_m \in (0, r_i]. \tag{17}$$

*Proof of Claim 1.* Base case $m = 1$: taking $l = r_i$ in equation 16 (both $r_i$ and $r_i - \beta_i r_i$ lie in $I$),

$$H_i(d_1) = H_i(r_i - \beta_i r_i) = H_i(r_i) - C = 0 - C = -C.$$

Induction step $m \to m + 1$: If $d_m \geq \beta_i r_i$, then $d_{m+1} = d_m - \beta_i r_i \in (0, r_i]$ and equation 16 with $l = d_m$ gives $H_i(d_{m+1}) = H_i(d_m) - C$. If instead $d_m < \beta_i r_i$, then $d_{m+1} = d_m - \beta_i r_i + r_i \in (0, r_i]$ and equation 16 yields $H_i(d_m) - H_i(d_m - \beta_i r_i) = C$; by $r_i$-periodicity, $H_i(d_m - \beta_i r_i) = H_i(d_m - \beta_i r_i + r_i) = H_i(d_{m+1})$, hence again $H_i(d_{m+1}) = H_i(d_m) - C$. This proves equation 17.

Claim 2. The sequence $\{d_m\}$ is dense in $(0, r_i]$.

*Proof of Claim 2.* By construction,

$$d_{m+1} \equiv d_m - \beta_i r_i \pmod{r_i}, \quad \text{so} \quad d_m \equiv r_i - m\beta_i r_i \pmod{r_i}.$$

To pass from a congruence class to its representative in $(0, r_i]$, use the identity

$$x \bmod r_i = x - r_i \lfloor \frac{x}{r_i} \rfloor = r_i \left\{ \frac{x}{r_i} \right\} \quad \text{for any } x \in \mathbb{R},$$

where $\lfloor x \rfloor$ is the greatest integer less than or equal to $x$ and $\{x\} = x - \lfloor x \rfloor$ is the fractional part in $[0, 1)$. Applying this to $x = r_i - m\beta_i r_i = r_i(1 - m\beta_i)$ and using the elementary relation $\{1 - y\} = 1 - \{y\}$ for $y \notin \mathbb{Z}$ (which holds here because $\beta_i \notin \mathbb{Q}$ implies $m\beta_i \notin \mathbb{Z}$ for all $m \geq 1$), we obtain

$$d_m \equiv \left(r_i - m\beta_i r_i\right) \bmod r_i = r_i \left\{ \frac{r_i - m\beta_i r_i}{r_i} \right\} = r_i \left\{ 1 - m\beta_i \right\} = r_i \left(1 - \{m\beta_i\}\right). \tag{18}$$

Since $\beta_i \notin \mathbb{Q}$, the set $\{\{m\beta_i\} : m \in \mathbb{N}\}$ is dense in $[0, 1)$ (Kronecker's theorem), hence $\{d_m\}$ is dense in $(0, r_i]$ by the continuity of $x \mapsto r_i(1 - x)$ on $[0, 1)$. In particular, there is a subsequence $d_{m_\ell} \to 0^+$.

From equation 17, $H_i(d_{m_\ell}) = -m_\ell C$. Since $d_{m_\ell} \to 0^+$ and $H_i$ is continuous with $H_i(0) = 0$, we have $H_i(d_{m_\ell}) \to 0$, hence $-m_\ell C \to 0$. As $m_\ell \to \infty$, the only possibility is $C = 0$. With

$C = 0$, equation 16 gives $H_i(l - \beta_i r_i) = H_i(l)$ on admissible pairs, so $H_i$ is both $r_i$- and $\beta_i r_i$-periodic. From equation 17 we also have $H_i(d_m) = 0$ for all $m$, so $H_i$ vanishes on the dense set $\{d_m\} \subset (0, r_i]$; by continuity, $H_i \equiv 0$ on $(0, r_i]$, and then by $r_i$-periodicity $H_i \equiv 0$ on $[-M_i, M_i]$.

*Case (ii): periods accumulating at $0$.* By hypothesis, for each $N$ large enough the sub-step $r_N := r_0/N$ is realized and $H_i$ is $r_N$-periodic on admissible pairs by Lemma 4. Since $r_N \to 0$, we prove $H_i$ is constant on $[-M_i, M_i]$ by a uniform-continuity argument.

Because $H_i$ is continuous on the compact interval $[-M_i, M_i]$, it is uniformly continuous: there exists a modulus $\omega(\cdot)$ with $\omega(\delta) \to 0$ as $\delta \to 0$ and $|H_i(x) - H_i(y)| \le \omega(|x-y|)$ for all $x, y \in [-M_i, M_i]$. Fix arbitrary $x, y \in [-M_i, M_i]$ and $\varepsilon > 0$. Choose $N$ so large that $r_N < \varepsilon$ and $\omega(r_N) < \varepsilon$, and such that the $r_N$-periodicity applies to every pair along the short segment we construct next. Pick $m \in \mathbb{Z}$ with $|x - (y + mr_N)| \le r_N$ and with the points $y, y + r_N, \ldots, y + mr_N$ all in $[-M_i, M_i]$ (possible since $r_N$ is small). Then $H_i(y + mr_N) = H_i(y)$ by $r_N$-periodicity, and

$$|H_i(x) - H_i(y)| = |H_i(x) - H_i(y + mr_N)| \le \omega\big(|x - (y + mr_N)|\big) \le \omega(r_N) < \varepsilon.$$

Since $\varepsilon > 0$ and $x, y$ are arbitrary, $H_i$ is constant on $[-M_i, M_i]$; evaluating at $0$ gives $H_i \equiv 0$.

In both cases, $H_i \equiv 0$ on $[-M_i, M_i]$, hence $\widehat{g}_i(d) = c_i d$ for all $d \in [-M_i, M_i]$, and substituting $d = q_{a_i}(t_i)$ yields $g_i(t_i) = c_i q_{a_i}(t_i)$ for all $t_i \in \mathcal{T}_{a_i}$. Apply the same argument to all $i \in [1, \cdots, d_{\text{sty}}]$, we can get the linearity of $g_i(t_i)$ for all $i$. $\qquad\square$

Intuition: The previous lemmas (Lemma 3, 4, 5) tell us that, along a single style coordinate, the learned rotation angle can be written as a linear function plus a periodic remainder that repeats with every realized step. Corollary 1 uses an extra richness assumption on the realized steps (either an irrational pair of step sizes, or steps that can be refined to arbitrarily small increments) to show that such a continuous, multi-periodic remainder must in fact be constant. Since the remainder is zero at zero displacement, it must be zero everywhere, so the angle becomes exactly linear in the latent displacement.

**Remark 2** (Empirical remainder and quantitative guarantees). *In our experiments the measured remainder $H_i(d) = \widehat{g}_i(d) - c_i d$ is numerically $\approx 0$, indicating that the learned $g_i$ are effectively linear. This aligns not only with the qualitative routes in Corollary 1, but also with the quantitative bounds of Proposition 5:*

(a) *(Uniform small remainder from the smallest realized step) If the realized family contains a smallest magnitude step $s_* > 0$, then*

$$\sup_{d \in [-M_i, M_i]} |H_i(d)| \le 2L|s_*|.$$

*Hence, as training realizes finer sub-steps ($|s_*| \to 0$), the remainder vanishes uniformly.*

(b) Near-loop self-consistency within the same augmentation. *For any realized step sizes $r, s$ and integers $m, n \ge 1$, if there is a basepoint $d \in [-M_i, M_i]$ such that the chains $d \to d + mr$ and $d \to d + ns$ are admissible and their endpoints are $\delta$-close ($|mr - ns| \le \delta$), then*

$$\left| \frac{\theta_r}{r} - \frac{\theta_s}{s} \right| \le \frac{2L\delta}{\max\{m|r|, n|s|\}}, \qquad |H_i(d + ns) - H_i(d + mr)| \le 2L\delta.$$

*Thus, tighter near-loops ($\delta$ small, $m, n$ large) force the inferred slopes to agree and make the remainder nearly periodic across different realized steps.*

*These quantitative effects explain why we observe $H_i \approx 0$ in practice under a single continuous augmentation: sub-step refinement drives (a) to zero, and even when only approximate rational relations are realized, (b) keeps the learned slopes and remainders tightly controlled.*

**Proposition 6b** (Weak cluster-wise disentanglement in the continuous case). *Assume*

    **B1 Linearity of $g$ (Colloary 1).** For $\forall i = [1, \cdots, d_{\text{sty}}]$, $\forall t_i \in \mathcal{T}_{a_i}$, $g_i(t_i) = c_i q_{a_i}(t_i)$, write $g(t) := (g_1(t_1), \cdots, g_{d_{\text{sty}}}(t_{d_{\text{sty}}}))^T$, $c_0 := (c_1, \cdots, c_{d_{\text{sty}}})^T$.

**B2 Chain-connectivity** *For any $z_A, z_B \in \mathcal{Z}_{\text{train}}$, with $z_A \sim_{\mathcal{A}} z_B$, there exists a finite chain*

$$z_A = z_0, z_1, \ldots, z_n = z_B \quad (\text{all in } \mathcal{Z}_{\text{train}})$$

*such that for each edge $(z_j, z_{j+1})$ there are two augmentations $u_j^+(t_j^+), u_j^-(t_j^-)$ and a latent point $z_j^\star \in \mathcal{Z}$ with*

$$\varphi^{-1}\big(u_j^+(t_j^+, \varphi(z_j))\big) = \varphi^{-1}\big(u^-(t_j^-, \varphi(z_{j+1}))\big) =: z_j^\star,$$

*and realized displacements $m_j^+ := u_j^+(t_j^+), \; m_j^- := u_j^-(t_j^-) \in \mathbb{R}^{d_{\text{sty}}}$ satisfying $m_j^{+,(i)}, m_j^{-,(i)} \in [-M_i, M_i]$ for all $i$.*

*Then every cluster admits **weak disentanglement** : for any two training points $x_A, x_B$ in the same cluster,*

$$\tilde{f}\big(\varphi(z_B)\big) = \mathcal{R}(m \odot c_0)\tilde{f}\big(\varphi(z_A)\big), \quad m = z_B - z_A$$

*Proof.* Since perfect equivariance is achieved, for a fixed edge $(z_j, z_{j+1})$ we have

$$\tilde{f}\big(\varphi(z_j^\star)\big) = \mathcal{R}(g(u_j^+))\tilde{f}\big(\varphi(z_j)\big) = \mathcal{R}(m_j^+ \odot c_0)\tilde{f}\big(\varphi(z_j)\big),$$

and also

$$\tilde{f}\big(\varphi(z_j^\star)\big) = \mathcal{R}(g(u_j^-))\tilde{f}\big(\varphi(z_{j+1})\big) = \mathcal{R}(m_j^- \odot c_0)\tilde{f}\big(\varphi(z_{j+1})\big).$$

Canceling the common term yields

$$\tilde{f}\big(\varphi(z_{j+1})\big) = \mathcal{R}\,(m_j^+ - m_j^-) \odot \theta_0\, \tilde{f}\big(\varphi(z_j)\big).$$

Compose over $j = 0, \ldots, n-1$ and use commutativity of block rotations:

$$\tilde{f}\big(\varphi(z_B)\big) = \mathcal{R}\Big(\sum_{j=0}^{n-1}(m_j^+ - m_j^-) \odot c_0\Big)\tilde{f}\big(\varphi(z_A)\big).$$

By construction of each edge, $z_{j+1} - z_j = m_j^+ - m_j^-$, hence $m = z_B - z_A = \sum_{j=0}^{n-1}(m_j^+ - m_j^-)$, which implies that:

$$\tilde{f}\big(\varphi(z_B)\big) = \mathcal{R}\big(m \odot c_0\big)\tilde{f}\big(\varphi(z_A)\big)$$

$\square$

Intuition: This proof is similar to Proposition 6a.

### C.4 PROOF OF PROPOSITION 3

**Definition 6** (Canonical nonnegative feature angle on a style plane). *Fix the counterclockwise orientation of each style plane. For the $i$-th plane, write $\tilde{f}_{\text{sty}}^{(i)}(x) = (u_i(x), v_i(x)) \in \mathbb{S}^1 \subset \mathbb{R}^2$. and two connected points $\varphi(z_A) = x_A \sim_{\mathcal{A}} x_B = \varphi(z_B)$ with $v := z_B - z_A$, denote the principal relative angle by*

$$\theta_i^{\text{pr}}(\tilde{f}(x_B); \tilde{f}(x_A)) := \text{atan2}\Big(\tilde{f}_{\text{sty}}^{(i)}(x_A) \times \tilde{f}_{\text{sty}}^{(i)}(x_B), \tilde{f}_{\text{sty}}^{(i)}(x_A) \cdot \tilde{f}_{\text{sty}}^{(i)}(x_B)\Big)$$

$$= \text{atan2}\Big(u_i(x_A)\,v_i(x_B) - v_i(x_A)\,u_i(x_B),\; u_i(x_A)\,u_i(x_B) + v_i(x_A)\,v_i(x_B)\Big)$$

$$\in (-\pi, \pi].$$

*Define the canonical feature angle*

$$\Theta_i^{\text{ccw}}(\tilde{f}(x_B); \tilde{f}(x_A)) := \begin{cases} \theta_i^{\text{pr}}(\tilde{f}(x_B); \tilde{f}(x_A)), & \theta_i^{\text{pr}} \geq 0, \\ \theta_i^{\text{pr}}(\tilde{f}(x_B); \tilde{f}(x_A)) + 2\pi, & \theta_i^{\text{pr}} < 0, \end{cases} \in [0, 2\pi),$$

*and the directed feature angle (follow the sign of $v^{(i)}$) by*

$$\Theta_i^{\text{dir}}(\tilde{f}(x_B); \tilde{f}(x_A); v^{(i)}) := \begin{cases} \Theta_i^{\text{ccw}}(\tilde{f}(x_B); \tilde{f}(x_A)) \in [0, 2\pi), & v^{(i)} \geq 0, \\ \Theta_i^{\text{ccw}}(\tilde{f}(x_B); \tilde{f}(x_A)) - 2\pi \in (-2\pi, 0], & v^{(i)} < 0. \end{cases}$$

**Proposition 7** (Strong cluster-wise disentanglement). *Let $\mathcal{A} = \{a_i(t_i)\}_{i=1}^{d_{\text{sty}}}$ be $d_{\text{sty}}$ parametric augmentations, each acting solely on the, without loss of generality, $i$-th style coordinate and assigning to $i$ th canonical rotation plane. Under the conditions:*

> **S1 Training-time equivariance with linear head** *(Corollary 1 or Lemma 2). There exist slopes $\theta_0^{(i)}/r_i \neq 0$ (for discrete case $r_i = 1$) such that any induced displacement $v$ by augmentation output angles $g(t) = c_0 \odot v$. i.e., the $i$-th style plane rotates by $c_0^{(i)}v^{(i)}$. W.l.o.g, assume $c_0^{(i)} > 0$ for all $i$.*

> **S2 Connectivity** *(Proposition. 6b or Prop. 6a). Any pair $\varphi(z_A) \sim_{\mathcal{A}} \varphi(z_B)$ is chain-connected with latent displacement $v := z_B - z_A$.*

*Fix the (counterclockwise) orientation per the style plane and let $\Theta_i^{[0,2\pi)}(\cdot;\cdot)$ denote the canonical nonnegative feature angle (Def. 6). Then, for each style plane $i$,*

$$\Theta_i^{\text{dir}}(\tilde{f}(x_B); \tilde{f}(x_A); v^{(i)}) \equiv (c_0^{(i)}\,v^{(i)}) \bmod 2\pi \in [0, 2\pi). \tag{19}$$

*If moreover the no-wrap budget holds, i.e., $|c_0^{(i)}v^{(i)}| < 2\pi$ for all $v^{(i)}$ on the data support, then no wrap occurs and the modulo in equation 19 disappears:*

$$\Theta_i^{\text{dir}}(\tilde{f}(x_B); \tilde{f}(x_A); v^{(i)}) = c_0^{(i)}v^{(i)} \quad \left(\in [0, 2\pi) \text{ if } v^{(i)} \geq 0, \in (-2\pi, 0] \text{ if } v^{(i)} < 0\right). \tag{20}$$

*Consequently, on each style plane $i$ the directed feature angle depends only on $v^{(i)}$ and varies* strictly monotonically *with it, achieving* **strong disentanglement**.

*Proof.* Fix a style plane $i$ and a connected pair

$$x_A = \varphi(z_A) \sim_{\mathcal{A}} x_B = \varphi(z_B), \qquad v := z_B - z_A, \qquad \theta := c_0^{(i)}\,v^{(i)}.$$

By **S1** (equivariance with a linear head) and **S2** (connectivity), the $i$-plane feature rotates by angle $\theta$:

$$\tilde{f}_{\text{sty}}^{(i)}(x_B) = R(\theta)\,\tilde{f}_{\text{sty}}^{(i)}(x_A), \qquad R(\theta) = \begin{pmatrix} \cos\theta & -\sin\theta \\ \sin\theta & \cos\theta \end{pmatrix}. \tag{21}$$

Write $a := \tilde{f}_{\text{sty}}^{(i)}(x_A) = (a_1, a_2)$ and $b := \tilde{f}_{\text{sty}}^{(i)}(x_B) = (b_1, b_2)$. From equation 21, either by direct multiplication or by writing $a = (\cos\alpha, \sin\alpha)$ and $b = (\cos(\alpha + \theta), \sin(\alpha + \theta))$, we obtain

$$d := a \cdot b = \cos\theta, \qquad s := a \times b = \sin\theta. \tag{22}$$

By Definition 6, the principal relative angle is

$$\theta_i^{\text{pr}}(\tilde{f}(x_B); \tilde{f}(x_A)) = \text{atan2}(s, d) = \text{atan2}(\sin\theta, \cos\theta) \in (-\pi, \pi].$$

By the defining property of $\text{atan2}$,

$$\theta_i^{\text{pr}}(\tilde{f}(x_B); \tilde{f}(x_A)) \equiv \theta \pmod{2\pi} \tag{23}$$

By the same definition,

$$\Theta_i^{\text{ccw}}(\tilde{f}(x_B); \tilde{f}(x_A)) = \begin{cases} \theta_i^{\text{pr}}(\tilde{f}(x_B); \tilde{f}(x_A)), & \theta_i^{\text{pr}}(\tilde{f}(x_B); \tilde{f}(x_A)) \geq 0, \\ \theta_i^{\text{pr}}(\tilde{f}(x_B); \tilde{f}(x_A)) + 2\pi, & \theta_i^{\text{pr}}(\tilde{f}(x_B); \tilde{f}(x_A)) < 0, \end{cases} \in [0, 2\pi).$$

Combining with equation 23 yields

$$\Theta_i^{\text{ccw}} \equiv \theta \pmod{2\pi} \tag{24}$$

By Definition 6,

$$\Theta_i^{\text{dir}}(\tilde{f}(x_B); \tilde{f}(x_A); v^{(i)}) = \begin{cases} \Theta_i^{\text{ccw}}(\tilde{f}(x_B); \tilde{f}(x_A)), & v^{(i)} \geq 0, \\ \Theta_i^{\text{ccw}}(\tilde{f}(x_B); \tilde{f}(x_A)) - 2\pi, & v^{(i)} < 0, \end{cases}$$

and equation 24 immediately gives the congruence

$$\Theta_i^{\text{dir}}(\tilde{f}(x_B); \tilde{f}(x_A); v^{(i)}) \equiv \theta = c_0^{(i)}v^{(i)} \pmod{2\pi}, \tag{25}$$

which is precisely equation 19.

Assume $|\theta| = |c_0^{(i)}v^{(i)}| < 2\pi$. Then:

- If $v^{(i)} \geq 0$ (hence $\theta \geq 0$), equation 24 gives $\Theta_i^{\mathrm{ccw}} = \theta \in [0, 2\pi)$, hence $\Theta_i^{\mathrm{dir}} = \Theta_i^{\mathrm{ccw}} = \theta = c_0^{(i)} v^{(i)}$.

- If $v^{(i)} < 0$ (hence $\theta < 0$), equation 24 gives $\Theta_i^{\mathrm{ccw}} = \theta + 2\pi \in (0, 2\pi)$, hence $\Theta_i^{\mathrm{dir}} = \Theta_i^{\mathrm{ccw}} - 2\pi = (\theta + 2\pi) - 2\pi = \theta = c_0^{(i)} v^{(i)} \in (-2\pi, 0]$.

This proves equation 20.

On the no-wrap domain, $\Theta_i^{\mathrm{dir}} = c_0^{(i)} v^{(i)}$ with $c_0^{(i)} > 0$ (by our orientation convention), so $v^{(i)} \mapsto \Theta_i^{\mathrm{dir}}$ is strictly increasing and does not depend on any $v^{(j)}$ with $j \neq i$. This is the stated strong disentanglement If some $c_0^{(i)} < 0$, flip the orientation of the $i$-th style plane (swap its axes), which replaces $c_0^{(i)}$ by $|c_0^{(i)}|$ and preserves all statements. $\square$

Intuition: As the previous lemmas and propositions says, along each style coordinate, changing $i$-th latent $z^{(i)}$ by augmentation always rotates the $i$-th style plane by a fixed slope times that change. In this proposition we just read that angle back from the features themselves: the two feature points lie on the unit circle, so their relative angle is determined by a standard geometric construction (dot/cross products and $\mathrm{atan2}$. If rotations never "wrap around" a full turn, there is no $2\pi$ ambiguity, so the measured feature angle is exactly this slope times augmentation induced displacement $v^{(i)}$.

**Remark 3** (Angle budget). *To enforce the budget $|c_0^{(i)} v^{(i)}| \leq 2\pi$ during training, constrain angles via Equation 13 in the main content and pick a safety factor $\eta$:*

- ***Cyclic Style Latents:*** *set $\eta = 1$ (full branch).*

- ***Non-cyclic Style Latents:*** *use a conservative $\eta < 1$ (e.g. $0.4 \sim 0.7$) to keep angles away from the endpoint and avoid wrap.*

- ***Approximate strength:*** *if $\max_{t \in T_{\mathrm{data}}} \tilde{q}_{a(t)}$ is only an estimate, simply decrease $\eta$. A smaller $\eta$ still yields strong disentanglement (no wrap), at the cost of reduced dynamic range, which is usually acceptable.*

*Intuitively, $\eta$ directly scales the maximum realized rotation. When in doubt, pick a smaller $\eta$; this preserves the equality (no modulo) in equation 20.*

Having established strong disentanglement, we can now read out the latent coordinate on each style plane by a simple geometric post-processing of the feature angles.

On a non-cyclic factor, the feature angle on the circle is already an affine function of the latent, but only defined modulo $2\pi$. For each style subspace, the data only occupies one continuous arc on the circle and there is a single "big gap" where the angles wrap around from the end back to the beginning. The construction in Corollary 2 simply detects this largest gap, takes the midpoint of the covered arc as a new reference direction, and then re-centers all angles around it. With this re-centering, the wrap disappears and the principal angle becomes a true linear function of the latent, up to a shift by the mid-point.

For cyclic factors, one full latent period must correspond to exactly one full $2\pi$ turn on the circle. Strong disentanglement forces the rotation angle to grow linearly with the latent, so "one period $\Rightarrow$ one full turn" immediately pin down the slope. The remaining freedom is just a global in-plane rotation that is fixed per cluster, so within each cluster the feature on that style plane is exactly a phase-shifted sine–cosine trace of the latent variable which preserve the cyclic structure, as shown in Corollary 3.

**Corollary 2** (Principal-angle post-processing when the largest gap is the endpoint wrap gap). *Fix the $i$-th style plane and an $\mathcal{A}$-connected cluster $c_k$ with non-cyclic latent support $z^{(i)} \in [a_k, b_k]$ (the range may depend on the cluster $c_k$). For each $x = \varphi(z)$ let $\tilde{f}_{\mathrm{sty}}^{(i)}(x) = (u_i(x), v_i(x)) \in \mathbb{S}^1$ and*

$$\theta(x) := \mathrm{atan2}\left(v_i(x), u_i(x)\right) \in (-\pi, \pi].$$

*Assume the plane-wise strong disentanglement condition (or, equivalently, weak edge-wise equiv-ariance + no-wrap span) so that there exist $c_0^{(i)} \neq 0$ and $\beta^{(i)} \in \mathbb{R}/2\pi\mathbb{Z}$ with*

$$\theta(x) \equiv c_0^{(i)} z^{(i)} + \beta^{(i)} \pmod{2\pi}. \tag{26}$$

*Let the principal angles be sorted $-\pi < \theta_1 < \cdots < \theta_n \leq \pi$, and define gaps $g_k := \theta_{k+1} - \theta_k$ for $k = 1, \ldots, n-1$ and $g_n := \theta_1 + 2\pi - \theta_n$. Let $k^* = \arg\max_k g_k$ and assume the largest gap $g_{k^*}$ is the wrap-around gap between the endpoint latents $a_k$ and $b_k$ (endpoint wrap-gap).*

**Center of the covered arc (no explicit wrapping).** *Let*

$$\theta_L := \theta_{k^*}, \quad \theta_R := \theta_{k^*+1} \text{ (with } \theta_{n+1} := \theta_1 + 2\pi\text{)},$$
$$e_L = (\cos\theta_L, \sin\theta_L), \quad e_R = (\cos\theta_R, \sin\theta_R), \quad m := e_L + e_R.$$

*Set*

$$\alpha := \begin{cases} \text{atan2}(m_y, m_x), & \text{if } g_{k^*} > \pi \ (L := 2\pi - g_{k^*} < \pi), \\ \text{atan2}(-m_y, -m_x), & \text{if } g_{k^*} \leq \pi \ (L := 2\pi - g_{k^*} \geq \pi), \end{cases} \tag{27}$$

*where $L$ is the length of the covered data arc (the complement of the largest gap). Finally, define the 1-D feature*

$$\widehat{f}^{(i)}(x) := \text{atan2}\big(\sin(\theta(x) - \alpha), \cos(\theta(x) - \alpha)\big) \in (-\pi, \pi]. \tag{28}$$

**Conclusion (mid-latent linearity).** *For all $x = \varphi(z)$,*

$$\widehat{f}^{(i)}(x) = c_0^{(i)}\left(z^{(i)} - \frac{a_k + b_k}{2}\right),$$

*i.e., an exact, continuous affine readout centered at the mid-latent $(a_k + b_k)/2$; if $c_0^{(i)} > 0$ it is strictly increasing, and if $c_0^{(i)} < 0$ it is strictly decreasing.*

*Proof.* For any $\varphi(z_0) = x_0 \sim_{\mathcal{A}} x = \varphi(z)$. Since unit vectors $a = (\cos\alpha, \sin\alpha)$ and $b = (\cos\beta, \sin\beta)$, $\text{atan2}(a \times b, a \cdot b) \equiv \beta - \alpha \pmod{2\pi}$, applying this to $a = \tilde{f}_{\text{sty}}^{(i)}(x_0)$, $b = \tilde{f}_{\text{sty}}^{(i)}(x)$ gives

$$\theta(x) - \theta(x_0) \equiv \theta_i^{\text{pr}}\big(\tilde{f}(x); \tilde{f}(x_0)\big) \pmod{2\pi}.$$

Since $\theta_i^{\text{dir}} \equiv \theta_i^{\text{pr}} \pmod{2\pi}$ by definition of $\theta_i^{\text{dir}}$, strong disentanglement yields for every connected pair:

$$\theta(x) - \theta(x_0) \equiv c_0^{(i)}\big(z^{(i)} - z_0^{(i)}\big) \pmod{2\pi}.$$

Set $\beta^{(i)} := \theta(x_0) - c_0^{(i)} z^{(i)}(x_0) \in \mathbb{R}/2\pi\mathbb{Z}$ to obtain equation 26.

*(1) Largest gap $\Rightarrow$ covered arc and its endpoints.* By construction the $n$ gaps $\{g_k\}$ partition the circle and sum to $2\pi$. Removing the *largest* gap $g_{k^*}$ leaves the *shortest* circular arc that covers all data; its endpoints are precisely the two unit directions with principal angles $\theta_R$ and $\theta_L$ adjacent across $g_{k^*}$. By the endpoint wrap-gap assumption, these endpoints are the samples at the latent endpoints $z^{(i)} = a$ and $z^{(i)} = b$, so the covered-arc length is $L = 2\pi - g_{k^*} \in (0, 2\pi)$.

*(2) Center via the two endpoint vectors.* For two unit vectors at angles $\phi_1, \phi_2$, the vector sum has direction

$$\arg(e^{i\phi_1} + e^{i\phi_2}) = \frac{\phi_1 + \phi_2}{2}$$

and points to the midpoint of the *shorter* of the two arcs between $\phi_1$ and $\phi_2$. In our notation, $m = e_L + e_R$ points to the midpoint of the shorter arc between $\theta_L$ and $\theta_R$. If $g_{k^*} > \pi$ then the *gap* is longer than the data arc ($L < \pi$), so the shorter arc is the data arc and $\arg(m)$ is already the covered-arc midpoint. If $g_{k^*} \leq \pi$ then the gap is the shorter arc (and the data arc has length $L \geq \pi$), so the covered-arc midpoint is the *antipode* of $\arg(m)$; this is achieved by $\arg(-m)$. This is exactly the definition equation 27 of $\alpha$, which lies in $(-\pi, \pi]$ by the properties of $\text{atan2}$.

*(3) Signed offsets about $\alpha$ are single-branch differences.* All principal angles $\theta(x)$ lie in the covered arc of length $L < 2\pi$ centered at $\alpha$, hence for every sample $|\theta(x) - \alpha| \leq L/2 < \pi$. Therefore, the signed principal difference computed by equation 28 equals the ordinary difference:

$$\widehat{f}^{(i)}(x) = \theta(x) - \alpha \qquad \forall x.$$

*(4) Mid-latent identity for $\alpha$ and exact linearity.* By the modular–affine law equation 26, there exist integers $m_a, m_b$ such that the *unwrapped* endpoint angles along the covered arc are

$$\phi(a_k) = c_0^{(i)} a_k + \beta^{(i)} + 2\pi m_a, \qquad \phi(b_k) = c_0^{(i)} b_k + \beta^{(i)} + 2\pi m_b,$$

with $\phi(b) - \phi(a) = L \in (0, 2\pi)$. The covered-arc midpoint is then

$$\alpha \equiv \frac{\phi(a_k) + \phi(b_k)}{2} \equiv c_0^{(i)} \frac{a_k + b_k}{2} + \beta^{(i)} \pmod{2\pi}.$$

Because every $\theta(x)$ lies on the single branch centered at $\alpha$ (Step 3), the congruence equation 26 lifts to the equality $\theta(x) = c_0^{(i)} z^{(i)} + \beta^{(i)}$ on this branch. Subtracting $\alpha \equiv c_0^{(i)} \frac{a_k + b_k}{2} + \beta^{(i)}$ yields

$$\widehat{f}^{(i)}(x) = \theta(x) - \alpha = c_0^{(i)} \left( z^{(i)} - \frac{a_k + b_k}{2} \right).$$

Continuity follows since no sample sits on the cut, and Step 3 used signed principal differences.

$\square$

**Corollary 3** (Single-period cyclic factor determines the slope (clusterwise phase)). *Fix a style plane $j$ and an $\mathcal{A}$-connected cluster $c_k$ whose latent support on this plane is intrinsically cyclic with period $p > 0$. Suppose there exists an interval $I = [a, a + p)$ (a single latent period) across which $z^{(j)}$ varies on the data support of $c_k$. Assume the plane-wise strong disentanglement. Then:*

$$\left| c_0^{(j)} \right| = \frac{2\pi}{p}, \quad and \quad \exists\, A_{j,k} \in SO(2) \text{ (independent of } z) \text{ s.t. for all } z^{(j)} \in I,$$

$$\tilde{f}_{\text{sty}}^{(j)}(\varphi(z)) = A_{j,k} \begin{pmatrix} \cos\left(\frac{2\pi}{p} z^{(j)}\right) \\ \sin\left(\frac{2\pi}{p} z^{(j)}\right) \end{pmatrix}.$$

*In particular, within each cluster $c_k$ the representation on plane $j$ is exactly periodic with period $p$ and linear in $z^{(j)}$ modulo a clusterwise fixed in-plane rotation $A_{j,k}$.*

*Proof.* Fix a cluster $k$, pick any $z_0 \sim_{\mathcal{A}} z$ and by the perfect equivariance condition,

$$\tilde{f}_{\text{sty}}^{(j)}(\varphi(z)) = R(c_0^{(j)}(z^{(j)} - z_0^{(j)}))\, \tilde{f}_{\text{sty}}^{(j)}(\varphi(z_0)), \qquad \forall z \in \mathcal{Z}_{\text{train}}. \tag{29}$$

As $z^{(j)}$ moves from $a$ to $a + p$, the feature's rotation angle changes by $c_0^{(j)}\big((a + p) - a\big) = c_0^{(j)} p$. Strong disentanglement makes $\tilde{f}_{\text{sty}}^{(j)}(\varphi(z))$ traverses the circle exactly once, forcing $|c_0^{(j)} p| = 2\pi$, hence $|c_0^{(j)}| = 2\pi/p$.

Let $\phi_0 \in \mathbb{R}$ be the phase of $\tilde{f}_{\text{sty}}^{(j)}(\varphi(z_0))$, i.e. $\tilde{f}_{\text{sty}}^{(j)}(\varphi(z_0)) = (\cos\phi_0, \sin\phi_0)$. Plug $c_0^{(j)}$ in equation 29:

$$\tilde{f}_{\text{sty}}^{(j)}(\varphi(z)) = R\Big(\tfrac{2\pi}{p}(z^{(j)} - z_0^{(j)})\Big) R(\phi_0)\, e_1 = R\Big(\underbrace{\tfrac{2\pi}{p} z^{(j)} + \phi_0 - \tfrac{2\pi}{p} z_0}_{=:\theta_j}\Big) e_1,$$

where we used $R(\alpha)R(\beta) = R(\alpha + \beta)$ and commutativity of planar rotations.

*Claim.* $\theta_j$ is independent of the choice of $z_0$.

Pick another reference $z_1 \sim_{\mathcal{A}} z$. Using the equivariance condition, we have

$$\tilde{f}_{\text{sty}}^{(j)}(\varphi(z_1)) = R\Big(\tfrac{2\pi}{p}(z_1^{(j)} - z_0^{(j)})\Big) \tilde{f}_{\text{sty}}^{(j)}(\varphi(z_0))$$

Writing $\tilde{f}_{\text{sty}}^{(j)}(\varphi(z_1)) = (\cos\phi_k, \sin\phi_k)$, this implies $\phi_1 \equiv \phi_0 + \frac{2\pi}{p}(z_1^{(j)} - z_0^{(j)}) \pmod{2\pi}$. Therefore

$$\phi_1 - \tfrac{2\pi}{p} z_1^{(j)} \equiv \big(\phi_0 + \tfrac{2\pi}{p}(z_1^{(j)} - z_0^{(j)})\big) - \tfrac{2\pi}{p} z_1^{(j)} = \phi_0 - \tfrac{2\pi}{p} z_0^{(j)} \pmod{2\pi},$$

and since $R(\cdot)$ is $2\pi$-periodic, $R(\phi_1 - \frac{2\pi}{p} z_1) = R(\phi_0 - \frac{2\pi}{p} z_0)$. Hence, prove the claim. Define the fixed rotation

$$A_{j,k} := R(\theta_j) \in SO(2).$$

Then

$$\tilde{f}_{\text{sty}}^{(j)}(\varphi(z)) \;=\; A_{j,k}\, R\!\left(\tfrac{2\pi}{p} z^{(j)}\right) e_1 \;=\; A_{j,k}\begin{pmatrix}\cos\!\left(\tfrac{2\pi}{p} z^{(j)}\right)\\ \sin\!\left(\tfrac{2\pi}{p} z^{(j)}\right)\end{pmatrix}, \qquad \forall z \in \mathcal{Z}_{\text{train}}, \qquad (30)$$

which is the claimed representation with plane rotation $A_{j,k}$ independent of reference $z_0$. $\qquad\square$

**Remark 4** (Global alignment after post-processing). ***Non-cyclic case.*** *If for plane $i$ all $\mathcal{A}$-connected clusters share the same midpoint $(a_i + b_i)/2$, then the principal-angle post-processing of Cor. 2 yields*

$$\widehat{f}^{(i)}(x) = c_0^{(i)}\!\left(z^{(i)} - \tfrac{a_i+b_i}{2}\right) \quad \text{for every cluster,}$$

*so the 1-D readout is* already aligned globally *across clusters.*

***Cyclic case.*** *For intrinsically cyclic planes, there is no lossless 1-D readout; features remain 2-D on $S^1$ and differ across clusters by a constant phase $A_{j,k} \in SO(2)$. Consequently, global comparison requires a* clusterwise *phase alignment (e.g., rotate each cluster by $A_{j,k}^{-1}$ to a chosen reference gauge); after this rotation, all clusters live in the same 2-D gauge up to one shared global in-plane rotation.*

In the last two propositions, we only prove the continuous case, which can be easily extended to the discrete case.

### C.5 Proof of Proposition 4

**Proposition 8** (Axis-separable linearity extension). *Under the assumption of Corollary 1. Moreover, except single-style augmentations $\mathcal{A}_{\text{single}} = \{a_i(t_i)\}_{i=1}^{d_{\text{sty}}}$, there are $a_{\text{as}}(t) \in \mathcal{A} \setminus \mathcal{A}_{\text{single}}$ such that:*

$$a_{\text{as}}(t_{\text{as}}, \cdot) = a_{i_m}(t_m) \circ \cdots \circ a_{i_1}(t_1, \cdot),$$

*(Here "$\circ$" means usual function composition: $(a \circ b)(x) = a(b(x))$.)*

*Suppose for all $t_{\text{as}} \in \mathcal{T}_{\text{as}}$ there exists $z_{\text{as}}$ and a finite chain*

$$z_0 := z_{\text{as}}, \; z_1, \ldots, z_{n-1} \in \mathcal{Z}_{\text{train}}, \quad z_n := a_{\text{as}}(t_{\text{as}}, z_{\text{as}}) \; (z_n \text{ not required in } \mathcal{Z}_{\text{train}}),$$

*such that for each edge $(z_j, z_{j+1})$ with $j = 0, \ldots, n-2$, there exist augmentations $u_j^+(t_j^+), u_j^-(t_j^-)$ such that*

$$\varphi^{-1}\!\big(u_j^+(t_j^+, \varphi(z_j))\big) = \varphi^{-1}\!\big(u_j^-(t_j^-, \varphi(z_{j+1}))\big) =: z_j^\star \in \mathcal{Z},$$

*realized displacements $m_j^+ := u_j^+(t_j^+), \; m_j^- := u_j^-(t_j^-) \in \mathbb{R}^{d_{\text{sty}}}$ satisfying $m_j^{+,(i)}, m_j^{-,(i)} \in [-M_i, M_i]$ for all $i$. For the last edge $(z_{n-1}, z_n)$, there exists a one-sided augmentation $u_{n-1}^+(t_{n-1}^+)$ with*

$$\varphi(z_n) = u_{n-1}^+\!\big(t_{n-1}^+, \varphi(z_{n-1})\big), \qquad m_{n-1}^+ := u_{n-1}^+(t_{n-1}^+), \; m_{n-1}^{+,(i)} \in [-M_i, M_i].$$

*Then $a_{\text{as}}(t_{\text{as}})$ is axis-separable with net latent displacement*

$$v := \sum_{j=0}^{n-2}\big(m_j^- - m_j^+\big) \;+\; m_{n-1}^+$$

*and $g_{\text{as}} : \mathcal{T}_{\text{as}} \to [-\pi, \pi]^{d_{\text{sty}}}$ is linear in the per-axis displacement:*

$$g_{\text{as}}(t_{\text{as}}) \;\equiv\; c_0 \odot v \quad (\text{mod } 2\pi \text{ per axis})$$

*If in addition, $v^{(i)} \in [-M_i, M_i]$, then*

$$g_{\text{as}}(t_{\text{as}}) \;=\; c_0 \odot v$$

*Proof.* Write $x_j := \varphi(z_j)$. For each interior edge, perfect equivariance on the two routes to $z_j^\star$ gives $\mathcal{R}(c_0 \odot m_j^+)\tilde{f}(x_j) = \tilde{f}(x_j^\star) = \mathcal{R}(c_0 \odot m_j^-)\tilde{f}(x_{j+1})$, hence $\tilde{f}(x_{j+1}) = \mathcal{R}_{c_0 \odot (m_j^- - m_j^+)}\tilde{f}(x_j)$.

For the terminal edge, one-sided equivariance yields $\tilde{f}(x_n) = \mathcal{R}_{c_0 \odot m_{n-1}^+} \tilde{f}(x_{n-1})$. Multiplying all steps and using those rotations on distinct planes commute and add in-plane,

$$\tilde{f}(x_n) = \mathcal{R}(c_0 \odot (\sum_{j=0}^{n-2}(m_j^- - m_j^+) + m_{n-1}^+)) \tilde{f}(x_0) = \mathcal{R}_{c_0 \odot v} \tilde{f}(x_0),$$

Since the output of $g_{\mathrm{as}}$ is bounded in $[-\pi, \pi]$, we can only have $g_{\mathrm{as}}(t_{\mathrm{as}}) \equiv c_0 \odot v$ (mod $2\pi$ per axis).

If in addition, we have $v^{(i)} \in [-M_i, M_i]$, since each $g_i$ output with bound $[-\pi, \pi]$, we can have $g_{\mathrm{as}}(t_{\mathrm{as}}) = c_0 \odot v$.

$\square$

### C.6   PROOF OF PROPOSITION 5

**Proposition 9** (Few–latent anchor, gated composite head, and linearity)**.** *Let* $\mathcal{A}_{\mathrm{single}} = \{a_i(t_i)\}_{i=1}^{d_{\mathrm{sty}}-1}$ *be single–style augmentations with $z$–independent per–axis displacements* $\Delta_{a_i}(t_i, z) = q_{a_i}(t_i) e_i$ *and* $|q_{a_i}(t_i)| \leq M_i$. *Let* $a_{\mathrm{cmp}}(t) \in \mathcal{A} \setminus \mathcal{A}_{\mathrm{single}}$ *be a (possibly) multi–style augmentation.*

*Assume:*

> *(A1)* **Pure at an anchor fiber.** *There is $J \subseteq \{1, \ldots, d_{\mathrm{sty}}-1\}$ and $\bar{z}_J$ such that on the fiber* $\mathsf{F}_J(\bar{z}_J) := \{z : \pi_J(z) = \bar{z}_J\}$,
>
> $$\Delta_{a_{\mathrm{cmp}}}(t, z) = q_{a_{\mathrm{cmp}}}(t) e_{d_{\mathrm{sty}}}, \qquad |q_{a_{\mathrm{cmp}}}(t)| \leq M_{d_{\mathrm{sty}}}.$$

> *(A2)* **Axis–definable off the anchor.** *For any $(t, z)$ there exists an $\mathcal{A}$–chain (as in Lemma 8) from $z$ to $a_{\mathrm{cmp}}(t, z)$ comprised of realized steps $m_j^{\pm}$ (each coordinatewise bounded by $M_i$) with net*
>
> $$v(t, z) := \sum_{j=0}^{n-2}(m_j^- - m_j^+) + m_{n-1}^+ \in \mathbb{R}^{d_{\mathrm{sty}}}.$$

**Gated composite head.**   *Define the selector $S : \mathbb{R}^{|J|} \to \{0, 1\}^{d_{\mathrm{sty}}}$ by*

$$S(\pi_J(z)) = \begin{cases} e_{d_{\mathrm{sty}}}, & \pi_J(z) = \bar{z}_J \quad \text{(on the anchor fiber)}, \\ \mathbf{1}_{d_{\mathrm{sty}}}, & \pi_J(z) \neq \bar{z}_J \quad \text{(off fiber)}, \end{cases}$$

*and take*

$$g_{\mathrm{cmp}} : \mathbb{R}^{|J|} \times \mathcal{T}_{a_{\mathrm{cmp}}} \to [-\pi, \pi]^{d_{\mathrm{sty}}}, \qquad g_{\mathrm{cmp}}(\pi_J(z), t) \in S(\pi_J(z)) \odot [-\pi, \pi]^{d_{\mathrm{sty}}}.$$

**Conclusions.**   *Assume perfect equivariance (Lemma 2), nondegeneracy $g_{\mathrm{cmp}} \not\equiv 0$ on non-identity displacement, and Cor. 1 for single–axis and pure–located cases. Then:*

> *(C1)* **On the anchor (one channel open).** *For any $z \in \mathsf{F}_J(\bar{z}_J)$ there exists $c_0^{(d_{\mathrm{sty}})} \neq 0$ s.t.*
>
> $$g_{\mathrm{cmp}}(\bar{z}_J, t) = c_0^{(d_{\mathrm{sty}})} q_{a_{\mathrm{cmp}}}(t).$$

> *(C2)* **Off the anchor (all channels open).** *For any $z \notin \mathsf{F}_J(\bar{z}_J)$,*
>
> $$g_{\mathrm{cmp}}(\pi_J(z), t) \equiv c_0 \odot v(t, z)$$
>
> *for some constant slope vector $c_0 \in \mathbb{R}^{d_{\mathrm{sty}}}$ determined by the single–axis calibrations.*

*Proof. Step 1: Equivariance to in–plane rotations.* By perfect equivariance, for any augmentation that effects a net latent displacement $w \in \mathbb{R}^{d_{\mathrm{sty}}}$, the aligned style features transform as

$$\tilde{f}(a(\cdot, \varphi(z))) = \mathcal{R}(\theta(w)) \tilde{f}(\varphi(z)), \quad \text{with} \quad \theta(w) = c_0 \odot w,$$

where $c_0$ is the (cluster/plane–wise) slope vector supplied by Cor. 1 from the single–axis identifications.

*Step 2: Anchor purity $\Rightarrow$ scalar linearity.* On $\mathsf{F}_J(\bar{z}_J)$, assumption (A1) gives $\Delta_{a_{\mathrm{cmp}}}(t, z) = q_{a_{\mathrm{cmp}}}(t)e_{d_{\mathrm{sty}}}$. Since the selector $S(\bar{z}_J) = e_{d_{\mathrm{sty}}}$ opens only channel $d_{\mathrm{sty}}$, $g_{\mathrm{cmp}}$ depends only on $t$ but not on other coordinates. By the single–axis linearity (Cor. 1) applied on the $d_{\mathrm{sty}}$–plane,

$$g_{\mathrm{cmp}}(\bar{z}_J, t) = c_0^{(d_{\mathrm{sty}})} q_{a_{\mathrm{cmp}}}(t),$$

with $c_0^{(d_{\mathrm{sty}})} \neq 0$ because $g_{\mathrm{cmp}}$ is nontrivial on non-identity displacements. This proves (C1).

*Step 3: Axis–definability off fiber $\Rightarrow$ vector linearity.* Off the anchor, (A2) provides an $\mathcal{A}$–chain whose realized net is $v(t, z)$. Equivariance composes along the chain (Lemma 8): each single–axis step contributes additively to the angle in its plane, so the total in–plane rotation equals $\theta\big(v(t, z)\big) = c_0 \odot v(t, z)$. Because $S(\pi_J(z)) = \mathbf{1}_{d_{\mathrm{sty}}}$, the head exposes all coordinates and thus reads out

$$g_{\mathrm{cmp}}\big(\pi_J(z), t\big) \equiv c_0 \odot v(t, z).$$

This yields (C2).

*Step 4: No–wrap regime under calibration.* If each coordinate is calibrated so that no coordinate wraps and the equality holds in $\mathbb{R}^{d_{\mathrm{sty}}}$, $g_{\mathrm{cmp}} = c_0 \odot v(t, z)$. $\qquad\square$

**Remark 5** (Training with "pure indicators", *fixed* gating code). *We do not observe latent $z$. We mark a subset $\mathcal{X}_{\mathrm{pure}} \subset \mathcal{X}_{\mathrm{train}}$ on which $a_{\mathrm{cmp}}$ is empirically pure on the $d_{\mathrm{sty}}$–plane and assign a fixed code to all such points:*

$$h(x) \equiv \mathbf{0} \in \mathbb{R}^{|J|} \quad \text{for all } x \in \mathcal{X}_{\mathrm{pure}} \qquad \text{(no learning of $h$)}.$$

*The gate uses $h(x)$ in place of $\pi_J(z)$: on $\mathcal{X}_{\mathrm{pure}}$, set $S(h(x)) = e_{d_{\mathrm{sty}}}$ (only channel $d_{\mathrm{sty}}$ open); off that set, open all channels, e.g. $S(h(x)) = \mathbf{1}_{d_{\mathrm{sty}}}$.*

*A practical loss is*

$$\mathcal{L}_{\mathrm{gate}} := \mathbb{E}_{x \in \mathcal{X}_{\mathrm{pure}}, t} \sum_{i \neq d_{\mathrm{sty}}} \big(g_{\mathrm{cmp}}^{(i)}(h(x), t)\big)^2,$$

*which closes all non–$d_{\mathrm{sty}}$ channels on pure points; off–fiber behavior is learned via the $\mathcal{A}$–chain supervision.*

**Why fixing $h$ on pure points is necessary.** *If $h(x)$ were allowed to vary across $x \in \mathcal{X}_{\mathrm{pure}}$, then equivariance alone would not guarantee a single angle $\theta^{(d_{\mathrm{sty}})}$ for the same augmentation $t$ across pure samples: the head could implement a sample–dependent reparametrization $\theta^{(d_{\mathrm{sty}})}(t, h(x))$, still satisfying equivariance but breaking identifiability of the common slope on the $d_{\mathrm{sty}}$–plane. Fixing $h(x) \equiv \mathbf{0}$ eliminates this degree of freedom and enforces a unique readout for the same augmentation.*

---

**Algorithm 1** Cluster-Dependent Rotational Equivariance (with post-processing)

---

1: **Input** Dataset $\{x\}$, augmentations $\mathcal{A} = \{a_i\}_{i=1}^m$, number of clusters $k$, transform nets $G = \{g_i\}_{i=1}^m$, feature extractor $f$, hyperparams $\{\lambda_{\text{theta}}, \lambda_{\text{radius}}\}$, subspace radius $\omega$, epochs $T_1, T_2$
    **Stage 1 (optional contrastive pretrain; only if $k > 1$)**
2: **for** $t = 1$ to $T_1$ **do**
3:     Sample $\{(x_j, x_j^+ = a(x_j))\}$ with $a \in \mathcal{A}$
4:     Compute $\mathcal{L}_{\text{InfoNCE}}$
5:     Update $f$ by $\nabla \mathcal{L}_{\text{InfoNCE}}$
6: **end for**
    **Stage 2 (Cluster-Dependent Rotational Equivariance)**
7: Initialize centers $\{c_\ell\}$ by spherical $k$-means on $u(x) := f(x)/\|f(x)\|_2$;
8: Set $r_\ell \leftarrow \text{normalize}(c_\ell)$; Constrain range of $g_i$ by Eq. (13)
9: **for** $t = 1$ to $T_2$ **do**
10:     Sample $\{(x_j, x_j^+ = a(t, x_j))\}$ with $a(t) \in \mathcal{A}$
11:     $u_j \leftarrow f(x_j)/\|f(x_j)\|_2, \quad u_j^+ \leftarrow f(x_j^+)/\|f(x_j^+)\|_2$
12:     Assign $c(x_j) = \arg\max_i u_j^\top r_i$ (same for $x_j^+$)
13:     Build Householder $H_{c(x)}$ that maps the rotation center $r_{c(x)}$ pole to the north pole
14:     $\theta \leftarrow G(t)$
15:     Build block-diag matrix $R(\theta)$
16:     Compute $\mathcal{L}_{\text{equiv}}, \mathcal{L}_{\text{radius}}, \mathcal{L}_{\text{theta}}$
17:     Update $f, \{g_i\}$ by $\nabla(\mathcal{L}_{\text{equiv}} + \lambda_{\text{radius}}\mathcal{L}_{\text{radius}} + \lambda_{\text{theta}}\mathcal{L}_{\text{theta}})$
18: **end for**
    **Optional: post-processing and final representation**
19: **for** each cluster $i$ and style plane $j$ **do**
20:     Collect $\theta(x) \leftarrow \text{atan2}\big([H_i f(x)]_{j,y}, [H_i f(x)]_{j,x}\big)$ for $x \in S_i$
21:     Sort $\{\theta(x)\}$ to $\theta_1 < \cdots < \theta_n$; $g_k \leftarrow \theta_{k+1} - \theta_k$ ($k = 1{:}n-1$), $g_n \leftarrow \theta_1 + 2\pi - \theta_n$
22:     $k^* \leftarrow \arg\max_k g_k$; $\theta_L \leftarrow \theta_{k^*}, \theta_R \leftarrow \theta_{k^*+1}$ (wrap)
23:     $m \leftarrow (\cos\theta_L, \sin\theta_L) + (\cos\theta_R, \sin\theta_R)$; $L \leftarrow 2\pi - g_{k^*}$
24:     $\alpha_{i,j} \leftarrow \begin{cases} \text{atan2}(m_y, m_x), & L < \pi \\ \text{atan2}(-m_y, -m_x), & L \geq \pi \end{cases}$
25: **end for**
26: **for** each $x$ with cluster $i = c(x)$ **do**
27:     **for** each plane $j$ **do**
28:       **if** plane $j$ is non-cyclic **then**
29:         $\widehat{f}^{(j)}(x) \leftarrow \text{atan2}\big(\sin(\theta^{(j)}(x) - \alpha_{i,j}), \cos(\theta^{(j)}(x) - \alpha_{i,j})\big)$   (scalar)
30:       **else**                           ▷ cyclic: keep 2-D, optionally phase-align by $A_{i,j}$
31:         $\widetilde{f}^{(j)}(x) \leftarrow A_{i,j}\,[H_i f(x)]_j \in \mathbb{R}^2$   (with $A_{i,j}$ fixed per cluster/plane)
32:       **end if**
33:     **end for**
34:     $\widehat{f}_{\text{style}}(x) \leftarrow \text{concat}\big(\{\widehat{f}^{(j)}(x)\}_{j \in \text{non-cyc}}, \{\widetilde{f}^{(j)}(x)\}_{j \in \text{cyc}}\big)$
35:     **Final rep:** $\widehat{r}(x) \leftarrow \text{concat}\big(\widehat{f}_{\text{style}}(x), e_i\big)$     (append one-hot cluster code $e_i \in \{0,1\}^k$)
36: **end for**

---

Table 2: Parameter ranges used in our setup.

| Parameter | Minimum value | Maximum value |
|---|---|---|
| Object position X | $-0.3$ | $0.3$ |
| Object position Y | $-0.3$ | $0.3$ |
| Object position Z | $-0.3$ | $0.3$ |
| Object rotation $\psi$ | $-0.5$ | $0.5$ |
| Floor hue | $0.3$ | $0.7$ |

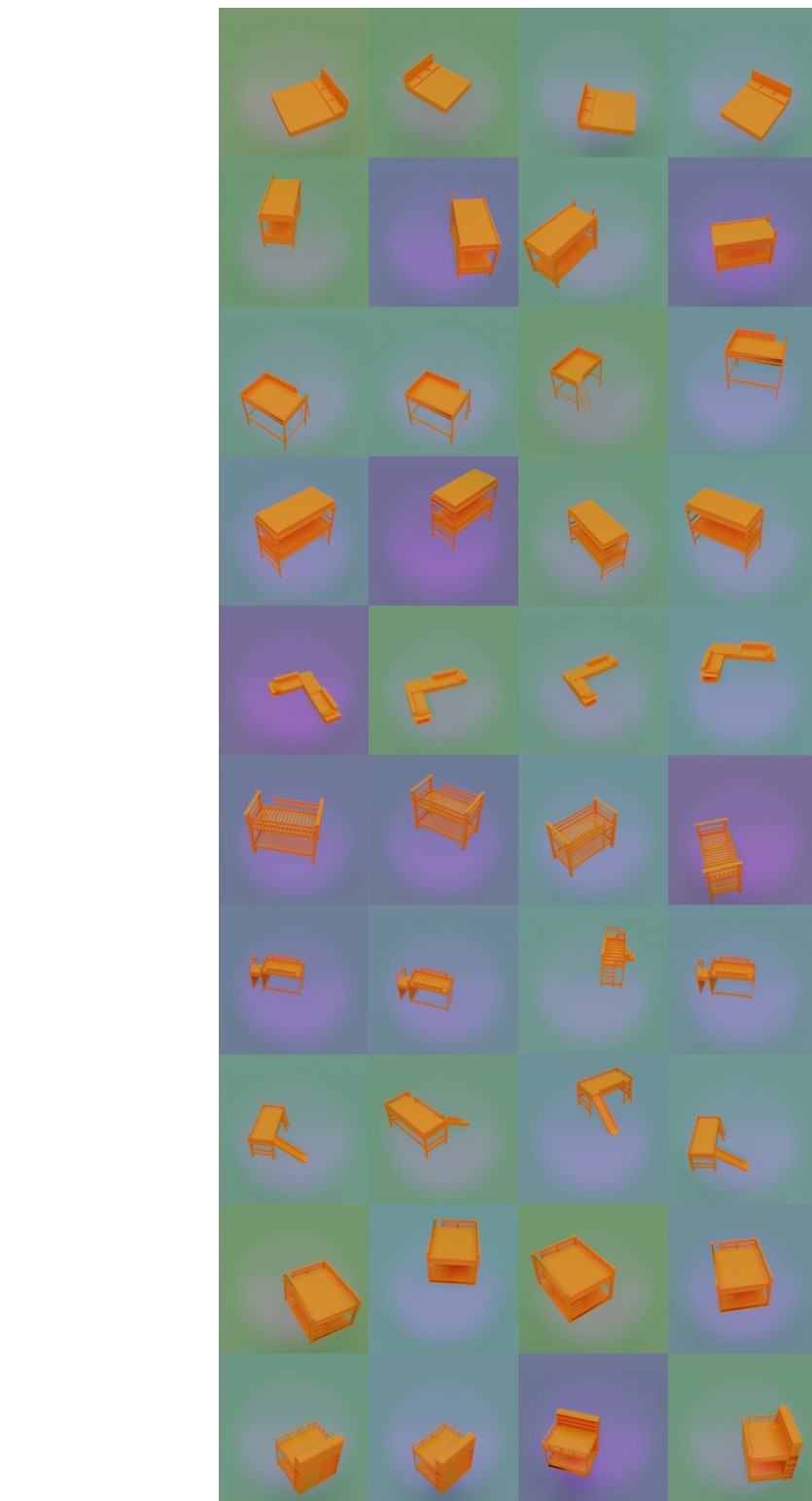

Figure 5: Samples from created 3DIEBench.

# D EXPERIMENT DETAILS

## D.1 DATASET

**Shape3D (Kim & Mnih, 2018)** This dataset provides 5 style latents (floor hue, wall hue, object hue, object scale, azimuth) and 4 object shapes as the class latent. For each style latent, we keep the middle half of its range and include all 4 classes, yielding a total of 16,000 images.

**MPI3D (Gondal et al., 2019)** This dataset contains 6 style latents (object color, object size, camera height, background color, horizontal rotation, vertical rotation) and 6 object shapes as the class latent. We keep the first four style latents in full. For the two rotations, we subsample the angles to $\{0, 4, 8, 12\}$ (uniformly spaced indices), resulting in a total of 10,384 images.

**3DIdent (Zimmermann et al., 2021)** This dataset has a single class and 10 continuous style factors: position $(x, y, z)$; rotation $(\phi, \theta, \psi)$; spotlight position; object hue; ground hue; and spotlight hue. We randomly sample 50,000 images for training.

**3DIEBench (Garrido et al., 2023)** 3DIEBench is derived from a ShapeNetCore subset (sourced from 3D Warehouse). To generate the non-symmetric images, we use 10 chair models as classes and vary 5 parameters using Blender to render images uniformly over their ranges (times $\pi$ in the generating object rotation $\psi$). We generate 20,000 images at a resolution of $224 * 224$. Parameter ranges are listed in Table 2, and example renderings are shown in Figure 5.

**Synthetic dataset** The synthetic dataset used in Section 4.3 is designed so that we can precisely control the underlying factors of variation and directly verify our theoretical results. It consists of $96 \times 96$ grayscale images of three distinct uppercase letters, "R," "E," and "D." For each letter, we generate samples by applying three controlled augmentations: horizontal translation (x-shift), vertical translation (y-shift), and in-plane rotation about the image center.

## D.2 HARDWARE

All experiments were performed on 2 NVIDIA Tesla V100 GPUs with 32GB accelerator RAM for a single training run. All experiments use the PyTorch deep learning framework (Paszke et al., 2019).

## D.3 TRAINING

Unless otherwise stated, all methods are trained for 500 epochs with a ResNet–18 backbone and a three-layer MLP projection head (hidden sizes 2048–2048–2048, PReLU activations). We use a batch size of 500 and Adam ($\mathrm{lr} = 10^{-4}$, $\beta_1 = 0.9$, $\beta_2 = 0.999$). Feature dimension is set to be $2 * \#$ latents $+1$, where 1 is the extra dimension as stated in main content. Method–specific hyperparameters are detailed below.

**SimCLR (Chen et al., 2020a).** We set the InfoNCE temperature to $\tau = 0.5$ for all experiments.

**CARE (Gupta et al., 2023).** We use $\tau = 0.5$ in the InfoNCE loss and $\lambda_{\mathrm{equi}} = 10$. The number of equivariance chunks is set to half the batch size ($B/2$).

**EquiMod (Devillers & Lefort, 2022).** We use $\tau = 0.5$ for InfoNCE loss and $\lambda_{\mathrm{equimod}} = 1$. The predictor head mirrors the projection head architecture.

**IP-IRM (Wang et al., 2021).** We use $\tau = 0.5$ for InfoNCE loss and $\lambda_1 = \lambda_2 = 0.2$ and change each step per 30 epochs. For the updating of partitions, the training epochs are set to be 50.

**Eastwood et al. (2023).** We use $\tau = 0.5$ for all the losses and for each subspaces we set the dimension $= 2$.

**Ours.** We adopt a two-stage schedule. Stage 1: SimCLR pretraining for 100 epochs with $\tau = 0.5$ (skipped on **3DIdent**). Stage 2: we train the transformation networks $G = \{g_i\}_i$ as three–layer MLPs (hidden sizes 2048–2048–2048, PReLU + LayerNorm) for the remaining 400 epochs (or 500 epochs on **3DIdent** when Stage 1 is skipped). Subspace radius $\omega$ is set to be 0.05 and extra dimension is set to be 1 for all the datasets. We simply set loss weights $\lambda_{\mathrm{radius}} = \lambda_{\mathrm{theta}} = 1$ for all four datasets.

## D.4 Evaluation

We evaluate with **DCI** (Eastwood & Williams, 2018), which summarizes three properties of a representation: *disentanglement*, *completeness*, and *informativeness*. Following Eastwood & Williams (2018), we train a supervised predictor from the learned features to the ground-truth factors and extract an *importance matrix* $R \in \mathbb{R}^{d \times k}_{\geq 0}$, where $R_{ij}$ measures the contribution of feature dimension $i$ to predicting factor $j$. In our implementation, we use either a Linear regressor or a Random Forest regressor: for Linear, we take $R_{ij} = |w_{ij}|$ (absolute regression weights), and for Random Forest, we use the mean decrease in impurity (Gini importance) (Breiman, 2001).

Let $\tilde{R}$ be $R$ normalized to probabilities: row-wise for disentanglement and column-wise for completeness. Disentanglement and completeness are then computed as one minus the normalized entropy of these distributions and aggregated across dimensions/factors with standard DCI weighting; informativeness is the prediction performance of the supervised model (we report $R^2$ / error as appropriate), exactly as in Eastwood & Williams (2018).

Because our post-processed features include a one-hot class indicator, we treat that block as a *single* semantic factor. Before normalization, we collapse the corresponding columns by summing their importances, so that the class indicator contributes as one factor in $R$.

## E Ablation

We ablate both the training protocol and the loss terms. (1) Without the SimCLR pretraining stage—i.e., if we initialize rotation centers by running $k$-means on randomly initialized features—the learned features degrade markedly. (2) Even with proper SimCLR pretraining, removing either the radius penalty $\mathcal{L}_{\mathrm{radius}}$ or the angle penalty $\mathcal{L}_{\mathrm{theta}}$ leads to feature collapse (features concentrate to a single point on the sphere). Quantitative and qualitative evidence is provided in Table 3 and Figure 6.

Table 3: Per-dimension Informativeness using linear regression on **Shapes3D**. Columns list ground-truth factors

| Method | Floor Hue | Wall Hue | Object Hue | Scale | Orientation | Class |
|---|---|---|---|---|---|---|
| w/o pretraining | 0.0648 | 0.2712 | 0.0177 | 0.0172 | 0.0010 | 0.0034 |
| w/o $\mathcal{L}_{\mathrm{radius}}$ | 0.0000 | 0.0000 | 0.0000 | 0.0000 | 0.0000 | 1.0000 |
| w/o $\mathcal{L}_{\mathrm{theta}}$ | 0.0000 | 0.0000 | 0.0000 | 0.0000 | 0.0000 | 1.0000 |
| CD-RED (after post-proc.) | 0.9995 | 0.9995 | 0.9994 | 0.9984 | 0.9994 | 1.0000 |

**Uniform rotation centers (optional).** To ensure maximally separated clusters, we may further add a repulsion term over rotation centers

$$\mathcal{L}_{\mathrm{uniform}} = -\mathbb{E}_{i \neq j} \left\| r_i - r_j \right\|_2^2, \tag{31}$$

which, when minimized jointly with our main losses, promotes a uniform configuration on the sphere. So each rotation center is initialized as the cluster centers right after the clustering. This stabilizes updates while allowing $\mathcal{L}_{\mathrm{uniform}}$ to shape inter–center geometry.

Under the same settings as Figure 4, we report the minimum pairwise center distance $d_{\min} = \min_{i \neq j} \left\| r_i - r_j \right\|_2$. With learnable rotation centers (i.e., $\mathcal{L}_{\mathrm{uniform}}$ on), the configuration approaches the Tammes–optimal spacing: for $k = 3$,

$$d_{\min} = \sqrt{2 \left( 1 + \frac{1}{k-1} \right)} = \sqrt{3} \approx 1.732, \tag{32}$$

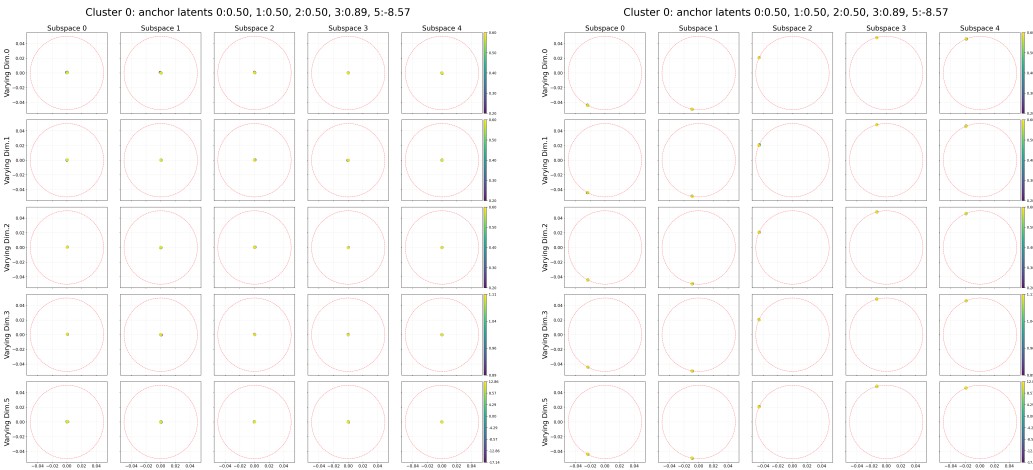

Figure 6: Visualization of Shape3D per aligned rotation subspace. Left: w/o $\mathcal{L}_{\text{radius}}$, all the features collapse to the rotation center. Right: w/o $\mathcal{L}_{\text{theta}}$, all subspaces collapse to a point, i.e., the learned rotation matrix is an identity matrix.

matching the theoretical maximum for three points on a sphere (Tammes, 1930). With fixed (un-learned) rotation centers, we obtain $d_{\min} = 1.719$.

## F    FULL EXPERIMENT RESULTS

### F.1    EXTENDED RESULTS FOR SECTION 4.2

Table 4: DCI using Linear regression on four datasets shown compactly as triples **D/C/I** = Disen-tanglement/Completeness/Informativeness.

| Method/Dataset | MPI3D (D/C/I) | Shape3D (D/C/I) | 3DIdent (D/C/I) | 3DIEBench (D/C/I) |
|---|---|---|---|---|
| (*Self-Supervised Invariance*) | | | | |
| SimCLR | 0.234 / 0.078 / 0.144 | **0.996** / 0.099 / 0.167 | 0.286 / 0.084 / 0.313 | 0.231 / 0.099 / 0.173 |
| (*Self-Supervised Equivariance*) | | | | |
| EquiMOD | 0.545 / 0.120 / 0.203 | 0.851 / 0.168 / 0.483 | 0.232 / 0.078 / 0.309 | 0.736 / 0.115 / 0.289 |
| CARE | 0.236 / 0.111 / 0.169 | 0.842 / 0.066 / 0.181 | 0.243 / 0.080 / 0.329 | 0.174 / 0.096 / 0.309 |
| (*Self-Supervised Disentanglement*) | | | | |
| IP-IRM | 0.654 / 0.147 / 0.346 | 0.357 / 0.114 / 0.132 | 0.202 / 0.070 / 0.431 | 0.534 / 0.122 / 0.250 |
| Eastwood et al. (2023) | 0.950 / 0.700 / 0.414 | 0.962 / 0.821 / 0.694 | 0.879 / 0.693 / 0.713 | 0.948 / 0.501 / 0.276 |
| CD-RED (before post-processing) | 0.628 / 0.505 / **1.000** | 0.630 / 0.494 / **1.000** | 0.599 / 0.471 / 0.996 | 0.106 / 0.067 / 0.309 |
| CD-RED (after post-processing) | **0.951** / **0.952** / 0.998 | 0.986 / **0.979** / 0.999 | **0.980** / **0.980** / 0.996 | **0.986** / **0.992** / **1.000** |

Table 5: Per-dimension Informativeness using linear regression on **Shapes3D**. Columns list ground-truth factors

| Method | Floor Hue | Wall Hue | Object Hue | Scale | Orientation | Class |
|---|---|---|---|---|---|---|
| SimCLR | 0.0000 | 0.0000 | 0.0000 | 0.0000 | 0.0000 | 1.0000 |
| EquiMOD | 0.3071 | 0.5003 | 0.1089 | 0.0005 | 0.9789 | 0.9997 |
| CARE | 0.0158 | 0.0191 | 0.0185 | 0.0106 | 0.0216 | 0.9986 |
| IP-IRM | 0.0279 | 0.0260 | 0.0049 | 0.0259 | 0.9949 | 0.9967 |
| Eastwood et al. (2023) | 0.5410 | 0.8655 | 0.6672 | 0.6553 | 0.0003 | 0.7868 |
| CD-RED (before post-proc.) | 0.6860 | 0.8241 | 0.9620 | 0.5743 | 0.8592 | 0.9998 |
| CD-RED (after post-proc.) | 0.9995 | 0.9995 | 0.9994 | 0.9984 | 0.9994 | 1.0000 |

Table 6: Per-dimension Informativeness using Random forest regression on **Shapes3D**. Columns list ground-truth factors

| Method | Floor Hue | Wall Hue | Object Hue | Scale | Orientation | Class |
|---|---|---|---|---|---|---|
| SimCLR | 0.8637 | 0.8719 | 0.8598 | 0.8582 | 0.8581 | 1.0000 |
| EquiMOD | 0.9798 | 0.9836 | 0.9497 | 0.8953 | 0.9998 | 1.0000 |
| CARE | 0.9200 | 0.9198 | 0.9011 | 0.9093 | 0.8968 | 1.0000 |
| IP-IRM | 0.9327 | 0.9415 | 0.9035 | 0.9489 | 1.0000 | 1.0000 |
| Eastwood et al. (2023) | 1.0000 | 1.0000 | 1.0000 | 1.0000 | 1.0000 | 1.0000 |
| CD-RED (before post-proc.) | 1.0000 | 1.0000 | 0.9999 | 1.0000 | 1.0000 | 1.0000 |
| CD-RED (after post-proc.) | 1.0000 | 1.0000 | 1.0000 | 1.0000 | 1.0000 | 1.0000 |

Table 7: Per-dimension Informativeness using linear regression on **MPI3D**. Columns list the dataset's ground-truth factors.

| Method | Obj. Color | Obj. Size | Cam. Height | Backg. Color | Arm Horiz. | Arm Vert. | Class |
|---|---|---|---|---|---|---|---|
| SimCLR | 0.0005 | 0.0011 | 0.0017 | 0.0008 | 0.0003 | 0.0008 | 1.0000 |
| EquiMOD | 0.4191 | 0.0001 | 0.0002 | 0.0008 | 0.0002 | 0.0002 | 0.9998 |
| CARE | 0.0323 | 0.0562 | 0.0749 | 0.0118 | 0.0021 | 0.0059 | 0.9999 |
| IP-IRM | 0.7589 | 0.0300 | 0.0205 | 0.0015 | 0.0018 | 0.0028 | 0.1049] |
| Eastwood et al. (2023) | 0.5896 | 0.4471 | 1.0000 | 1.0000 | 0.9996 | 0.8018 | 0.0199] |
| CD-RED (before post-proc.) | 0.3758 | 0.0466 | 0.2571 | 0.3304 | 0.0552 | 0.8311 | 0.8388 |
| CD-RED (after post-proc.) | 0.9953 | 0.9983 | 0.9984 | 0.9950 | 0.9977 | 0.9982 | 1.0000 |

Table 8: Per-dimension Informativeness using Random forest regression on **MPI3D**. Columns list the dataset's ground-truth factors.

| Method | Obj. Color | Obj. Size | Cam. Height | Backg. Color | Arm Horiz. | Arm Vert. | Class |
|---|---|---|---|---|---|---|---|
| SimCLR | 0.8575 | 0.8580 | 0.8576 | 0.8562 | 0.8572 | 0.8564 | 1.0000 |
| EquiMOD | 0.9911 | 0.9916 | 0.9967 | 0.9962 | 0.8724 | 0.8614 | 0.9999 |
| CARE | 0.8795 | 0.8772 | 0.8885 | 0.8597 | 0.8648 | 0.8667 | 1.0000 |
| IP-IRM | 0.9881 | 0.9527 | 0.9785 | 0.9768 | 0.8822 | 0.8877 | 0.9994 |
| Eastwood et al. (2023) | 1.0000 | 0.9992 | 1.0000 | 1.0000 | 1.0000 | 1.0000 | 0.9994 |
| CD-RED (before post-proc.) | 0.9995 | 0.9999 | 0.9998 | 1.0000 | 0.9997 | 0.9998 | 1.0000 |
| CD-RED (after post-proc.) | 0.9999 | 1.0000 | 1.0000 | 1.0000 | 1.0000 | 1.0000 | 1.0000 |

Table 9: Per-dimension Informativeness using Linear Regression on **3DIdent**. Columns follow the latents: Position $(X, Y, Z)$, Rotation $(\phi, \theta, \psi)$, Spotlight Position, and Hue (Object, Ground, Spotlight).

| Method | Position | | | Rotation | | | Spotlight Pos. | Hue | | |
|---|---|---|---|---|---|---|---|---|---|---|
| | X | Y | Z | $\phi$ | $\theta$ | $\psi$ | Position | Obj. | Ground | Spotlight |
| SimCLR | 0.3728 | 0.0121 | 0.2591 | 0.0005 | 0.0037 | 0.0006 | 0.2361 | 0.4108 | 0.1706 | 0.9747 |
| EquiMOD | 0.4786 | 0.0884 | 0.4086 | 0.0008 | 0.0304 | 0.0021 | 0.5336 | 0.5095 | 0.2511 | 0.0996 |
| CARE | 0.4726 | 0.0352 | 0.3177 | 0.0007 | 0.0061 | 0.0013 | 0.2237 | 0.4929 | 0.1067 | 0.9599 |
| IP-IRM | 0.5563 | 0.0937 | 0.6513 | 0.0013 | 0.0270 | 0.0037 | 0.4954 | 0.7152 | 0.2584 | 0.9309 |
| Eastwood et al. (2023) | 0.7366 | 0.6961 | 0.7102 | 0.8021 | 0.7524 | 0.8062 | 0.7258 | 0.6957 | 0.5405 | 0.3730 |
| CD-RED (before post-proc.) | 0.9965 | 0.9971 | 0.9971 | 0.9985 | 0.9982 | 0.9988 | 0.9943 | 0.9805 | 0.9904 | 0.9981 |
| CD-RED (after post-proc.) | 0.9968 | 0.9974 | 0.9974 | 0.9987 | 0.9983 | 0.9989 | 0.9947 | 0.9809 | 0.9908 | 0.9984 |

Table 10: Per-dimension Informativeness using Random forest regression on **3DIdent**. Columns follow the latents: Position $(X, Y, Z)$, Rotation $(\phi, \theta, \psi)$, Spotlight Position, and Hue (Object, Ground, Spotlight).

| Method | Position | | | Rotation | | | Spotlight Pos. | Hue | | |
|---|---|---|---|---|---|---|---|---|---|---|
| | X | Y | Z | $\phi$ | $\theta$ | $\psi$ | Position | Obj. | Ground | Spotlight |
| SimCLR | 0.9599 | 0.8738 | 0.9521 | 0.8533 | 0.8567 | 0.8517 | 0.9457 | 0.9629 | 0.9326 | 0.9993 |
| EquiMOD | 0.9584 | 0.8824 | 0.9651 | 0.8558 | 0.8627 | 0.8562 | 0.9535 | 0.9802 | 0.9536 | 0.9956 |
| CARE | 0.9554 | 0.8741 | 0.9555 | 0.8543 | 0.8564 | 0.8540 | 0.9419 | 0.9796 | 0.9349 | 0.9994 |
| IP-IRM | 0.9546 | 0.8799 | 0.9655 | 0.8554 | 0.8616 | 0.8561 | 0.9541 | 0.9813 | 0.9468 | 0.9923 |
| Eastwood et al. (2023) | 1.0000 | 1.0000 | 0.9999 | 0.9997 | 0.9999 | 0.9997 | 0.9999 | 0.9938 | 0.9969 | 0.9666 |
| CD-RED (before post-proc.) | 0.9995 | 0.9996 | 0.9996 | 0.9998 | 0.9997 | 0.9998 | 0.9992 | 0.9977 | 0.9989 | 0.9998 |
| CD-RED (after post-proc.) | 0.9996 | 0.9996 | 0.9996 | 0.9998 | 0.9998 | 0.9998 | 0.9993 | 0.9978 | 0.9989 | 0.9998 |

Table 11: Per-dimension Informativeness on **3DIEBench** using Linear Regression. Columns list the dataset's ground-truth factors.

| Method | Position X | Position Y | Position Z | Rotation $\psi$ | Ground Hue | Class |
|---|---|---|---|---|---|---|
| SimCLR | 0.0007 | 0.0027 | 0.0057 | 0.0014 | 0.0299 | 0.9997 |
| EquiMOD | 0.0008 | 0.0007 | 0.0016 | 0.0006 | 0.7297 | 1.0000 |
| CARE | 0.0011 | 0.0099 | 0.0131 | 0.0006 | 0.8314 | 0.9996 |
| IP-IRM | 0.0080 | 0.1693 | 0.3599 | 0.0200 | 0.0394 | 0.8489 |
| Eastwood et al. (2023) | 0.0011 | 0.2802 | 0.2391 | 0.0173 | 0.3762 | 0.7424 |
| CD-RED (before post-proc.) | 0.4318 | 0.0687 | 0.0600 | 0.2215 | 0.0069 | 0.9993 |
| CD-RED (after post-proc.) | 0.9989 | 0.9988 | 0.9988 | 0.9987 | 0.9994 | 1.0000 |

Table 12: Per-dimension Informativeness on **3DIEBench** using Random forest Regression. Columns list the dataset's ground-truth factors.

| Method | Position X | Position Y | Position Z | Rotation $\psi$ | Ground Hue | Class |
|---|---|---|---|---|---|---|
| SimCLR | 0.8564 | 0.8738 | 0.8805 | 0.8635 | 0.9475 | 1.0000 |
| EquiMOD | 0.8555 | 0.8730 | 0.8725 | 0.8697 | 0.9977 | 1.0000 |
| CARE | 0.8687 | 0.9376 | 0.9495 | 0.9133 | 0.9897 | 1.0000 |
| IP-IRM | 0.8696 | 0.9353 | 0.9527 | 0.9118 | 0.9046 | 1.0000 |
| Eastwood et al. (2023) | 0.8654 | 1.0000 | 0.9999 | 0.8958 | 0.9502 | 1.0000 |
| CD-RED (before post-proc.) | 0.9997 | 0.9996 | 0.9997 | 0.9997 | 0.9998 | 1.0000 |
| CD-RED (after post-proc.) | 0.9998 | 0.9998 | 0.9998 | 0.9998 | 0.9999 | 1.0000 |

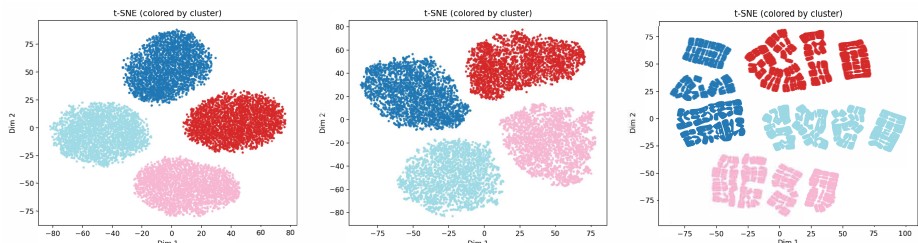

Figure 7: t-SNE visualization of learned features on **Shapes3D**. **Left**: Simclr; **Middle**: CARE; **Right**: Ours.

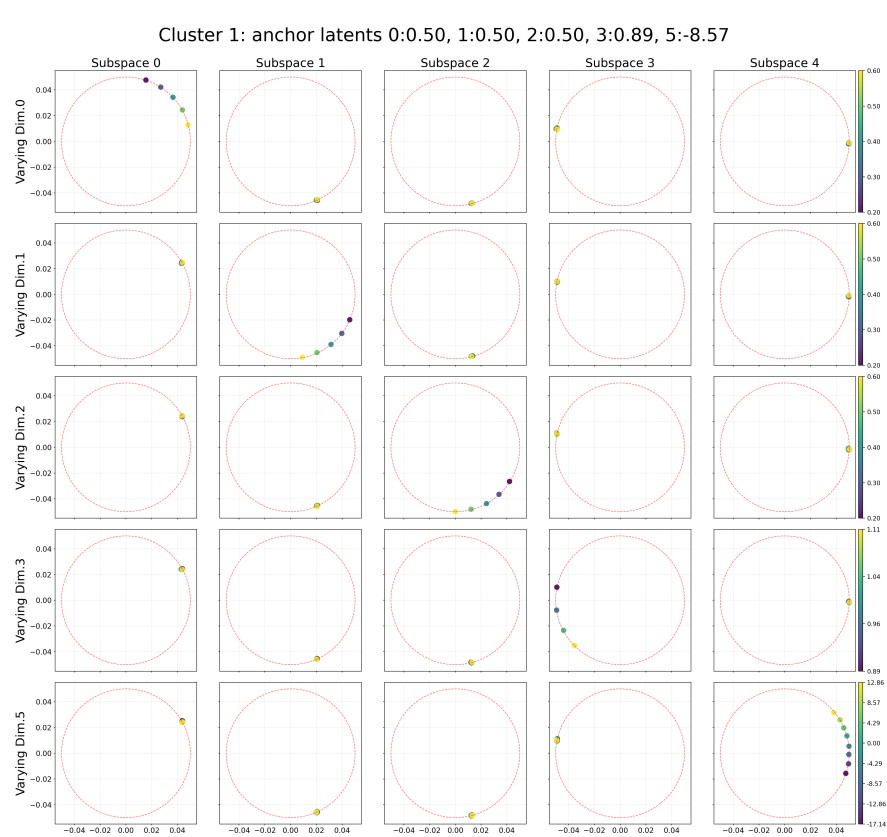

Figure 8: Randomly pick an anchor in one cluster and visualize each latent dimension varying on **Shapes3D**. Dim 0: Floor Hue, Dim 1: Wall Hue, Dim 2: Object Hue Dim 3: Scale Dim 5: Orientation

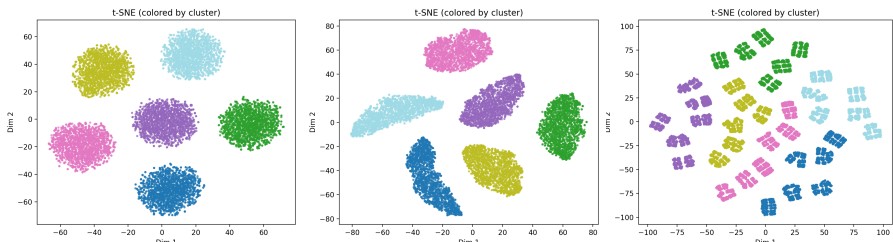

Figure 9: t-SNE visualization of learned features on **MPI3D**. **Left**: Simclr; **Middle**: CARE; **Right**: Ours.

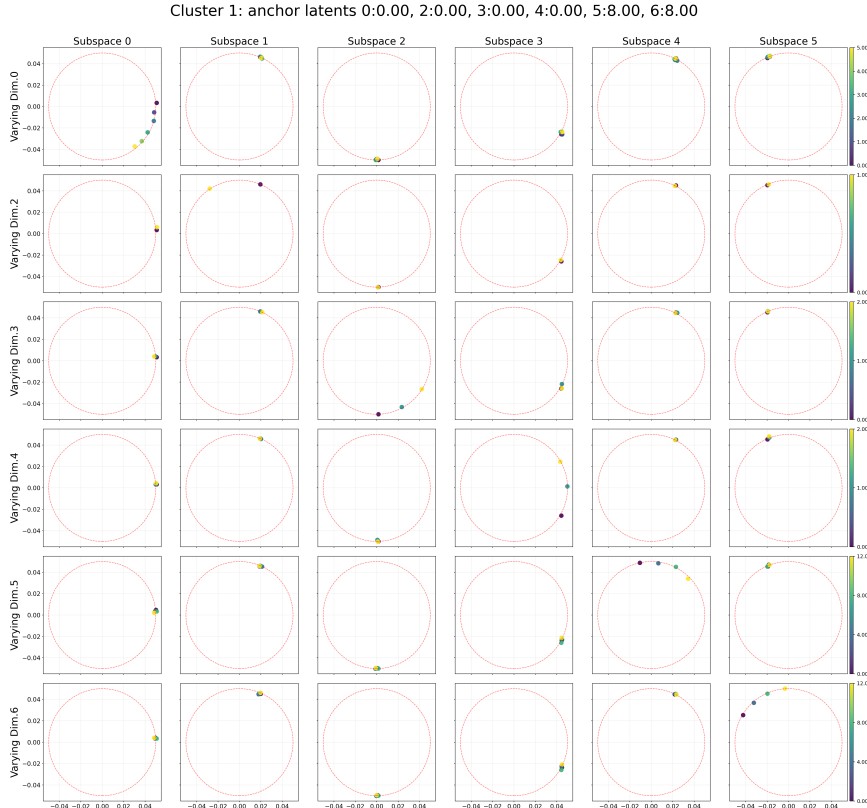

Figure 10: Randomly pick an anchor in one cluster and visualize each latent dimension varying on **MPI3D**. Dim 0: Floor Hue, Dim 1: Wall Hue, Dim 2: Object Hue, Dim 3: Scale, Dim 5: Orientation

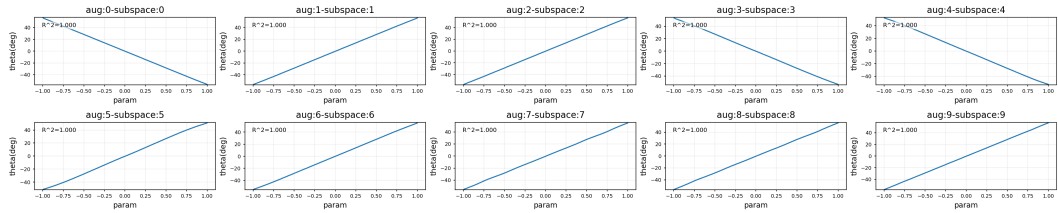

Figure 11: Transformation network output for each augmentation in **3DIdent**. Randomly pick 1000 points within the augmentation range.

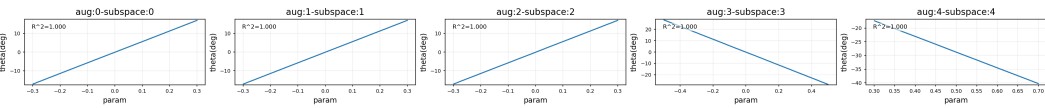

Figure 12: Transformation network output for each augmentation **3DIEBench**. Randomly pick 1000 points within the augmentation range.

## F.2 EXTENDED RESULTS FOR SECTION 4.3

Figure 13: Experiment with x-translation, y-translation, and xy-translation. It has effectively disentangled the x-translation and y-translation, while the xy-translation learns to perfectly decompose and align its caused variations to the two subspaces assigned to the x-translation and y-translation.

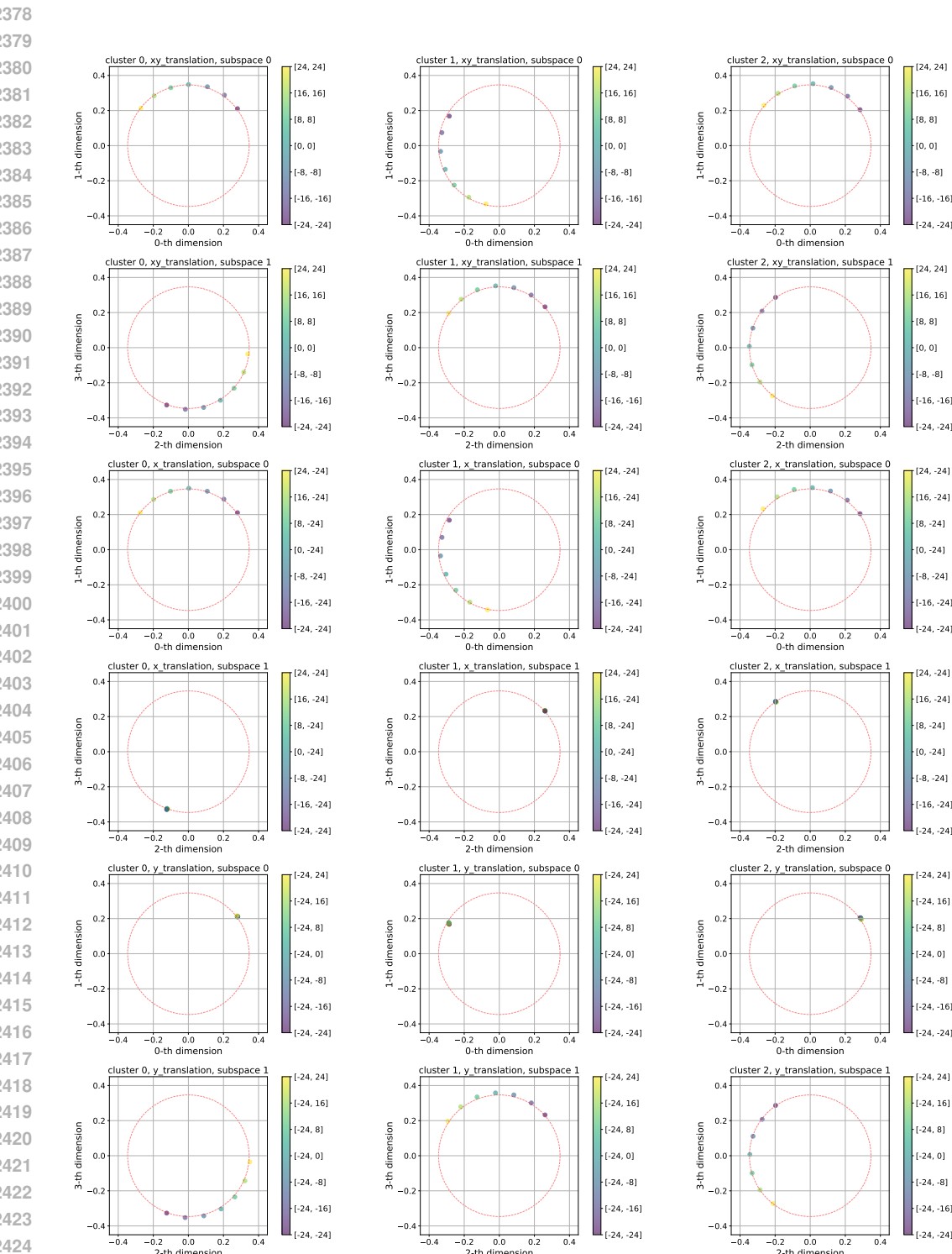

Figure 14: Full visualization: x-translation, y-translation, xy-translation.

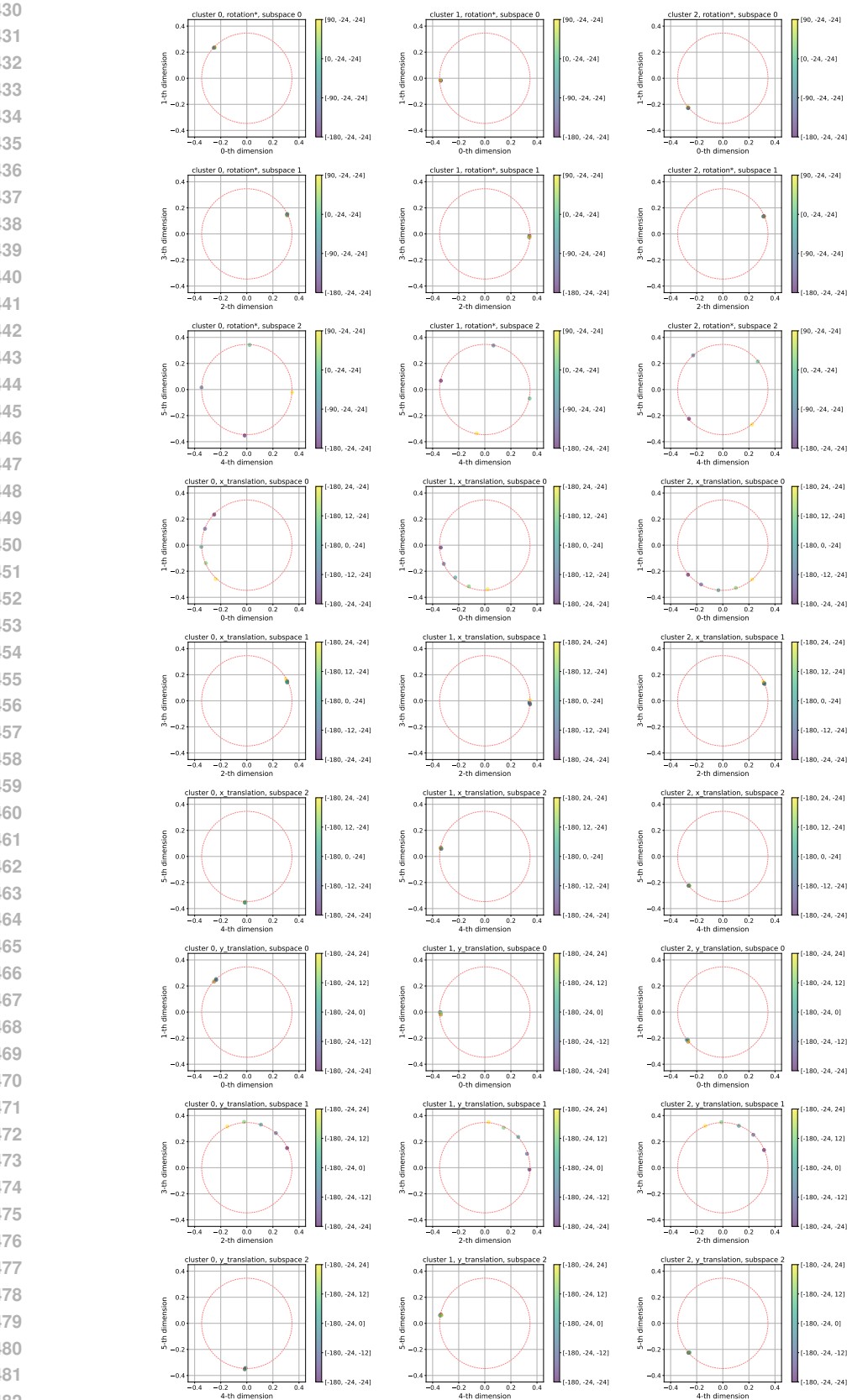

Figure 15: Full visualization: x-translation, y-translation, image-centered rotation.

