# OpenReview forum: "Self-Supervised Disentanglement via Cluster-Dependent Rotational Equivariance"
_ICLR.cc/2026/Conference — Submitted to ICLR 2026_

### Official Review · Reviewer_q1ZX · 2025-10-29

**Soundness:** 2
**Presentation:** 2
**Contribution:** 1
**Rating:** 4
**Confidence:** 4

**Summary:**

The paper aims to achieve self-supervised disentanglement of content and style factors in images without labels by enforcing geometric constraints on learned representations. Specifically, it proposes a method that learns features that form content-invariant clusters and conforms to style-equivariant rotations. Under assumptions that augmentations affect only one style factor at a time and leave content invariant, CD-RED provably recovers content and style subspaces up to rotation. Experiments on synthetic datasets show that the proposed approach can achieve near perfect disentanglement after post-processing, surpassing baselines.

**Strengths:**

1. The proposed approach unifies content invariance and style equivariance into a single self-supervised pipeline, hence it can in principle be built atop existing SSL backbones (SimCLR, MoCo,...) with minimal architectural changes.
2. It introduces a clean geometric model of disentanglement by representing each style factor as rotations in 2-D subspaces.
3. Within controlled, well-specified environments, the method consistently yields well-disentangled features.

**Weaknesses:**

- The proposed approach makes several strong assumptions that are hard to satisfy off the lab bench:
    1. The numbers of style factors are known a priori.
    2. $L_\theta$ assumes access to a user-provided proxy that bounds the true amount an augmentation moves the underlying latent.
    3. Augmentations change only the style but never content.
    4. User essentially has access to the ground-truth data generating process, being able to intervene the styles of images through augmentations while knowing the exact magnitude of the intervention.

- The strong disentanglement results rely on a hand-crafted post-processing stage. This leads to unfair comparisons with baselines, which do not leverage post-processing. For example, the post-processing includes concatenation of a one-hot content code, which can boosts completeness and disentanglement score.

- Experiments are only on synthetic datasets such as Shapes3D, MPI3D, 3DIdent, where they use the ground-truth latent transformations as augmentations (with the exact parameter 𝑡). This is impractical for the real self-supervised regime in the wild, where estimating 𝑡 and $q_a(t)$ is difficult. There’s no experimental evidence on natural images (ImageNet-like) or real augmentations with unknown latent magnitudes (e.g., crop, jitter).

**Questions:**

1. What happens if only a subset of ground-truth style augmentation can be intervened? For example, what if a dataset primarily contains objects from several viewpoints, and viewpoint (as a style latent) can not be intervened?
2. How are the hyperparameters chosen? Could you show a sensitivity analysis?
3. How does CD-RED perform on natural images with standard SSL augmentations?
4. Without appending the one-hot cluster code, how close do D/C get after the same post-processing?

---

> ### Author Response · Authors · 2025-11-24
> **Response to Reviewer q1ZX Part 1/2**
>
> Thank you for the insightful review. We sincerely appreciate your recognition of the overall strengths of our work. We address the concerns and questions you raised as follows.
>
> **Q1: What happens if augmentation can intervene only on a subset of ground-truth styles? For example, what if a dataset primarily contains objects from several viewpoints, and the viewpoint (as a style latent) can not be intervened on?**
>
> - In our framework, "style" is defined as the set of factors that can be changed by the chosen set of augmentations. If a ground-truth factor (such as viewpoint) cannot be altered by any augmentation, then our method will not treat it as a style latent. If samples with different viewpoints are not linked through the employed augmentations, they will fall into different clusters; within each cluster, the viewpoint is effectively fixed and thus becomes part of the "content" latent. Note that this "style"-"content" division is, in fact, intrinsic to augmentation-based self-supervised learning.
>
> **Q2: How are the hyperparameters chosen? Could you show a sensitivity analysis?**
>
> - The hyperparameters used in our experiments are listed in Appendix D.2. The key hyperparameters in our framework are $\lambda_{\text{radius}}$ and $\lambda_{\text{theta}}$, which, according to our experiments, are generally insensitive. We typically set them to $1$. If some loss rebalancing is required, we additionally consider $0.1$ and $10$, which generally suffice.
>
> **Q3: How does CD-RED perform on natural images with standard SSL augmentations?**
>
> - Our framework requires deterministic augmentations with known and accessible augmentation parameters. Standard SSL augmentations are largely stochastic, so we need to use their parameterized deterministic counterparts.
>
> - An unfortunate fact is that standard SSL augmentations, when applied to natural-image datasets, cannot effectively link all samples within each semantic cluster. This is intrinsically why self-supervised classification accuracies remain imperfect and represents a long-term fundamental challenge for augmentation-based self-supervised learning.
>
> **Q4: Without appending the one-hot cluster code, how close do D/C get after the same post-processing?**
>
> - Here is a comparison of the D/C/I results with and without appending the one-hot cluster code, using random forest regression.
> | | | |
> |:----------|:-------------------------|:----------------------------|
> | Dataset   | w/o one hot code (D/C/I) | with one hot code (D/C/I)         |
> | Shape3d   | 0.745 / 0.834 / 0.978    | **1.000** / **1.000** / **1.000** |
> | MPI3D     | 0.789 / 0.857 / 0.980    | **1.000** / **1.000** / **1.000** |
> | 3DIEBench | 0.745 / 0.829 / 0.976    | **0.996** / **0.996** / **1.000** |
>
> - Our post-processing essentially extracts the style-variation features within each cluster and encodes them into a shared interpretable coordinate, which by itself contains no content information unless the cluster code is appended. In other words, in our setting, appending the cluster code is necessary to reintroduce the content information.

---

> ### Author Response · Authors · 2025-11-24
> **Response to Reviewer q1ZX Part 2/2**
>
> **W1-1: The number of style factors is known a priori.**
>
> - As clarified in our response to Q1, "style" in our framework is defined as the set of factors that can be changed by the chosen augmentations. Therefore, we only need to know the number of style factors affected by the chosen set of augmentations, which is generally possible and easy to determine.
>
> **W1-2: $L_{\theta}$ assumes access to a user-provided proxy that bounds the true amount an augmentation moves the underlying latent.**
>
> - At its core, it suffices to ensure that the transformation associated with any non-trivial augmentation is non-trivial. We introduce such a user-provided proxy for a more sophisticated solution because we believe it is generally easy to provide, as it only requires a lower bound. At its very least, we need to be provided with whether the augmentation is trivial or not under the used augmentation parameter.
>
> **W1-3: Augmentations change only the style but never the content.**
>
> - As clarified in our response to Q1, "style" in our framework is defined as the set of factors that can be changed by the chosen augmentations, and "content" is the set of factors that cannot be changed.
>
> - In other words, if augmentations change the so-called content, then the content is actually an augmentation-changeable "style" in our framework.
>
> **W1-4: User essentially has access to the ground-truth data-generating process, knowing the exact magnitude of the intervention caused by augmentation.**
>
> - We only require a lower bound of the magnitude, which, as noted in response to W1-2, can even reduce to whether the augmentation is trivial or not under the used augmentation parameter.
>
> **W2: The strong disentanglement results rely on a hand-crafted post-processing stage. This leads to unfair comparisons with baselines.**
>
> - In Table 1, we report the CD-RED results both before and after post-processing. Both consistently outperform the baselines.
>
> **W3-1: For the real self-supervised regime in the wild, estimating $t$ and $q_{a(t)}$ is difficult.**
>
> - We need to use parameterized deterministic versions of these augmentations. Then $t$ is simply the parameter used for the augmentation, which is directly accessible and needs no estimation.
>
> - We only require $\tilde{q}\_{a(t)}$ as a lower-bound for the true magnitude $q_{a(t)}$, and as we noted in the response to W1-2, it can even reduce to whether the augmentation is trivial or not under the used augmentation parameter.
>
> **W3-2: There’s no experimental evidence on real augmentations with unknown latent magnitudes (e.g., crop, jitter).**
>
> - For stochastic augmentations like crop and jitter, we need to use their parameterized deterministic counterparts. Then, given the augmentation parameters, the magnitudes are estimable.
>
> - Crop and jitter changes multiple latents simultaneously. However, they can typically be decomposed (truly or virtually) into simpler augmentations. For instance, a crop can be decomposed into x-translation, y-translation, x-scaling, and y-scaling.
>
> **W3-3: There’s no experimental evidence on natural images (ImageNet-like).**
>
> - Our work, in its current form, is primarily theoretical: we propose a new framework with provably strong disentangling guarantees and empirically substantiate these results on synthetic and controlled datasets. These existing elements in the paper are self-contained and constitute a solid, standalone contribution.
>
> - Attaining disentanglement on real-world data requires substantial additional effort. Since commonly used augmentations for real-world datasets can not effectively link all samples within each cluster. This is a common fundamental practical challenge for augmentation-based self-supervised learning, which is, in fact, hard to resolve. Besides, practically available augmentations are typically stochastic, with their parameters not directly accessible. Our method instead requires deterministic augmentation with known and accessible augmentation parameters. Addressing this and other similar practical issues also requires a non-trivial amount of engineering effort. Therefore, we would like to leave this line of investigation as future work.
>
> - Genuine real-world disentanglement results are largely absent in existing self-supervised equivariant or disentanglement methods. Some works report classification accuracy as a proxy; however, this metric primarily reflects clustering ability and offers little insight into true disentanglement quality.
>
> ---
> We sincerely hope that the above clarifications adequately address your concerns. Please let us know if any aspect remains unclear. We would be glad to provide further details.

---

### Official Review · Reviewer_tJGx · 2025-10-29

**Soundness:** 3
**Presentation:** 2
**Contribution:** 3
**Rating:** 6
**Confidence:** 4

**Summary:**

- The paper introduces CD-RED, a self-supervised framework for learning disentangled and equivariant representations without labels.
- CD-RED addresses this via a two-stage training pipeline: Stage 1: Contrastive (InfoNCE) training to obtain semantic clusters with uniform hyperspherical distribution; Stage 2: Cluster-wise equivariance learning using explicit SO(2) block rotations within orthogonal subspaces, centered at cluster centroids and aligned via Householder transformations.
- Theoretically, the authors prove equivariance and strong disentanglement under mild conditions, supported by clear geometric reasoning.
- Empirically, CD-RED achieves near-perfect DCI scores on MPI3D, Shape3D, 3DIdent, and 3DIEBench, outperforming SimCLR, EquiMOD, and CARE.

**Strengths:**

- The paper is logically structured and clear geometric intuition (rotations on hypersphere) nicely complement theoretical intuitions.
- Clean geometric design with explicit group structure where augmentations are encoded as a direct product of $SO(2)$ planes.

**Weaknesses:**

- The work assumes access to augmentation parameters (or good proxies) to drive the loss. In real SSL pipelines (random crops, color jitter, CutOut), reliable $q_{a(t)}$ may not exist or may be weakly correlated with semantic style change (e.g., crop boxes can be semantically neutral or catastrophic depending on content).
- The paper “specifies” which dimensions each augmentation acts on rather than learning the subspace; this is robust on synthetic datasets but could fail for real augmentations which often co-vary multiple unknown factors
- The method is beautifully matched to geometric transforms (rot/shift) that act like circle motions. Photometric transforms amongst others (e.g., color jitter, noise, blur) do not naturally map to SO(2) with clean anchors.
- Empirical scope is limited to synthetic datasets (MPI3D, Shape3D, etc.); real-world or high-dimensional visual domains (ImageNet, CIFAR100, etc.) are not tested. This leaves open questions about scalability and robustness to complex augmentations.
- The “perfect disentanglement” claim is heavily theoretical and relies on ideal cluster assignments and clean augmentation-factor separability. Performance under imperfect clustering would be interesting to study
- The assumption that augmentations can be neatly mapped to orthogonal 2D planes may not generalize to non-orthogonal or stochastic transformations (e.g., blur + rotation).

**Questions:**

See weaknesses above

---

> ### Author Response · Authors · 2025-11-24
> **Response to Reviewer tJGx Part 1/2**
>
> Thank you for your insightful and inspiring review. We sincerely appreciate your recognition of the clear geometric intuition and the overall geometric clarity of our work. We address the concerns you raised as follows.
>
> **W1: In real SSL pipelines (random crops, color jitter, CutOut), reliable $q_{a(t)}$ may not exist or may be weakly correlated with semantic style change.**
>
> - Such composed stochastic augmentations can typically be replaced (truly or virtually) by simpler deterministic augmentations with known parameters and strengths. For example, a random crop can be decomposed into x-translation, y-translation, x-scaling, and y-scaling.
>
> **W2: Real augmentations often (co-)vary multiple unknown factors.**
>
> - These composed stochastic augmentations can typically be decomposed into, and replaced by, multiple simpler deterministic augmentations.
>
> **W3: Photometric transforms (e.g., color jitter, noise, blur) do not naturally map to SO(2) with clean anchors.**
>
> Photometric transformations are indeed more complex, but our framework can still handle them to a substantial extent.
>
> - Brightness and contrast in color jitter are inherently disentangled and can be modeled by two separate SO(2) spaces. Hue, saturation, and value (i.e., HSV) are also approximately disentangled, which may be well modeled in three SO(2) spaces. However, these two sets of factors operate in different color spaces, and the intrinsic dimensionality of color jitter is high. Jointly modeling all these factors may therefore require further theoretical development, which we consider an interesting direction for future work.
>
> - Noise is essentially random. Equivariant modeling based on, for example, its mean and variance is likely not theoretically feasible. Nonetheless, our framework may be reasonably extended to model it as invariance, with its associated rotational transformation restricted to be trivial (i.e., identity). Note that, in our equivariant modeling, transformations are regularized to be non-trivial.
>
> - Blur, when used as a binary augmentation (on/off), can be naturally modeled with a single SO(2) space. When blur is instead parameterized by a continuous strength, a principled theoretical modeling in a SO(2) space would require it to form a semi-group: if $g_1$ and $g_2$ are two valid blur transformations in the group, then their composition $g_1 \odot g_2$ must also be in the group. This would require some adjustment to the typically used blur augmentation.
>
> **W4: Real-world or high-dimensional visual domains are not tested. This leaves open questions about scalability and robustness to complex augmentations.**
>
> - Theoretically, our method will not be negatively impacted when the data dimension scales up, as long as augmentations can effectively link samples within each cluster.
>
> - Commonly used complex augmentations can typically be decomposed into simpler ones. For those that cannot, our method can also simultaneously allocate multiple subspaces for effective modeling.

---

> ### Author Response · Authors · 2025-11-24
> **Response to Reviewer tJGx Part 2/2**
>
> **W5: Performance under imperfect clustering would be interesting to study.**
>
> - We use a first-stage training of InfoNCE for clustering initialization. However, the second stage does not fixedly use the initial cluster assignment, but instead dynamically assigns samples to their nearest centers. Thereby, minor errors in cluster assignments can be corrected. This relieves their dependence on perfect cluster assignments. We have clarified and highlighted this in the revised manuscript.
>
> - If two semantic clusters somehow collapse into a single one (i.e., sharing one rotation center), the theoretical result is that they will just share the same feature space without mutual interference. We will then be unable to distinguish the two semantic clusters based on their features, and the strong disentanglement will become infeasible since the feature space used may be (and only be) partially overlapped.
>
> **W6: The assumption that augmentations can be neatly mapped to orthogonal 2D planes may not generalize to non-orthogonal or stochastic transformations (e.g., blur + rotation).**
>
> - We need to use the parameterized deterministic counterparts of stochastic transformations.
>
> - Our framework assumes data variation factors are orthogonal to each other: only in this sense, they can be disentangled. Nevertheless, our framework can be easily extended to model data variation factors that are non-orthogonal to each other, allocating multiple SO(2) spaces simultaneously. It is also worth noting that our framework can handle composed augmentations that vary multiple orthogonal factors simultaneously in an arbitrary (e.g., correlated/non-orthogonal) manner.
>
> - For blur and rotation, their changed semantics are in fact orthogonal to each other. Nonetheless, their practical implementations can interfere with each other and potentially lead to inconsistent observations with the same semantics. Such inconsistency can be mitigated by using a larger blur kernel. However, the inconsistency cannot be entirely eliminated due to the rasterization of images and numerical errors, and this in fact commonly exists for a range of augmentations, including blur, scaling, and rotation.
>
> ---
> We sincerely hope that the above clarifications adequately address your concerns. Please let us know if any aspect remains unclear. We would be glad to provide further details.

---

> > ### Comment · Reviewer_tJGx · 2025-11-26
> >
> > Thank you for the detailed rebuttal and clarifications. However, I remain concerned that several key limitations are not fully resolved but rather shifted into assumptions. In practice, many SSL augmentations (crop, CutOut, heavy photometric mixes) do not admit a clean, semantics-preserving deterministic decomposition with reliable parameters, yet CD-RED still fundamentally requires access to augmentation parameters/proxies to define the per-factor actions.
> >
> > Similarly, “allocating multiple SO(2) spaces” does not address the core issue that the method prespecifies subspace assignments instead of learning them, which is precisely what may break outside synthetic settings where factors are clean and orthogonal.
> >
> > Finally, the photometric extensions read more like plausible future directions than something the current framework supports cleanly. Further, the lack of evaluation on real-world benchmarks (e.g., ImageNet) makes the scalability and generalization claims difficult to assess in practice.

---

> > > ### Author Response · Authors · 2025-11-27
> > > **Further Response to Reviewer tJGx Part 1/2**
> > >
> > > Thank you for your prompt and clear reply.
> > >
> > > We acknowledge that our work, in its current form, is primarily theoretical. At this stage, we are unable to empirically address the practical concerns you raised. Our primary goal here is to clarify that, from a theoretical standpoint, our proposed framework does not inherently lack the capabilities in question.
> > >
> > > For **the theoretical modeling ability**, we would like to further clarify the following:
> > > - Crop can be decomposed into x-translation, y-translation, x-scaling, and y-scaling. We have clarified this in the previous response. Did you miss it or have a different view? Perhaps we could have a more detailed discussion. Essentially, we can recover any crop operation by first shifting the image center by x- and y-translations and then rescaling the image along x and y.
> > > - CutOut intrinsically contains four independent factors: the upper-left x, upper-left y, bottom-right x, and bottom-right y coordinates of the mask. Thus, CutOut can be viewed as a composition of four disentangled factors. Accordingly, we may allocate four separate SO(2) subspaces, one for each factor, for effective modeling.
> > > - By "heavy photometric mixes", do you mean brightness + contrast + hue + saturation + value as in color jitter? As we mentioned in the previous response, we can model brightness + contrast; we may also model hue + saturation + value, but we cannot jointly model all these together. We would like to clarify that this is not a theoretical limitation of our method; rather, it is a fundamental challenge that is essentially impossible for any equivariant approach to fully address.
> > > - The feasibility of modeling noise (or any augmentation whose induced data variation is orthogonal to all other variations) as invariance is clear. The key here is that these variations are orthogonal to others. There is no geometric contradiction. You may imagine that their allocated subspaces are forcibly collapsed.
> > > - For blur and color jitter, in addition to the theoretical treatments mentioned earlier, we note that another option is to model them as invariances. In particular, modeling all non-orthogonal factors in color jitter jointly as invariances provides a principled solution to the challenge discussed above.
> > > - For variation factors that are non-orthogonal, they are fundamentally entangled and therefore cannot be disentangled in principle. Consequently, the need to model them using multiple SO(2) subspaces is not a limitation of our framework. On the contrary, the ability to model such non-orthogonal variations is a noteworthy advantage of our method, which existing self-supervised disentanglement methods generally lack.
> > > - For complex augmentations that are not easily decomposable, modeling them using multiple SO(2) subspaces can still provide theoretical benefits. Although their internal factors remain entangled, the augmentation as an entity is still disentangled from variations encoded in other SO(2) subspaces.
> > >
> > > For **the lack of evaluation on real-world benchmarks**, we would like to clarify the following:
> > > - Our work, in its current form, is primarily theoretical: we propose a new framework with provably strong disentangling guarantees and empirically substantiate these results on synthetic and controlled datasets. These existing elements in the paper are self-contained and constitute a solid, standalone contribution. Importantly, to the best of our knowledge, it is the only disentanglement method capable of achieving near-perfect DCI. In contrast, many existing methods have theoretical issues that prevent them from achieving so, even with synthetic data.
> > > - Attaining disentanglement on real-world data requires substantial additional effort. Since commonly used augmentations for real-world datasets can not effectively link all samples within each cluster. This is a common fundamental challenge for augmentation-based self-supervised learning, which is, in fact, hard to resolve. Besides, practically available augmentations are typically stochastic, with their parameters not directly accessible. Our method instead requires deterministic augmentation with known and accessible augmentation parameters. Addressing this and other similar practical issues also requires a non-trivial amount of engineering effort. Therefore, we would like to leave this line of investigation as future work.
> > > - Genuine real-world disentanglement results are largely absent in existing self-supervised equivariant or disentanglement methods. Some works report classification accuracy as a proxy; however, this metric primarily reflects clustering ability and offers little insight into true disentanglement quality.

---

> > > ### Author Response · Authors · 2025-11-27
> > > **Further Response to Reviewer tJGx Part 2/2**
> > >
> > > For **the core issue: the method prespecifies subspace assignments instead of learning them**, we would like to clarify the following:
> > > - Our framework requires manually defining the number of independent subspaces and prespecifying subspace assignments. Given that the set of augmentations is user-defined, we may assume the user knows what factors each augmentation affects. For simple augmentations, manually specifying it is, in fact, a more principled way than learning it: learning only introduces unnecessary complexity.
> > > - For more complex scenarios, a promising future direction is that we simply provide a sufficiently large number of independent subspaces and let each augmentation automatically determine its required subspaces based on a sparse-as-possible usage principle. We would like to leave this as future work.

---

### Official Review · Reviewer_bD91 · 2025-11-01

**Soundness:** 4
**Presentation:** 4
**Contribution:** 4
**Rating:** 8
**Confidence:** 4

**Summary:**

This paper introduces Cluster-Dependent Rotational Equivariance for Disentanglement (CD-RED), a self-supervised framework designed to achieve perfect disentanglement of latent factors without labels.
The method builds on the observation that existing self-supervised and equivariant learning frameworks (e.g., SimCLR and CARE) either lack precise geometric control or impose global equivariance that limits cluster separation.

CD-RED introduces two key innovations:

> A two-stage training scheme that decouples invariance (via InfoNCE-based clustering) from equivariance (via local rotation learning).

> A cluster-dependent rotational model that encodes data variations explicitly as products of SO(2) rotations in orthogonal hyperspherical subspaces.

Theoretical proofs establish that CD-RED achieves “perfect equivariance,” and consequently, “perfect disentanglement” under mild assumptions.
Empirically, the method achieves near-perfect DCI (Disentanglement/Completeness/Informativeness) scores across both discrete and continuous latent datasets (i.e., MPI3D, Shape3D, 3DIdent, and 3DIEBench) outperforming prior self-supervised baselines such as EquiMOD and CARE by large margins.
Visual analyses (Figures 3, 13–15) demonstrate interpretable, axis-aligned subspaces that correspond neatly to independent generative factors (e.g., translation, rotation, hue).

**Strengths:**

+ Originality: Introduces the first explicit cluster-dependent rotational system and two-stage self-supervised disentanglement framework.

+ Rigor: Comprehensive theoretical grounding with formal proofs, linking equivariance and disentanglement in a provable manner.

+ Empirical quality: Extensive evaluation on four benchmark datasets, including both synthetic and realistic 3D settings, with DCI ≈ 1.0 across the board (Tables 1 & 4–12).

+ Clarity: Figures 1–3 effectively contrast SimCLR, CARE, and CD-RED, showing superior geometric structure.

+ Significance: Demonstrates that perfect disentanglement is achievable without supervision, a long-standing challenge in representation learning.

**Weaknesses:**

- Generality: Current evaluation is confined to controlled 3D datasets with clean augmentation-latent correspondences. Real-world visual domains or partial symmetries are not tested.

- Computational complexity: The two-stage pipeline (InfoNCE + rotational training) and per-cluster SO(2) alignment may scale poorly with large-scale or high-dimensional datasets.

- Accessibility of proofs: Some theoretical sections (Appendix C) are exceedingly detailed and might hinder comprehension without accompanying intuition.

- Empirical baselines: While strong, the comparison set omits recent group-equivariant self-supervised models (e.g., SE(3)-Transformers, GroupVAE variants), which could contextualize the method’s advantages beyond CARE and EquiMOD.

**Questions:**

> Scalability: How does CD-RED perform with hundreds of semantic clusters or higher-dimensional representations (e.g., d > 512)?

> Robustness: What happens if augmentations are imperfect or stochastic (e.g., illumination, blur)? Does CD-RED remain stable when equivariance assumptions are only approximately valid?

> Continuous vs. discrete factors: Can the framework adaptively infer the number of independent rotational subspaces without prior knowledge of m?

> Extension to SE(3): Since the method is based on SO(2) subspaces, can it be generalized to 3D rotational groups SO(3) or product groups like SE(3) for spatial tasks?

---

> ### Author Response · Authors · 2025-11-24
> **Response to Reviewer bD91 Part 1/2**
>
> Thank you for your thorough and insightful review. We also sincerely appreciate your recognition of the soundness and significance of our work. We address the concerns and questions you raised as follows.
>
> **Q1. Scalability: How does CD-RED perform with hundreds of semantic clusters or higher-dimensional representations (e.g., $d > 512$)?**
>
> - Theoretically, more semantic clusters would lead to a denser distribution of clusters over the hypersphere. However, this will not cause a problem as long as we set the rotational radius small enough such that the clusters are disjoint.
>
> - Higher-dimensional representations would tend to enlarge the separation between uniformly distributed clusters, which would actually make the learning easier. Theoretically, there should be at least one dimension that is not allocated to the rotational subspaces, and having more such dimensions is even preferable.
>
> **Q2-1. Robustness: What happens if augmentations are imperfect or stochastic (e.g., illumination, blur)?**
>
> - Augmentations are assumed to be able to link all samples within each semantic cluster. We assume imperfect augmentations mean they would fail to achieve so. As clusters are theoretically defined by augmentation-induced linkages, imperfect augmentations would cause one semantic cluster to fragment into multiple smaller augmentation-connected sub-clusters.
>
> - Our method, like other equivariance-based approaches that predict feature transformations from augmentation parameters, requires augmentations to be deterministic, with their parameters known and accessible. Therefore, commonly used stochastic augmentations cannot be directly applied. Nevertheless, a deterministic version with randomly sampled parameters can be used instead.
>
> **Q2-2. Robustness: Does CD-RED remain stable when equivariance assumptions are only approximately valid?**
>
> - Do you mean the equivariance $T_a f(x) = f(a(x))$ only approximately holds? This would not cause instability. It would only lead to slightly less accurate representations.
>
> **Q3. Continuous vs discrete factors: Can the framework adaptively infer the number of independent rotational subspaces without prior knowledge of $m$?**
>
> - Our framework, in its current form, requires manually defining the number of independent subspaces. As a promising future direction, we may simply provide a sufficiently large number of independent subspaces and let each augmentation automatically determine its required subspaces based on a sparse-as-possible usage principle.
>
> **Q4. Extension to SE(3): Since the method is based on SO(2) subspaces, can it be generalized to 3D rotational groups $SO(3)$ or product groups like SE(3) for spatial tasks?**
>
> - The SO(2) subspace in our framework serves merely as a container for a one-dimensional latent. Nevertheless, it is expressive enough to represent arbitrarily complex compositions of such one-dimensional latents. On the other hand, if the latent is one-dimensional by itself, using SO(3) is unnecessary and even causes trouble in disentanglement. SE(3) may be unsuitable for our framework, as our features are constrained to lie on a hypersphere: valid SE(3) reduces to SO(3). For spatial tasks, the translations (e.g., x-, y-, z-translations) will be encoded in separate SO(2) subspaces.

---

> > ### Author Response · Authors · 2025-11-24
> > **Response to Reviewer bD91 Part 2/2**
> >
> > **W1. Generality: The current evaluation is confined to controlled 3D datasets with clean augmentation-latent correspondences. Real-world visual domains or partial symmetries are not tested.**
> >
> > - Our work, in its current form, is primarily theoretical: we propose a new framework with provably strong disentangling guarantees and empirically substantiate these results on synthetic and controlled datasets. These existing elements in the paper are self-contained and constitute a solid, standalone contribution.
> >
> > - Attaining disentanglement on real-world data requires substantial additional effort. Since commonly used augmentations for real-world datasets can not effectively link all samples within each cluster. This is a common fundamental practical challenge for augmentation-based self-supervised learning, which is, in fact, hard to resolve. Besides, practically available augmentations are typically stochastic, with their parameters not directly accessible. Our method instead requires deterministic augmentation with known and accessible augmentation parameters. Addressing this and other similar practical issues also requires a non-trivial amount of engineering effort. Therefore, we would like to leave this line of investigation as future work.
> >
> > - Genuine real-world disentanglement results are largely absent in existing self-supervised equivariant or disentanglement methods. Some works report classification accuracy as a proxy; however, this metric primarily reflects clustering ability and offers little insight into true disentanglement quality.
> >
> > **W2. Computational complexity: The two-stage pipeline (InfoNCE + rotational training) and per-cluster SO(2) alignment may scale poorly with large-scale or high-dimensional datasets.**
> >
> > - Theoretically, our method will not be negatively impacted by the number of clusters, the number of samples in each cluster, or the data dimension of samples, as long as data augmentation can effectively link samples in each cluster.
> >
> > - Our computational cost is typically 2–4 times that of InfoNCE training due to the existence of the second stage. Nevertheless, the ratio is unlikely to increase notably as the datasets become larger or more complex, since InfoNCE also requires more training for complex datasets. We note that the cost of constructing and applying these rotation matrices is negligible.
> >
> > **W3. Accessibility of proofs: Some theoretical sections (Appendix C) are exceedingly detailed and might hinder comprehension without accompanying intuition.**
> >
> > - Thank you for the feedback. We have accompanied each proof with an intuitive explanation of the proof ideas.
> >
> > **W4. Empirical baselines: While strong, the comparison set omits recent group-equivariant self-supervised models (e.g., SE(3)-Transformers, GroupVAE variants), which could contextualize the method’s advantages beyond CARE and EquiMOD.**
> >
> > - To provide a more comprehensive empirical comparison beyond CARE and EquiMOD, we have added baselines IP-IRM and Eastwood et al. (2023), which exploit group-invariance for self-supervised disentanglement.
> >
> > - SE(3)-Transformers and related architectures are specifically designed to be equivariant to specific group actions, e.g., SE(3)/SO(3), which, by definition, cannot model more general equivariance. GroupVAE variants, on the other hand, rely on weak supervision and learn group-equivariant representations, which is quite a different setting from ours.
> >
> > - Although SE(3)-Transformers and GroupVAE variants are not directly comparable as empirical baselines in our setup, we now discuss these works in the revised related-work section to better position our contribution within the broader literature. If preferred, specifically tailored settings for effectively comparing them can also be provided.
> >
> > ---
> > We sincerely hope that the above clarifications adequately address your concerns. Please let us know if any aspect remains unclear. We would be glad to provide further details.

---

### Official Review · Reviewer_pLzm · 2025-11-01

**Soundness:** 2
**Presentation:** 2
**Contribution:** 2
**Rating:** 4
**Confidence:** 4

**Summary:**

This paper presents Cluster-Dependent Rotational Equivariance for Disentanglement (CD‑RED), a self-supervised framework that addresses the limitations of existing methods in learning disentangled representations. CD‑RED overcomes the problems of existing methods by learning cluster-dependent rotational equivariance, explicitly encoding variations as rotations via a direct product of groups in orthogonal hyperspherical subspaces. The method enables neat equivariance, uniformly distributed clusters, and theoretically and experimentally achieves disentangled representations.

**Strengths:**

1. The paper identifies an important issue: the imposition of uniform equivariance across all samples could reduce inter‑cluster distances for features on the hypersphere.

2. The theoretical formulation based on rotational SO(2) groups and block‑diagonal equivariant mapping seems solid.

3. On benchmark datasets (Shapes3D, MPI3D, 3DIdent, 3DIEBench), CD‑RED achieves near‑perfect DCI metrics, demonstrating stable disentanglement performance.

**Weaknesses:**

1. The analysis of related work appears insufficient and somewhat unclear, which weakens the overall contribution of this paper. The proposed method lies at the intersection of self-supervised learning and unsupervised disentangled representation learning; however, the paper overlooks a substantial body of literature on self-supervised clustering methods that aim to learn meaningful cluster structures without relying on invariance to data augmentations. Likewise, numerous studies in disentangled representation learning incorporate conditional invariance constraints, which should be discussed to better position this work within existing research.

2. The experiments are conducted exclusively on synthetic datasets rather than real-world data, which limits the practical validity and generalizability of the proposed method. Evaluating on real-world benchmarks would strengthen the empirical analysis and demonstrate the method’s broader applicability.

3. Although the primary stage employs InfoNCE to establish non-collapsed semantic clusters, this alone may not be sufficient to achieve true disentanglement. Such limitations could potentially lead to clustering errors, which would further affects disentanglement in the later stage, as the clusters are given as conditions.

**Questions:**

Does the proposed method ensure that errors or biases from the initial InfoNCE-based clustering do not propagate and negatively affect disentanglement in the subsequent stages, given that these clusters serve as conditioning factors?

---

> ### Author Response · Authors · 2025-11-24
> **Response to Reviewer pLzm Part 1/2**
>
> Thank you for your insightful review. We are glad that you found our identified issue important and our theoretical formulation solid. We address the concerns and questions you raised as follows.
>
> **Q1: Will the errors or biases from the initial InfoNCE-based clustering propagate and negatively affect disentanglement in the second stage?**
>
> The short answer is that errors or biases in clustering typically will not negatively impact the second stage, since the second stage can inherently correct such imperfections. More specifically,
>
> - InfoNCE can typically serve as an excellent initialization for the second stage, since it can theoretically produce uniformly distributed semantic clusters. In cases, the initial InfoNCE-based clustering can be imperfect, which has multiple potential causes, including imperfect optimization of the first stage, the mismatched clustering properties between the employed clustering algorithm and InfoNCE, and imperfect linking ability of the used data augmentations.
>
> - In the second stage, the clustering assignment of each sample is dynamically reassigned to its closest center during training. Theoretically and as we have also actually observed, the second stage is capable of recovering from imperfect clustering due to imperfect optimization of the first stage and the mismatched clustering properties between the employed clustering algorithm and InfoNCE, as long as the initial clustering is correct for the majority of the samples for each cluster. We have revised the manuscript, highlighting this in the paper.
>
> - In contrast, the imperfect clustering due to the limited linking ability of the used data augmentations can not be resolved in the second stage, since the second stage relies on the same assumption as InfoNCE clustering that data augmentations can link all the samples within each cluster. This stands as a long-term and significant challenge of augmentation-based self-supervised learning, and is beyond the scope of our paper.
>
> - For the clustering centers, simply fixing them as the first stage initialized is no problem, as long as they are distant enough, even if there is certain unevenness. In chasing a perfect uniform distribution of clusters, we may add the optional objective Eq. (31) in the appendix to promote a definite uniform distribution of clusters.
>
> **W1: The analysis of related work appears insufficient and somewhat unclear.**
>
> Thank you for this constructive feedback. To address this, we have supplemented a clearer and more comprehensive discussion of related work in Section 5.
>
> - We clarify that self-supervised clustering is not the core focus of our paper. It only serves as the initialization of our method for learning equivariance-based disentanglement. For self-supervised clustering, we introduced and employed InfoNCE as a representative, since it is theoretically well-grounded and, more crucially, its theoretical clustering properties are highly compatible with our needs in the second stage. We have supplemented some additional discussions on this in the revised manuscripts. If any important work is missing or more is preferred, we would be happy to amend further.
>
> - For "disentangled representation learning incorporates conditional invariance constraints", we are not entirely sure which specific line of work this refers to. If possible, we would be grateful if you could point us to some concrete examples. We assume you may be referring to works such as [1–4]. Compared with our method, these approaches focus on generative disentanglement using paired or grouped data and rely on weak supervision to impose conditional invariance constraints. In contrast, our method is purely self-supervised and achieves significantly stronger disentanglement. We have now included a discussion of these works in the revised manuscript.
>
> [1] Junxiang Chen and Kayhan Batmanghelich. Weakly supervised disentanglement by pairwise similarities. Proceedings of the AAAI Conference on Artificial Intelligence, pp. 3495–3502, Jun 2020.
>
> [2] Francesco Locatello, Ben Poole, Gunnar Raetsch, Bernhard Scholkopf, Olivier Bachem, and
> Michael Tschannen. Weakly-supervised disentanglement without compromises. International
> Conference on Machine Learning, 2020.
>
> [3] Zhu, Jiageng, Hanchen Xie, and Wael Abd-Almageed. Weakly supervised invariant representation learning via disentangling known and unknown nuisance factors. European Conference on Computer Vision. Cham: Springer Nature Switzerland, 2022.
>
> [4] Rui Shu, Yining Chen, Abhishek Kumar, Stefano Ermon, and Ben Poole. Weakly supervised disentanglement with guarantees. arXiv preprint arXiv:1910.09772, 2019.

---

> > ### Author Response · Authors · 2025-11-24
> > **Response to Reviewer pLzm Part 2/2**
> >
> > **W2: Evaluating on real-world benchmarks would strengthen the empirical analysis.**
> >
> > - Our work, in its current form, is primarily theoretical: we propose a new framework with provably strong disentangling guarantees and empirically substantiate these results on synthetic and controlled datasets. These existing elements in the paper are self-contained and constitute a solid, standalone contribution.
> >
> > - Attaining disentanglement on real-world data requires substantial additional effort. Since commonly used augmentations for real-world datasets can not effectively link all samples within each cluster. This is a common fundamental practical challenge for augmentation-based self-supervised learning, which is, in fact, hard to resolve. Besides, practically available augmentations are typically stochastic, with their parameters not directly accessible. Our method instead requires deterministic augmentation with known and accessible augmentation parameters. Addressing this and other similar practical issues also requires a non-trivial amount of engineering effort. Therefore, we would like to leave this line of investigation as future work.
> >
> > - Genuine real-world disentanglement results are largely absent in existing self-supervised equivariant or disentanglement methods. Some works report classification accuracy as a proxy; however, this metric primarily reflects clustering ability and offers little insight into true disentanglement quality.
> >
> > **W3: InfoNCE alone may not be sufficient to achieve true disentanglement. Such limitations could potentially lead to clustering errors and further affect disentanglement in the later stage.**
> >
> > - We clarify that we do not rely on InfoNCE to achieve disentanglement, nor can it do so in theory. We rely solely on its ability to learn semantic clusters, which it is theoretically capable of achieving.
> >
> > - Practical clustering errors, as discussed in response to Q1, typically will not negatively affect disentanglement in the second stage, since the clustering assignments are dynamically reassigned and the cluster centers can also be optimized towards definite uniformity.
> >
> > ---
> > We sincerely hope that the above clarifications adequately address your concerns. Please let us know if any aspect remains unclear. We would be glad to provide further details.

---

### Author Response · Authors · 2025-12-02
**Summary of Reviews and Responses**

Dear Chairs,

Thank you for taking the time to evaluate our work.

Given the special circumstance, we sincerely appreciate the Area Chair’s additional effort. For your convenience, we summarize the reviews and our responses as follows (using review codes such as W1/Q1 for cross-reference):
\
&nbsp;
- Overall / Ratings:
  - Our paper received initial ratings of **4, 8, 6, 4** from reviewers pLzm, bD91, tJGx, and q1ZX, respectively.
  - *bD91* acknowledges that our work **"demonstrates that perfect disentanglement is achievable without supervision, *a long-standing challenge in representation learning*"**.
\
&nbsp;
- Highlights from the Reviewers:
  - *pLzm:*
    - **"The paper identifies an important issue"**.
    - **"The theoretical formulation ... seems solid"**.
    \
    &nbsp;
  - *bD91:*
    - **"Introduces the first explicit cluster-dependent rotational system and two-stage self-supervised disentanglement framework"**.
    - **"Figures 1-3 effectively contrast SimCLR, CARE, and CD-RED, showing superior geometric structure"**.
    - **"Linking equivariance and disentanglement in a provable manner"**.
    \
    &nbsp;
  - *tJGx:*
    - **"Clear geometric intuition nicely complements theoretical intuitions"**.
    - **"Clean geometric design with explicit group structure"**.
    \
    &nbsp;
  - *q1ZX:*
    - **"Unifies content invariance and style equivariance into a single self-supervised pipeline"**.
    - **"Introduces a clean geometric model of disentanglement"**.
    \
    &nbsp;
- Reviewer-Specific Concerns and Responses:
  - *pLzm:*
    - **The analysis of related work appears insufficient** (W1)
      - Addressed via introducing an *Additional Related Work* section.
    \
    &nbsp;
  - *bD91:*
    - **Computational complexity for large-scale datasets** (W2)
      - Clarified: computational overhead relative to InfoNCE remains stable.
    - **Accessibility of proofs** (W3)
      - Accompanied each proof with an intuitive explanation.
    - **More baselines** (W4)
      - Added more baselines and discussions.
    - **Scalability w.r.t. cluster count or representation dimensionality** (Q1)
      - Clarified: no problem.
    - **What if augmentations are imperfect** (Q2)
      - Explained: cluster fragments into sub-clusters.
    - **Require manually specifying subspaces** (Q3)
      - Feasible for simple scenarios; a future direction is using a "sparse-as-possible" principle to automatically determine their required subspaces.
    \
    &nbsp;
  - *tJGx:*
    - **The applicability to real-world SSL augmentations** (W1, W2, W3)
      - Explained: can handle to a substantial extent.
    - **May not generalize to non-orthogonal** (W6)
      - Clarified: full disentanglement requires orthogonal; CD-RED can also handle non-orthogonal.
    \
    &nbsp;
  - *q1ZX:*
    - **Assume the number of style factors is known a priori** (W1-1)
      - Clarified: only need to know the number of style factors affected by the chosen set of augmentations.
    - **Assume augmentations change only the style but never the content** (W1-3, Q1)
      - Clarified: by definition, content is the latents that are invariant under the chosen set of augmentations.
    - **Assume knowing the exact magnitude** (W1-4, W1-2)
      - Clarified: only requires a lower bound, which can even reduce to whether the augmentation is trivial or not.
    - **Fairness of comparisons regarding post-processing** (W2)
      - Clarified: results both before and after post-processing were reported.
    - **Hyperparameter sensitivity** (Q2)
      - Clarified: not sensitive to these hyperparameters; typically fixed.
    \
    &nbsp;
- General Concerns and Responses:
  - **Evaluating on real-world benchmarks is missing** (pLzm-W2, bD91-W1, tJGx-W4, q1ZX-W3/Q3)
    - Our work, in its current form, is primarily theoretical. Existing elements in the paper are self-contained.
    - Attaining disentanglement on real-world data is non-trivial and requires substantial additional effort.
      - For real-world datasets, standard SSL augmentations cannot effectively link all the samples within each cluster.
      - Lack of evaluation metrics for disentanglement on real-world data.
    - Genuine real-world disentanglement results are largely absent in existing self-supervised equivariant or disentanglement methods.
  - **Initial clustering may have errors or biases** (pLzm-W3/Q1, tJGx-W5)
    - The second stage can correct initialization errors.
  - **Modeling stochastic augmentations** (tJGx-W6', bD91-Q2')
    - Our framework needs to access the augmentation parameters, so we need to use their deterministic counterparts (with their random augmentation parameters accessible).
    \
    &nbsp;
 - Revisions:
   - All substantive changes are highlighted in blue in the manuscript.
    \
    &nbsp;

We hope that our clarifications and additional experiments have sufficiently addressed the concerns. We sincerely appreciate your consideration.

Sincerely,

Authors

---

### Meta-Review · Area_Chair_9cYa · 2025-12-14

**Summary:**

Two of the four reviewers provided negative initial ratings. Although the authors responded to them, I think the lack of experiments on real data and the rationale of the assumption are not fully alleviated.  I recognize the rigor and significance of the theoretical contribution. The lack of experiments on real data makes me concerned about the rationale of the assumption. Therefore, I tend to reject this paper, but I wouldn't mind if the paper gets accepted

**Reviewer Concerns:**

Reviewer q1ZX and Reviewer pLzm provided negative initial ratings by conceding the lack of experiments on real data, the sufficiency of the related work, the fairness, and the rationale of the assumption. The authors provided additional discussion on the related work and the fairness. However, I think the lack of experiments on real data and the rationale of the assumption are not fully alleviated. If the experiments on real data can be provided, we may believe the rationale of the assumption. Only Reviewer tJGx participated in the discussion and remains concerned that several key limitations are not fully resolved.

**Reviewer Scores:**

Only Reviewer tJGx participated in the discussion. Although he provided positive initial ratings, he remains concerned that several key limitations are not fully resolved. Therefore, I believe he may not change his scores. Based on the lack of experiments on real data and the rationale of the assumption, I think Reviewer q1ZX and Reviewer pLzm may not change their scores towards acceptance. Therefore, I think the final scores may be the same as the initial ones as follows.
- q1ZX		Rating: 4 / Confidence: 4
- pLzm		Rating: 4 / Confidence: 4
- tJGx		Rating: 6 / Confidence: 4
- bD91		Rating: 8 / Confidence: 4

---

### Decision · Program_Chairs · 2026-01-26

Reject